# BEYOND STATIONARITY:
# CONVERGENCE ANALYSIS OF STOCHASTIC SOFTMAX POLICY GRADIENT METHODS

**Sara Klein, Simon Weissmann & Leif Döring**
Institute of Mathematics, University of Mannheim
{sara.klein, simon.weissmann, leif.doering}@uni-mannheim.de

## ABSTRACT

Markov Decision Processes (MDPs) are a formal framework for modeling and solving sequential decision-making problems. In finite-time horizons such problems are relevant for instance for optimal stopping or specific supply chain problems, but also in the training of large language models. In contrast to infinite horizon MDPs optimal policies are not stationary, policies must be learned for every single epoch. In practice all parameters are often trained simultaneously, ignoring the inherent structure suggested by dynamic programming. This paper introduces a combination of dynamic programming and policy gradient called dynamic policy gradient, where the parameters are trained backwards in time.

For the tabular softmax parametrisation we carry out the convergence analysis for simultaneous and dynamic policy gradient towards global optima, both in the exact and sampled gradient settings without regularisation. It turns out that the use of dynamic policy gradient training much better exploits the structure of finite-time problems which is reflected in improved convergence bounds.

## 1 INTRODUCTION

Policy gradient (PG) methods continue to enjoy great popularity in practice due to their model-free nature and high flexibility. Despite their far-reaching history (Williams, 1992; Sutton et al., 1999; Konda & Tsitsiklis, 1999; Kakade, 2001), there were no proofs for the global convergence of these algorithms for a long time. Nevertheless, they have been very successful in many applications, which is why numerous variants have been developed in the last few decades, whose convergence analysis, if available, was mostly limited to convergence to stationary points (Pirotta et al., 2013; Schulman et al., 2015; Papini et al., 2018; Clavera et al., 2018; Shen et al., 2019; Xu et al., 2020b; Huang et al., 2020; Xu et al., 2020a; Huang et al., 2022). In recent years, notable advancements have been achieved in the convergence analysis towards global optima (Fazel et al., 2018; Agarwal et al., 2021; Mei et al., 2020; Bhandari & Russo, 2021; 2022; Cen et al., 2022; Xiao, 2022; Yuan et al., 2022; Alfano & Rebeschini, 2023; Johnson et al., 2023). These achievements are partially attributed to the utilisation of (weak) gradient domination or Polyak-Łojasiewicz (PL) inequalities (lower bounds on the gradient) (Polyak, 1963).

As examined in Karimi et al. (2016) a PL-inequality and $\beta$-smoothness (i.e. $\beta$-Lipschitz continuity of the gradient) implies a linear convergence rate for gradient descent methods. In certain cases, only a weaker form of the PL inequality can be derived, which states that it is only possible to lower bound the norm of the gradient instead of the squared norm of the gradient by the distance to the optimum. Despite this limitation, $\mathcal{O}(1/n)$-convergence can still be achieved in some instances.

This article deals with PG algorithms for finite-time MDPs. Finite-time MDPs differ from discounted infinite-time MDPs in that the optimal policies are not stationary, i.e. depend on the epochs. While a lot of recent theoretical research focused on discounted MDPs with infinite-time horizon not much is known for finite-time MDPs. However, there are many relevant real world applications which require non-stationary finite-time solutions such as inventory management in hospital supply chains (Abu Zwaida et al., 2021) or optimal stopping in finance (Becker et al., 2019). There is a prevailing thought that finite-time MDPs do not require additional scrutiny as they can be transformed

into infinite horizon MDPs by adding an additional time-coordinate. Seeing finite-time MDPs this way leads to a training procedure in which parameters for all epochs are trained simultaneously, see for instance Guin & Bhatnagar (2023). While there are practical reasons to go that way, we will see below that ignoring the structure of the problem yields worse convergence bounds. The aim of this article is two-fold. Firstly, we analyse the simultaneous PG algorithm. The analysis for exact gradients goes along arguments of recent articles, the analysis of the stochastic PG case is novel. Secondly, we introduce a new approach to PG for finite-time MDPs. We exploit the dynamic programming structure and view the MDP as a nested sequence of contextual bandits. Essentially, our algorithm performs a sequence of PG algorithms backwards in time with carefully chosen epoch dependent training steps. We compare the exact and stochastic analysis to the simultaneous approach. Dynamic PG can bee seen as a concrete algorithm for Policy Search by Dynamic Programming, where policy gradient is used to solve the one-step MDP (Bagnell et al., 2003; Scherrer, 2014). There are some recent articles also studying PG of finite-time horizon MDPs from a different perspective considering fictitious discount algorithms (Guo et al., 2022) or finite-time linear quadratic control problems (Hambly et al., 2021; 2023; Zhang et al., 2021; 2023b;a; Zhang & Başar, 2023).

This article can be seen to extend a series of recent articles from discounted MDPs to finite-time MDPs. In Agarwal et al. (2021), the global asymptotic convergence of PG is demonstrated under tabular softmax parametrisation, and convergence rates are derived using log-barrier regularisation and natural policy gradient. Building upon this work, Mei et al. (2020) showed the first convergence rates for PG using non-uniform PL-inequalities (Mei et al., 2021), specifically for tabular softmax parametrisation. The convergence rate relies heavily on the discount factor as $(1-\gamma)^{-6}$ and does not not readily convert to non-discounted MDPs. Through careful analysis, we establish upper bounds involving $H^5$ for simultaneous PG, contrasting with $H^3$ for dynamic PG. Essentially, dynamic PG offers a clear advantage. Examining the PG theorem for finite-

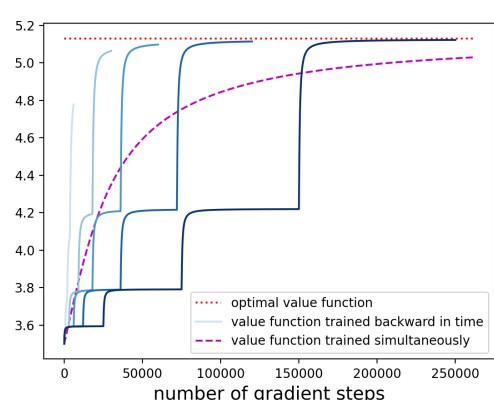

Figure 1: Evolution of the value function during training.

time MDPs reveals that early epochs should be trained less if policies for later epochs are suboptimal. A badly learned $Q$-function-to-go leads to badly directed gradients in early epochs. Thus, simultaneous training yields ineffective early epoch training, addressed by our dynamic PG algorithm, optimizing policies backward in time with more training steps. To illustrate this phenomenon we implemented a simple toy example where the advantage of dynamic PG becomes visible. In Figure 1 one can see 5 simulations of the dynamic PG with different target accuracies (blue curves) plotted against one version of the simultaneous PG with target accuracy $0.1$ (dashed magenta curve). The time-horizon is chosen as $H = 5$. More details on the example can be found in Appendix E.

A main further contribution of this article is a stochastic analysis, where we abandon the assumption that the exact gradient is known and focus on the model free stochastic PG method. For this type of algorithm, very little is known about convergence to global optima even in the discounted case. Many recent articles consider variants such as natural PG or mirror descent to analyse the stochastic scenario (Agarwal et al., 2021; Fatkhullin et al., 2023; Xiao, 2022; Alfano et al., 2023). Ding et al. (2022) derive complexity bounds for entropy-regularised stochastic PG. They use a well-chosen stopping time which measures the distance to the set of optimal parameters, and simultaneously guarantees convergence to the regularised optimum prior to the occurrence of the stopping time by using a small enough step size and large enough batch size. As we are interested in convergence to the unregularised optimum, we consider stochastic softmax PG without regularisation. Similar to the previous idea, we construct a different stopping time, which allows us to derive complexity bounds for an approximation arbitrarily close to the global optimum that does not require a set of optimal parameters and this is relevant when considering softmax parametrisation. To the best of our knowledge, the results presented in this paper provide the first convergence analysis for dynamic programming inspired PG under softmax parametrisation in the finite-time MDP setting. Both for exact and batch sampled policy gradients without regularisation.

## 2 FINITE-TIME HORIZON MDPs AND POLICY GRADIENT METHODS.

A finite-time MDP is defined by a tuple $(\mathcal{H}, \mathcal{S}, \mathcal{A}, r, p)$ with $\mathcal{H} = \{0, \ldots, H-1\}$ decision epochs, finite state space $\mathcal{S} = \mathcal{S}_0 \cup \cdots \cup \mathcal{S}_{H-1}$, finite action space $\mathcal{A} = \bigcup_{s \in \mathcal{S}} \mathcal{A}_s$, a reward function $r : \mathcal{S} \times \mathcal{A} \to \mathbb{R}$ and transition function $p : \mathcal{S} \times \mathcal{A} \to \Delta(\mathcal{S})$ with $p(\mathcal{S}_{h+1}|s, a) = 1$ for every $h < H-1$, $s \in \mathcal{S}_h$ and $a \in \mathcal{A}_s$. Here $\Delta(D)$ denotes the set of all probability measures over a finite set $D$.

Throughout the article $\pi = (\pi_h)_{h=0}^{H-1}$ denotes a time-dependent policy, where $\pi_h : \mathcal{S}_h \to \Delta(\mathcal{A})$ is the policy in decision epoch $h \in \mathcal{H}$ with $\pi_h(\mathcal{A}_s|s) = 1$ for every $s \in \mathcal{S}_h$. It is well-known that in contrast to discounted infinite-time horizon MDPs non-stationary policies are needed to optimise finite-time MDPs. An optimal policy in time point $h$ depends on the time horizon until the end of the problem (see for example Puterman (2005)). The epoch-dependent value functions under policy $\pi$ are defined by

$$V_h^{\pi_{(h)}}(\mu_h) := \mathbb{E}_{\mu_h}^{\pi_{(h)}} \Big[ \sum_{k=h}^{H-1} r(S_k, A_k) \Big], \quad h \in \mathcal{H}, \tag{1}$$

where $\mu_h$ is an initial distribution, $\pi_{(h)} = (\pi_k)_{k=h}^{H-1}$ denotes the sub-policy of $\pi$ from $h$ to $H-1$ and $\mathbb{E}_{\mu_h}^{\pi_{(h)}}$ is the expectation under the measure such that $S_h \sim \mu_h$, $A_k \sim \pi_k(\cdot|S_k)$ and $S_{k+1} \sim p(\cdot|S_k, A_k)$ for $h \le k < H-1$. The target is to find a (time-dependent) policy that maximises the state-value function $V_0$ at time 0. In the following we will discuss two approaches to solve finite-time MDPs with PG:

- An algorithm that is often used in practice, where parametrised policies are trained simultaneously, i.e. the parameters for $\pi_0, \ldots, \pi_{H-1}$ are trained at once using the objective $V_0$.

- A new algorithm that trains the parameters sequentially starting at the last epoch. We call this scheme dynamic PG because it combines dynamic programming (backwards induction) and PG.

In fact, one can also consider PG algorithms that train stationary policies (i.e. independent of $h$) for finite-time MDPs. However, this violates the intrinsic nature of finite-time MDPs (optimal policies will only be stationary in trivial cases). In order to carry out a complete theoretical analysis assumptions are required. In this article we will assume that all policies are softmax parametrised, an assumption that appeared frequently in the past years. It is a first step towards a full understanding and already indicates why PG methods should use the dynamic programming structure inherent in finite-time MDPs. This paper should not be seen as limited to the softmax case, but more like a kick-off to analyse a new approach which is beneficial in many scenarios.

**Simultaneous Policy Gradient.** Let us start by formulating the simultaneous PG algorithm that is often used in practice. The action spaces may depend on the current state and the numbers of possible actions in epoch $h$ is denoted by $d_h = \sum_{s \in \mathcal{S}_h} |\mathcal{A}_s|$. To perform a PG algorithm all policies $\pi_h$ (or the entire policy $\pi$) must be parametrised. While the algorithm does not require a particular policy we will analyse the tabular softmax parametrisation

$$\pi^\theta(a|s_h) = \frac{\exp(\theta(s_h, a))}{\sum_{a'} \exp(\theta(s_h, a'))}, \quad \theta = (\theta(s_h, a))_{s_h \in \mathcal{S}^{[\mathcal{H}]}, a \in \mathcal{A}_{s_h}} \in \mathbb{R}^{\sum_h d_h}, \tag{2}$$

where the notation $\mathcal{S}^{[\mathcal{H}]}$ defines the enlarged state space, containing all possible states associated to their epoch (see Remark A.1 for more details). The tabular softmax parametrisation uses a single parameter for each possible state-action pair at all epochs. Other parametrised policies, e.g. neural networks, take states from all epochs, i.e. from the enlarged state space $\mathcal{S}^{[\mathcal{H}]}$, as input variables. The simultaneous PG algorithm trains all parameters at once and solves the optimisation problem (to maximize the state value function at time 0) by gradient ascent over all parameters (all epochs) simultaneously.

Most importantly, the algorithm does not treat epochs differently, the same training effort goes into all epochs. For later use the objective function will be denoted by

$$J(\theta, \mu) := V_0^{\pi^\theta}(\mu) = \mathbb{E}_\mu^{\pi^\theta} \Big[ \sum_{h=0}^{H-1} r(S_h, A_h) \Big] \tag{3}$$

---

**Algorithm 1:** Simultaneous Policy Gradient for finite-time MDPs

---

**Result:** Approximate policy $\hat{\pi}^* \approx \pi^*$

initialise $\theta^{(0)} \in \mathbb{R}^{\sum_h d_h}$

Choose fixed step sizes $\eta > 0$, number of training steps $N$ and start distribution $\mu$

**for** $n = 0, \ldots, N-1$ **do**

$\qquad \theta^{(n+1)} = \theta^{(n)} + \eta \, \nabla_\theta V_0^{\pi^{\theta^{(n)}}}(\mu)\big|_{\theta^{(n)}}$

**end**

Set $\hat{\pi}^* = \pi^{\theta^{(N)}}$

---

Furthermore, let $\rho_\mu^{\pi^\theta}(s) = \sum_{h=0}^{H-1} \mathbb{P}_\mu^{\pi^\theta}(S_h = s)$ be the state-visitation measure on $\mathcal{S}$ and $d_\mu^{\pi^\theta}(s) = \frac{1}{H}\rho_\mu^{\pi^\theta}(s)$ be the normalised state-visitation distribution. We denote by $J^*(\mu) = \sup_\theta J(\theta, \mu)$ the optimal value of the objective function and note that $J^*(\mu) = V_0^*(\mu) = \sup_{\pi: \text{Policy}} V_0^\pi(\mu)$ under the tabular softmax parametrisation, as an optimal policy can be approximated arbitrarily well.

**Dynamic Policy Gradient.**    First of all, recall that the inherent structure of finite-time MDPs is a backwards induction principle (dynamic programming), see for instance (Puterman, 2005). To see backwards induction used in learning algorithms we refer for instance to Bertsekas & Tsitsiklis (1996, Sec 6.5). In a way, finite-time MDPs can be viewed as nested contextual bandits. The dynamic PG approach suggested in this article builds upon this intrinsic structure and sets on top a PG scheme. Consider $H$ different parameters $\theta_0, \ldots, \theta_{H-1}$ such that $\theta_h \in \mathbb{R}^{d_h}$. A parametric policy $(\pi^{\theta_h})_{h=0}^{H-1}$ is defined such that the policy in epoch $h$ depends only on the parameter $\theta_h$. An example is the tabular softmax parametrisation formulated slightly differently than above. For each decision epoch $h \in \mathcal{H}$ the tabular softmax parametrisation is given by

$$\pi^{\theta_h}(a|s) = \frac{\exp(\theta_h(s,a))}{\sum_{a' \in \mathcal{A}} \exp(\theta_h(s,a'))}, \quad \theta_h = (\theta_h(s,a))_{s \in \mathcal{S}_h, a \in \mathcal{A}_s} \in \mathbb{R}^{d_h}. \tag{4}$$

The total dimension of the parameter tensor $(\theta_0, \ldots, \theta_{H-1})$ equals the one of $\theta$ from the Equation 2 because $\theta_h(s_h, a) = \theta(s_h, a)$ for $s_h \in \mathcal{S}_h \subset \mathcal{S}^{[\mathcal{H}]}$. The difference is that the epoch dependence is made more explicit in Equation 4.

The main idea of this approach is as follows. The dynamic programming perspective suggests to learn policies backwards in time. Thus, we start by training the last parameter vector $\theta_{H-1}$ on the sub-problem $V_{H-1}$, a one-step MDP which can be viewed as contextual bandit. After convergence up to some termination condition, it is known how to act near optimality in the last epoch and one can proceed to train the parameter vector from previous epochs by exploiting the knowledge of acting near optimal in the future. This is what the proposed dynamic PG algorithm does. A policy is trained up to some termination condition and then used to optimise an epoch earlier.

---

**Algorithm 2:** Dynamic Policy Gradient for finite-time MDPs

---

**Result:** Approximate policy $\hat{\pi}^* \approx \pi^*$

initialise $\theta^{(0)} = (\theta_0^{(0)}, \ldots, \theta_{H-1}^{(0)}) \in \Theta$

**for** $h = H-1, \ldots, 0$ **do**

$\qquad$ Choose fixed step size $\eta_h$, number of training steps $N_h$ and start distribution $\mu_h$

$\qquad$ **for** $n = 0, \ldots, N_h - 1$ **do**

$\qquad \qquad \theta_h^{(n+1)} = \theta_h^{(n)} + \eta_h \nabla_{\theta_h} V_h^{(\pi^{\theta_h}, \hat{\pi}_{(h+1)}^*)}(\mu_h)\big|_{\theta_h^{(n)}}$

$\qquad$ **end**

$\qquad$ Set $\hat{\pi}_h^* = \pi^{\theta_h^{(N_h)}}$

**end**

---

A bit of notation is needed to analyse this approach. Given any fixed policy $\tilde{\pi}$, the objective function $J_h$ in epoch $h$ is defined to be the $h$-state value function in state under the extended policy

$$(\pi^{\theta_h}, \tilde{\pi}_{(h+1)}) := (\pi^{\theta_h}, \tilde{\pi}_{h+1}, \ldots, \tilde{\pi}_{H-1}),$$

$$J_h(\theta_h, \tilde{\pi}_{(h+1)}, \mu_h) := V_h^{(\pi^{\theta_h}, \tilde{\pi}_{(h+1)})}(\mu_h) = \mathbb{E}_{\mu_h}^{(\pi^{\theta_h}, \tilde{\pi}_{(h+1)})}\Big[\sum_{k=h}^{H-1} r(S_k, A_k)\Big]. \tag{5}$$

While the notation is a bit heavy the intuition behind is easy to understand. If the policy after epoch $h$ is already trained (this is $\tilde{\pi}_{(h+1)}$) then $J_h$ as a function of $\theta_h$ is the parametrised dependence of the value function when only the policy for epoch $h$ is changed. Gradient ascent is then used to find a parameter $\theta_h^*$ that maximises $J_h(\cdot, \tilde{\pi}_{(h+1)}, \delta_s)$, for all $s \in \mathcal{S}_h$, where $\delta_s$ the dirac measure on $s$. Note that to train $\theta_h$ one chooses $\tilde{\pi}_{(h+1)} = \hat{\pi}_{(h+1)}^*$ in Algorithm 2.

A priori it is not clear if simultaneous or dynamic programming inspired training is more efficient. Dynamic PG has an additional loop but trains less parameters at once. We give a detailed analysis for the tabular softmax parametrisation but want to give a heuristic argument why simultaneous training is not favorable. The policy gradient theorem, see Theorem A.5, states that

$$\nabla J(\theta, \mu) = \sum_{s_h \in \mathcal{S}^{[\mathcal{H}]}} \tilde{\rho}_\mu^{\pi^\theta}(s_h) \sum_{a \in \mathcal{A}_{s_h}} \pi^\theta(a|s_h) \nabla \log(\pi^\theta(a|s_h)) Q_h^{\pi^\theta}(s_h, a),$$

involving $Q$-values under the current policy[1]. It implies that training policies at earlier epochs are massively influenced by estimation errors of $Q_h^{\pi^\theta}$. Reasonable training of optimal decisions is only possible if all later epochs have been trained well, i.e. $Q_h^{\pi^\theta} \approx Q_h^*$. This may lead to inefficiency in earlier epochs when training all epochs simultaneously. It is important to note that the policy gradient formula is independent of the parametrisation. While our precise analysis is only carried out for tabular softmax parametrisations this general heuristic remains valid for all classes of policies.

*Assumption* 2.1. Throughout the remaining manuscript we assume that the rewards are bounded in $[0, R^*]$, for some $R^* > 0$. The positivity is no restriction of generality, bounded negative rewards can be shifted using the base-line trick.

In what follows we will always assume the tabular softmax parametrisation and analyse both PG schemes. First under the assumption of exact gradients, then with sampled gradients à la REIN-FORCE.

# 3 CONVERGENCE OF SOFTMAX POLICY GRADIENT WITH EXACT GRADIENTS

In the following, we analyse the convergence behavior of the simultaneous as well as the dynamic approach under the assumption to have access to exact gradient computation. The presented convergence analysis in both settings is inspired from the discounted setting considered recently in Agarwal et al. (2021); Mei et al. (2020). The idea is to combine smoothness of the objective function and a (weak) PL-inequality in order to derive a global convergence result.

## 3.1 SIMULTANEOUS POLICY GRADIENT

To prove convergence in the simultaneous approach we will interpret the finite-time MDP as an undiscounted stationary problem with state-space $\mathcal{S}^{[H]}$ and deterministic absorption time $H$. This MDP is undiscounted but terminates in finite-time. Building upon Agarwal et al. (2021); Mei et al. (2020); Yuan et al. (2022) we prove that the objective function defined in Equation 3 is $\beta$-smooth with parameter $\beta = H^2 R^*(2 - \frac{1}{|\mathcal{A}|})$ and satisfies a weak PL-inequality of the form

$$\|\nabla J(\theta, \mu)\|_2 \geq \frac{\min_{s_h \in \mathcal{S}^{[\mathcal{H}]}} \pi^\theta(a^*(s_h)|s_h)}{\sqrt{|\mathcal{S}^{[\mathcal{H}]}|}} \Big\|\frac{d_\mu^{\pi^*}}{d_\mu^{\pi^\theta}}\Big\|_\infty^{-1} (J^*(\mu) - J(\theta, \mu)).$$

Here $\pi^*$ denotes a fixed but arbitrary deterministic optimal policy for the enlarged state space $\mathcal{S}^{[\mathcal{H}]}$ and $a^*(s_h) = \text{argmax}_{a \in \mathcal{A}_{s_h}} \pi^*(a|s_h)$ is the best action in state $s_h$. The term

$$\Big\|\frac{d_\mu^{\pi^*}}{d_\mu^{\pi^\theta}}\Big\|_\infty := \max_{s \in \mathcal{S}} \frac{d_\mu^{\pi^*}(s)}{d_\mu^{\pi^\theta}(s)} \tag{6}$$

---

[1]See Appendix A, Equation 12 and Equation 13 for the definition of the state-action value function $Q$ and the enlarged state visitation measure $\tilde{\rho}$.

is the distribution mismatch coefficient introduced in Agarwal et al. (2021, Def 3.1). Both properties are shown in Appendix B.1. To ensure that the distribution mismatch coefficient can be bounded from below uniformly in $\theta$ (see also Remark B.4) we make the following assumption.

*Assumption* 3.1. For the simultaneous PG algorithm we assume that the state space is constant over all epochs, i.e. $\mathcal{S}_h = \mathcal{S}$ for all epochs.

As already pointed out in Mei et al. (2020) one key challenge in providing global convergence is to bound the term $\min_{s \in \mathcal{S}} \pi^\theta(a_h^*(s)|s)$ from below uniformly in $\theta$ appearing in the gradient ascent updates. Techniques introduced in Agarwal et al. (2021) can be extended to the finite-horizon setting to prove asymptotic convergence towards global optima. This can then be used to bound $c = c(\theta^{(0)}) = \inf_n \min_{s \in \mathcal{S}} \pi^{\theta^{(n)}}(a_h^*(s)|s) > 0$ (Lemma B.5). Combining smoothness and the gradient domination property results in the following global convergence result.

**Theorem 3.2.** *Under Assumption 3.1, let $\mu$ be a probability measure such that $\mu(s) > 0$ for all $s \in \mathcal{S}$, let $\eta = \frac{1}{5H^2 R^*}$ and consider the sequence $(\theta^{(n)})$ generated by Algorithm 1 with arbitrary initialisation $\theta^{(0)}$. For $\epsilon > 0$ choose the number of training steps as $N = \frac{10 H^5 R^* |\mathcal{S}|}{c^2 \epsilon} \left\| \frac{d_\mu^{\pi^*}}{\mu} \right\|_\infty^2$. Then it holds that*

$$V_0^*(\mu) - V_0^{\pi^{\theta^{(N)}}}(\mu) \le \epsilon.$$

One can compare this result to Mei et al. (2020, Thm 4) for discounted MDPs. A discounted MDP can be seen as an undiscounted MDP stopped at an independent geometric random variable with mean $(1-\gamma)^{-1}$. Thus, it comes as no surprise that algorithms with deterministic absorption time $H$ have analogous estimates with $H$ instead of $(1-\gamma)^{-1}$. See Remark B.6 for a detailed comparison. Furthermore, it is noteworthy that it cannot be proven that $c$ is independent of $H$. We omitted this dependency when we compare to the discounted case because the model dependent constant there could also depend on $\gamma$ in the same sense.

## 3.2 Dynamic Policy Gradient

We now come to the first main contribution of this work, an improved bound for the convergence of the dynamic PG algorithm. The optimisation objectives are $J_h$ defined in Equation 5. The structure of proving convergence is as follows. For each fixed $h \in \mathcal{H}$ we provide global convergence given that the policy after $h$ is fixed and denoted by $\tilde{\pi}$. After having established bounds for each decision epoch, we apply backwards induction to derive complexity bounds on the total error accumulated over all decision epochs. The $\beta$-smoothness for different $J_h$ is then reflected in different training steps for different epochs.

The backwards induction setting can be described as a nested sequence of contextual bandits (one-step MDPs) and thus, can be analysed using results from the discounted setting by choosing $\gamma = 0$. Using PG estimates for dicounted MDPs (Mei et al., 2020; Yuan et al., 2022) we prove in Appendix B.2 that the objective $J_h$ from Equation 5 is a smooth function in $\theta_h$ with parameter $\beta_h = 2(H - h)R^*$ and satisfies also a weak PL-inequality of the form

$$\|\nabla J_h(\theta_h, \tilde{\pi}_{(h+1)}, \mu_h)\|_2 \ge \min_{s \in \mathcal{S}_h} \pi^{\theta_h}(a_h^*(s)|s)(J_h^*(\tilde{\pi}_{(h+1)}, \mu_h) - J_h(\theta_h, \tilde{\pi}_{(h+1)}, \mu_h)).$$

It is crucial to keep in mind that classical theory from non-convex optimisation tells us that less smooth (large $\beta$) functions must be trained with more gradient steps. It becomes clear that the dynamic PG algorithm should spend less training effort on later epochs (earlier in the algorithm) and more training effort on earlier epochs (later in the algorithm). In fact, we make use of this observation by applying backwards induction in order to improve the convergence behavior depending on $H$ (see Theorem 4.2). The main challenge is again to bound $\min_{s \in \mathcal{S}} \pi^{\theta_h}(a_h^*(s)|s)$ from below uniformly in $\theta_h$ appearing in the gradient ascent updates from Algorithm 2. In this setting the required asymptotic convergence follows directly from the one-step MDP viewpoint using $\gamma = 0$ obtained in Agarwal et al. (2021, Thm 5) and it holds $c_h = \inf_{n \ge 0} \min_{s \in \mathcal{S}_h} \pi^{\theta_h^{(n)}}(a_h^*(s)|s) > 0$ (Lemma B.10).

There is another subtle advantage in the backwards induction point of view. The contextual bandit interpretation allows using refinements of estimates for the special case of contextual bandits. A slight generalisation of work of Mei et al. (2020) for stochastic bandits shows that the unpleasant unknown constants $c_h$ simplify if the PG algorithm is uniformly initialised:

**Proposition 3.3.** *For fixed $h \in \mathcal{H}$, let $\mu_h$ be a probability measure such that $\mu_h(s) > 0$ for all $s \in \mathcal{S}_h$ and let $0 < \eta_h \leq \frac{1}{2(H-h)R^*}$. Let $\theta_h^{(0)} \in \mathcal{R}^{d_h}$ be an initialisation such that the initial policy is a uniform distribution, then $c_h = \frac{1}{|\mathcal{A}|} > 0$.*

This property is in sharp contrast to the simultaneous approach, where to the best of our knowledge it is not known how to lower bound $c$ explicitly. Comparing the proofs of $c > 0$ and $c_h > 0$ one can see that this advantage comes from the backward inductive approach and is due to fixed future policies which are not changing during training. For fixed decision epoch $h$ combining $\beta$-smoothness and weak PL inequality yields the following global convergence result for the dynamic PG generated in Algorithm 2.

**Lemma 3.4.** *For fixed $h \in \mathcal{H}$, let $\mu_h$ be a probability measure such that $\mu_h(s) > 0$ for all $s \in \mathcal{S}_h$, let $\eta_h = \frac{1}{2(H-h)R^*}$ and consider the sequence $(\theta_h^{(n)})$ generated by Algorithm 2 with arbitrary initialisation $\theta_h^{(0)}$ and $\tilde{\pi}$. For $\epsilon > 0$ choose the number of training steps as $N_h = \frac{4(H-h)R^*}{c_h^2 \epsilon}$. Then it holds that*

$$V_h^{(\pi_h^*, \tilde{\pi}_{(h+1)})}(\mu_h) - V_h^{(\pi^{\theta_h^{(N_h)}}, \tilde{\pi}_{(h+1)})}(\mu_h) \leq \epsilon$$

*Moreover, if $\theta_h^{(0)}$ initialises the uniform distribution the constants $c_h$ can be replaced by $\frac{1}{|\mathcal{A}|}$.*

The error bound depends on the time horizon up to the last time point, meaning intuitively that an optimal policy for earlier time points in the MDP (smaller $h$) is harder to achieve and requires a longer learning period then later time points ($h$ near to $H$). We remark that the assumption on $\mu_h$ is not a sharp restriction and can be achieved by using a strictly positive start distribution $\mu$ on $\mathcal{S}_0$ followed by a uniformly distributed policy. Note that assuming a positive start distribution is common in the literature and Mei et al. (2020) showed the necessity of this assumption. Accumulating errors over time we can now derive the analogous estimates to the simultaneous PG approach. We obtain a linear accumulation such that an $\frac{\epsilon}{H}$-error in each time point $h$ results in an overall error of $\epsilon$ which appears naturally from the dynamic programming structure of the algorithm.

**Theorem 3.5.** *For all $h \in \mathcal{H}$, let $\mu_h$ be probability measures such that $\mu_h(s) > 0$ for all $s \in \mathcal{S}_h$, let $\eta_h = \frac{1}{2(H-h)R^*}$. For $\epsilon > 0$ choose the number of training steps as $N_h = \frac{4(H-h)HR^*}{c_h^2 \epsilon} \left\| \frac{1}{\mu_h} \right\|_\infty$. Then for the final policy from Algorithm 2, $\hat{\pi}^* = (\pi^{\theta_0^{(N_0)}}, \dots, \pi^{\theta_{H-1}^{(N_{H-1})}})$, it holds for all $s \in \mathcal{S}_0$ that*

$$V_0^*(s) - V_0^{\hat{\pi}^*}(s) \leq \epsilon.$$

*If $\theta_h^{(0)}$ initialises the uniform distribution the constants $c_h$ can be replaced by $\frac{1}{|\mathcal{A}|}$.*

### 3.3 COMPARISON OF THE ALGORITHMS

Comparing Theorem 3.5 to the convergence rate for simultaneous PG in Theorem 3.2, we first highlight that the constant $c_h$ in the dynamic approach can be explicitly computed under uniform initialisation. This has not yet been established in the simultaneous PG (see Remark B.11) and especially it cannot be guaranteed that $c$ is independent of the time horizon. Second, we compare the overall dependence of the training steps on the time horizon. In the dynamic approach $\sum_h N_h$ scales with $H^3$ in comparison to $H^5$ in the convergence rate for the simultaneous approach. In particular for large time horizons the theoretical analysis shows that reaching a given accuracy is more costly for simultaneous training of parameters. In the dynamic PG the powers are due to the smoothness constant, the $\frac{\epsilon}{H}$ error which we have to achieve in every epoch and finally the sum over all epochs. In comparison, in the simultaneous PG a power of 2 is due to the smoothness constant, another power of 2 is due to the distribution mismatch coefficient in the PL-inequality which we need to bound uniformly in $\theta$ (see also Remark B.3) and the last power is due to the enlarged state space $|\mathcal{S}^{[H]}| = |\mathcal{S}|H$. Note that we just compare upper bounds. However, we refer to Appendix E for a toy example visualising that the rate of convergence in both approaches is of order $\mathcal{O}(\frac{1}{n})$ and the constants in the dynamic approach are indeed better then for the simultaneous approach.

# 4 CONVERGENCE ANALYSIS OF STOCHASTIC SOFTMAX POLICY GRADIENT

In the previous section, we have derived global convergence guarantees for solving a finite-time MDP via simultaneous as well as dynamic PG with exact gradient computation. However, in practical scenarios assuming access to exact gradients is not feasible, since the transition function $p$ of the underlying MDP is unknown. In the following section, we want to relax this assumption by replacing the exact gradient by a stochastic approximation. To be more precise, we view a model-free setting where we are only able to generate trajectories of the finite-time MDP. These trajectories are used to formulate the stochastic PG method for training the parameters in both the simultaneous and dynamic approach.

Although in both approaches we are able to guarantee almost sure asymptotic convergence similar to the exact PG scheme, we are no longer able to control the constants $c$ and $c_h$ respectively along trajectories of the stochastic PG scheme due to the randomness in our iterations. Therefore, the derived lower bound in the weak PL-inequality may degenerate in general. In order to derive complexity bounds in the stochastic scenario, we make use of the crucial property that $c$ (and $c_h$ respectively) remain strictly positive along the trajectory of the exact PG scheme. To do so, we introduce the stopping times $\tau$ and $\tau_h$ stopping the scheme when the stochastic PG trajectory is too far away from the exact PG trajectory (under same initialisation). Hence, conditioning on $\{\tau \geq n\}$ (and $\{\tau_h \geq n\}$ respectively) forces the stochastic PG to remain close to the exact PG scheme and hence, guarantees non-degenerated weak PL-inequalities. The proof structure in the stochastic setting is then two-fold:

1. We derive a rate of convergence of the stochastic PG scheme under non-degenerated weak PL-inequality on the event $\{\tau \geq n\}$. Since we consider a constant step size, the batch size needs to be increased sufficiently fast for controlling the variance occurring through the stochastic approximation scheme. See Lemma D.4 and Lemma D.8.
2. We introduce a second rule for increasing the batch-size depending on a tolerance $\delta > 0$ leading to $\mathbb{P}(\tau \leq n) < \delta$. This means, that one forces the stochastic PG to remain close to the exact PG with high probability. See Lemma D.5 and Lemma D.9.

A similar proof strategy has been introduced in Ding et al. (2022) for proving convergence for entropy-regularised stochastic PG in the discounted case. Their analysis heavily depends on the existence of an optimal parameter which is due to regularisation. In the unregularised problem this is not the case since the softmax parameters usually diverge to $+/-\infty$ in order to approximate a deterministic optimal solution. Consequently, their analysis does not carry over straightforwardly to the unregularised setting. One of the main challenges in our proof is to construct a different stopping time, independent of optimal parameters, such that the stopping time still occurs with small probability given a large enough batch size. We again first discuss the simultaneous approach followed by the dynamic approach.

**Simultaneous stochastic policy gradient estimator:** Consider $K$ trajectories $(s_h^i, a_h^i)_{h=0}^{H-1}$, for $i = 1, \ldots, K$, generated by $s_0^i \sim \mu$, $a_h^i \sim \pi^\theta(\cdot|s_h^i)$ and $s_h^i \sim p(\cdot|s_{h-1}^i, a_{h-1}^i)$ for $0 < h < H$. The gradient estimator is defined by

$$\widehat{\nabla} J^K(\theta, \mu) = \frac{1}{K} \sum_{i=1}^{K} \sum_{h=0}^{H-1} \nabla \log(\pi^\theta(a_h^i|s_h^i)) \hat{R}_h^i, \tag{7}$$

where $\hat{R}_h^i = \sum_{k=h}^{H-1} r(s_k^i, a_k^i)$ is an unbiased estimator of the $h$-state-action value function in $(s_h^i, a_h^i)$ under policy $\pi^\theta$. This gradient estimator is unbiased and has bounded variance (Lemma D.1). Then the stochastic PG updates for training the softmax parameter are given by

$$\bar{\theta}^{(n+1)} = \bar{\theta}^{(n)} + \eta \widehat{\nabla} J^K(\bar{\theta}^{(n)}, \mu). \tag{8}$$

Our main result for the simultaneous stochastic PG scheme is given as follows.

**Theorem 4.1.** *Under Assumption 3.1, let $\mu$ be a probability measure such that $\mu(s) > 0$ for all $s \in \mathcal{S}$. Consider the final policy using Algorithm 1 with stochastic updates from Equation 8 denoted by $\hat{\pi}^* = \pi^{\bar{\theta}^{(N)}}$. Moreover, for any $\delta, \epsilon > 0$ assume that the number of training steps satisfies $N \geq \left(\frac{21|\mathcal{S}|H^5 R^*}{\epsilon \delta c^2}\right)^2 \left\|\frac{d_\mu^{\pi^*}}{\mu}\right\|_\infty^4$, let $\eta = \frac{1}{5H^2 R^* \sqrt{N}}$ and $K \geq \frac{10\max\{R^*, 1\}^2 N^3}{c^2 \delta^2}$. Then it holds true that*

$$\mathbb{P}\left(V_0^*(\mu) - V_0^{\hat{\pi}^*}(\mu) < \epsilon\right) > 1 - \delta.$$

**Dynamic stochastic policy gradient estimator:** For fixed $h$ consider $K_h$ trajectories $(s_k^i, a_k^i)_{k=h}^{H-1}$, for $i = 1, \ldots, K_h$, generated by $s_h^i \sim \mu_h$, $a_h^i \sim \pi^\theta$ and $a_k^i \sim \tilde{\pi}_k$ for $h < k < H$. The estimator is defined by

$$\widehat{\nabla} J_h^K(\theta, \tilde{\pi}_{(h+1)}, \mu_h) = \frac{1}{K_h} \sum_{i=1}^{K_h} \nabla \log(\pi^\theta(a_h^i | s_h^i)) \hat{R}_h^i, \tag{9}$$

where $\hat{R}_h^i = \sum_{k=h}^{H-1} r(s_k^i, a_k^i)$ is an unbiased estimator of the $h$-state-action value function in $(s_h^i, a_h^i)$ under policy $\tilde{\pi}$. Then the stochastic PG updates for training the parameter $\theta_h$ are given by

$$\bar{\theta}_h^{(n+1)} = \bar{\theta}_h^{(n)} + \eta_h \widehat{\nabla} J_h^{K_h}(\bar{\theta}_h^{(n)}, \tilde{\pi}_{(h+1)}, \mu_h). \tag{10}$$

Our main result for the dynamic stochastic PG scheme is given as follows.

**Theorem 4.2.** *For all $h \in \mathcal{H}$, let $\mu_h$ be probability measures such that $\mu_h(s) > 0$ for all $h \in \mathcal{H}$, $s \in \mathcal{S}_h$. Consider the final policy using Algorithm 2 with stochastic updates from Equation 10 denoted by $\hat{\pi}^* = (\pi^{\bar{\theta}_0^{(N_0)}}, \ldots, \pi^{\bar{\theta}_{H-1}^{(N_{H-1})}})$. Moreover, for any $\delta, \epsilon > 0$ assume that the numbers of training steps satisfy $N_h \geq \left( \frac{12(H-h)R^* H^2 \left\| \frac{1}{\mu_h} \right\|_\infty}{\delta c_h^2 \epsilon} \right)^2$, let $\eta_h = \frac{1}{2(H-h)R^*\sqrt{N_h}}$ and $K_h \geq \frac{5N_h^3 H^2}{c_h^2 \delta^2}$. Then it holds true that*

$$\mathbb{P}\left( \forall s \in \mathcal{S}_0 : V_0^*(s) - V_0^{\hat{\pi}^*}(s) < \epsilon \right) > 1 - \delta.$$

**Comparison** In both scenarios the derived complexity bounds for the stochastic PG uses a very large batch size and small step size. It should be noted that the choice of step size and batch size are closely connected and both strongly depend on the number of training steps $N$. Specifically, as $N$ increases, the batch size increases, while the step size tends to decrease to prevent exceeding the stopping time with high probability. However, it is possible to increase the batch size even further and simultaneously benefit from choosing a larger step size, or vice versa.

An advantage of the dynamic approach is that $c_h$ can be explicitly known for uniform initialisation. Hence, the complexity bounds for the dynamic approach results in a practicable algorithm, while $c$ is unknown and possibly arbitrarily small for the simultaneous approach. Finally, we will also compare the complexity with respect to the time horizon. For the simultaneous approach the number of training steps scales with $H^{10}$, and the batch size with $H^{30}$, while in the dynamic approach the overall number of training steps scale with $H^7$ and the batch size with $H^{20}$. We are aware that these bounds are far from tight and irrelevant for practical implementations. Nevertheless, these bounds highlight once more the advantage of the dynamic approach in comparison to the simultaneous approach and show (the non-trivial fact) that the algorithms can be made to converge without knowledge of exact gradients and without regularisation.

## 5  CONCLUSION AND FUTURE WORK

In this paper, we have presented a convergence analysis of two PG methods for undiscounted MDPs with finite-time horizon in the tabular parametrisation. Assuming exact gradients we have obtained an $\mathcal{O}(1/n)$-convergence rate for both approaches where the behavior regarding the time horizon and the model-dependent constant $c$ is better in the dynamic approach than in the simultaneous approach. In the model-free setting we have derived complexity bounds to approximate the error to global optima with high probability using stochastic PG. It would be desirable to derive tighter bounds using for example adaptive step sizes or variance reduction methods that lead to more realistic batch sizes.

Similar to many recent results, the presented analysis relies on the tabular parametrisation. However, the heuristic from the policy gradient theorem does not, and the dynamic programming perspective suggests that parameters should be trained backwards in time. It would be interesting future work to see how this theoretical insight can be implemented in lower dimensional parametrisations using for instance neural networks.

ACKNOWLEDGMENTS

Special thanks to the anonymous reviewers for their constructive feedback and insightful discussions, which greatly improved this paper. We also acknowledge the valuable input received from the anonymous reviewers of previous submissions. Sara Klein thankfully acknowledges the funding support by the Hanns-Seidel-Stiftung e.V. and is grateful to the DFG RTG1953 "Statistical Modeling of Complex Systems and Processes" for funding this research.

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

## A  PRELIMINARY RESULTS

Before we prove some preliminary results we will give a more detailed description of the enlarged state space $\mathcal{S}^{[\mathcal{H}]}$ and introduce more functions and notation used throughout the proofs.

*Remark* A.1. The enlarged state space introduced for the simultaneous approach encompasses all possible states across all epochs. Therefore initially, states are associated with their respective epochs, resulting in disjoint state spaces between epochs, which are subsequently fused into a single comprehensive state space $\mathcal{S}^{[\mathcal{H}]}$. Formally, this means that for every state space $\mathcal{S}_h = \{s^1, \ldots, s^{L_h}\}$ one constructs disjoint sets $\mathcal{D}_h = \mathcal{S}_h \times \{h\} = \{s_h^1, \ldots, s_h^{L_h}\}$ for $h = 0, \ldots, H-1$. Then, $\mathcal{S}^{[\mathcal{H}]} := \mathcal{D}_0 \uplus \cdots \uplus \mathcal{D}_{H-1}$ contains all possible states associated with their epoch.

The $h$-state-action value function for every tuple $(s,a) \in \mathcal{S}_h \times \mathcal{A}_s$ is defined by

$$Q_h^{\pi_{(h+1)}}(s,a) := r(s,a) + \sum_{s' \in \mathcal{S}_{h+1}} p(s'|s,a) V_{h+1}^{\pi_{(h+1)}}(s'), \quad h \le H-2, \tag{11}$$

where $V_h^\pi(s) = V_h^\pi(\delta_s)$ the $h$-state value function with start state $s \in \mathcal{S}_h$. Note that $Q_h$ is independent of policy $\pi_h$ and for $H-1$, $Q_{H-1}(s,a) := r(s,a)$ independently of any policy. Furthermore, define the $h$-state-action advantage function

$$A_h^{\pi_{(h)}}(s,a) := Q_h^{\pi_{(h+1)}}(s,a) - V_h^{\pi_{(h)}}(s), \quad s \in \mathcal{S}_h, a \in \mathcal{A}_s. \tag{12}$$

In the following, we will suppress the dependence of $\pi_{(h)}$ and write $\pi$ in the superscripts of $V_h$, $Q_h$ and $A_h$, when the policy is clear out of context.

*Remark* A.2. Note that we can drop the subscript $h$ in the value function, state-action value function or advantage function, when we define them on the enlarged state-space $\mathcal{S}^{[\mathcal{H}]}$. Then, $V$ is a vector of dimension $|\mathcal{S}^{[\mathcal{H}]}|$ and $Q$ and $A$ are matrices of dimension $|\mathcal{S}^{[\mathcal{H}]}| \times |\mathcal{A}|$. Hence, using the state $s_h \in \mathcal{S}^{[\mathcal{H}]}$ assigned to the epoch $h$, we use the notation $V^{\pi^\theta}(s_h) := V_h^{\pi^\theta}(s_h)$ to denote the assigned value function in epoch $h$. Similar also for $Q$ and $A$.

Moreover we define the state visitation measure on the enlarged state space as

$$\tilde{\rho}_\mu^{\pi^\theta}(s_h) := \mathbb{P}_\mu^{\pi^\theta}(S_h = s_h), \tag{13}$$

for every $s_h \in \mathcal{S}^{[\mathcal{H}]}$ and the state visitation distribution as $\tilde{d}_\mu^{\pi^\theta} = \frac{1}{H}\tilde{\rho}_\mu^{\pi^\theta}$. Note that it holds $\sum_{s_h \in \mathcal{S}^{[\mathcal{H}]}} \tilde{\rho}_\mu^{\pi^\theta}(s_h) = \sum_{s \in \mathcal{S}} \rho_\mu^{\pi^\theta}(s) = H$.

The performance difference lemma (Kakade & Langford, 2002) is a useful identity to compare policies. It turns out to be very useful to prove convergence of PG methods (Agarwal et al., 2021). For finite-time MDPs we obtain the following version.

**Lemma A.3** (Performance difference lemma). *For any $h \in \mathcal{H}$ and for any pair of policies $\pi$ and $\pi'$ the following holds true for every $s \in \mathcal{S}_h$:*

$$V_h^\pi(s) - V_h^{\pi'}(s) = \sum_{k=h}^{H-1} \mathbb{E}_{S_h=s}^\pi \left[ A_k^{\pi'}(S_k, A_k) \right].$$

*Proof.*

$$V_h^\pi(s) - V_h^{\pi'}(s) = \mathbb{E}_{S_h=s}^{\pi_{(h)}} \left[ \sum_{k=h}^{H-1} r(S_k, A_k) \right] - V_h^{\pi'}(s)$$

$$= \mathbb{E}_{S_h=s}^{\pi_{(h)}} \left[ \sum_{k=h}^{H-1} r(S_k, A_k) + \sum_{k=h}^{H-1} V_k^{\pi'}(S_k) - \sum_{k=h}^{H-1} V_k^{\pi'}(S_k) \right] - V_h^{\pi'}(s)$$

$$= \mathbb{E}_{S_h=s}^{\pi_{(h)}} \left[ \sum_{k=h}^{H-1} r(S_k, A_k) + \sum_{k=h+1}^{H-1} V_k^{\pi'}(S_k) - \sum_{k=h}^{H-1} V_k^{\pi'}(S_k) \right]$$

$$= \mathbb{E}_{S_h=s}^{\pi_{(h)}} \left[ \sum_{k=h}^{H-1} r(S_k, A_k) + \sum_{k=h}^{H-2} V_{k+1}^{\pi'}(S_{k+1}) - \sum_{k=h}^{H-1} V_k^{\pi'}(S_k) \right]$$

$$= \mathbb{E}_{S_h=s}^{\pi^{(h)}} \Big[ \sum_{k=h}^{H-1} \big( r(S_k, A_k) + V_{k+1}^{\pi'}(S_{k+1}) - V_k^{\pi'}(S_k) \big) \Big]$$

$$= \mathbb{E}_{S_h=s}^{\pi^{(h)}} \Big[ \sum_{k=h}^{H-1} A_k^{\pi'}(S_k, A_k) \Big]$$

$$= \sum_{k=h}^{H-1} \mathbb{E}_{S_h=s}^{\pi^{(h)}} \Big[ A_k^{\pi'}(S_k, A_k) \Big],$$

where we have used that $r(S_k, A_k) + V_{k+1}^{\pi'}(S_{k+1}) = Q_k^{\pi'}(S_k, A_k)$. In the fifth equation we used the notation $V_H \equiv 0$ and note that $Q_{H-1} \equiv r$ independent of any policy. $\qquad \square$

This implies a corollary for the two objectives $J(\theta, \mu)$ and $J_h(\theta_h, \tilde{\pi}_{(h+1)}, \mu_h)$.

**Corollary A.4.** *For the objective $J(\theta, \mu)$ defined in Equation 3 and $J_h(\theta_h, \tilde{\pi}_{(h+1)}, \mu_h)$ defined in Equation 5 it holds*

$$J^*(\mu) - J(\theta, \mu) = \mathbb{E}_{\mu}^{\pi^*} \Big[ \sum_{h=0}^{H-1} A_h^{\pi^\theta}(S_h, A_h) \Big] = \sum_{s_h \in \mathcal{S}^{[\mathcal{H}]}} \tilde{\rho}_\mu^{\pi^*}(s_h) A_h^{\pi^\theta}(s_h, a^*(s_h))$$

*and*

$$J_h^*(\tilde{\pi}_{(h+1)}, \mu) - J_h(\theta_h, \tilde{\pi}_{(h+1)}, \mu_h) = \mathbb{E}_{\mu}^{\pi^*} \Big[ A_h^{(\pi^\theta, \tilde{\pi}_{(h+1)})}(S_h, A_h) \Big].$$

*Proof.* The first claim follows directly from Lemma A.3 and the definition of the state visitation measure in Equation 13.

For the second claim, we proof a more general result: For any $h \in \mathcal{H}$ and two policies $\pi$ and $\pi'$: If $\pi_{(h+1)} = \pi'_{(h+1)}$, it holds that

$$V_h^\pi(s) - V_h^{\pi'}(s) = \mathbb{E}_{S_h=s}^{\pi^{(h)}} \Big[ A_h^{\pi'}(S_h, A_h) \Big].$$

To see this, let $k > h$, then

$$\mathbb{E}_{S_h=s}^{\pi^{(h)}} \Big[ A_k^{\pi'}(S_k, A_k) \Big]$$

$$= \sum_{a \in \mathcal{A}} \pi_h(a|s) \sum_{s' \in \mathcal{S}} p(s'|s,a) \mathbb{E}_{S_{h+1}=s'}^{\pi^{(h+1)}} \Big[ Q_k^{\pi'}(S_k, A_k) - V_k^{\pi'}(S_k) \Big]$$

$$= \sum_{a \in \mathcal{A}} \pi_h(a|s) \sum_{s' \in \mathcal{S}} p(s'|s,a) \mathbb{E}_{S_{h+1}=s'}^{\pi'_{(h+1)}} \Big[ Q_k^{\pi'}(S_k, A_k) - V_k^{\pi'}(S_k) \Big]$$

$$= \sum_{a \in \mathcal{A}} \pi_h(a|s) \sum_{s' \in \mathcal{S}} p(s'|s,a) \Big( \mathbb{E}_{S_{h+1}=s'}^{\pi'_{(h+1)}} \Big[ \mathbb{E}_{S_k}^{\pi'} [Q_k^{\pi'}(S_k, A_k)] \Big] - \mathbb{E}_{S_{h+1}=s'}^{\pi'_{(h+1)}} \Big[ V_k^{\pi'}(S_k) \Big] \Big)$$

$$= \sum_{a \in \mathcal{A}} \pi_h(a|s) \sum_{s' \in \mathcal{S}} p(s'|s,a) \Big( \mathbb{E}_{S_{h+1}=s'}^{\pi'_{(h+1)}} \Big[ V_k^{\pi'}(S_k) \Big] - \mathbb{E}_{S_{h+1}=s'}^{\pi'_{(h+1)}} \Big[ V_k^{\pi'}(S_k) \Big] \Big)$$

$$= 0.$$

The claim follows with Lemma A.3. $\qquad \square$

Next we derive the policy gradient theorems for finite-time horizon MDPs in both, the simultaneous and the dynamic approach.

**Theorem A.5** (Policy Gradient Theorem for the simultaneous approach). *Consider any parametrisation $\pi^\theta$ on the enlarged state space $\mathcal{S}^{[\mathcal{H}]}$, then the gradient of the $J(\theta, \mu)$ defined in Equation 3 is given by*

$$\nabla J(\theta, \mu) = \mathbb{E}_{\mu}^{\pi^\theta} \Big[ \sum_{h=0}^{H} \nabla \log(\pi^\theta(A_h|S_h)) Q_h^{\pi^\theta}(S_h, A_h) \Big]$$

$$= \sum_{s_h \in \mathcal{S}^{[\mathcal{H}]}} \tilde{\rho}_\mu^{\pi^\theta}(s_h) \sum_{a \in \mathcal{A}_{s_h}} \pi^\theta(a|s_h) \nabla \log(\pi^\theta(a|s_h)) Q_h^{\pi^\theta}(s_h, a_h).$$

*Proof.* The second equality follows directly from the definition of the state visitation measure in Equation 13.

For the first equality consider the probability of a trajectory $\tau = (s_0, a_0, \ldots, s_{H-1}, a_{H-1})$ under the policy $\pi^\theta$ and initial state distribution $\mu$, i.e.

$$p_\mu^{\pi^\theta}(\tau) = \mu(s_h)\pi^\theta(a_0|s_0) \prod_{k=1}^{H-1} p(s_k|s_{k-1}, a_{k-1})\pi^\theta(a_k|s_k).$$

Then,

$$\nabla \log(p_\mu^{\pi^\theta}(\tau)) = \nabla \Big( \log(\mu(s_h)) + \log(\pi^\theta(a_0|s_0))$$
$$+ \sum_{k=1}^{H-1} \log(p(s_k|s_{k-1}, a_{k-1})) + \log(\pi^\theta(a_k|s_k)) \Big)$$
$$= \nabla \sum_{k=0}^{H-1} \log(\pi^\theta(a_k|s_k)),$$

which is known as the log-trick. Let $\mathcal{W}$ be the set of all trajectories from $0$ to $H-1$. Note that $\mathcal{W}$ is finite due to the assumption that state and action space is finite. Then,

$$\nabla J(\theta, \mu) = \nabla \sum_{\tau \in \mathcal{W}} p_\mu^{\pi^\theta}(\tau) \sum_{k=0}^{H-1} r(s_k, a_k)$$
$$= \sum_{\tau \in \mathcal{W}} p_\mu^{\pi^\theta}(\tau) \nabla \log(p_\mu^{\pi^\theta}(\tau)) \sum_{k=0}^{H-1} r(s_k, a_k)$$
$$= \sum_{\tau \in \mathcal{W}} p_\mu^{\pi^\theta}(\tau) \sum_{h=0}^{H-1} \nabla \log(\pi^\theta(a_h|s_h)) \sum_{k=0}^{H-1} r(s_k, a_k)$$
$$= \sum_{\tau \in \mathcal{W}} p_\mu^{\pi^\theta}(\tau) \sum_{h=0}^{H-1} \nabla \log(\pi^\theta(a_h|s_h)) \sum_{k=h}^{H-1} r(s_k, a_k)$$
$$= \mathbb{E}_\mu^{\pi^\theta} \Big[ \sum_{h=0}^{H-1} \nabla \log(\pi^\theta(A_h|S_h)) \sum_{k=h}^{H-1} r(S_k, A_k) \Big]$$
$$= \mathbb{E}_\mu^{\pi^\theta} \Big[ \sum_{h=0}^{H-1} \nabla \log(\pi^\theta(A_h|S_h)) \mathbb{E}_{S_h}^{\pi^\theta} \Big[ \sum_{k=h}^{H-1} r(S_k, A_k) \big| S_h, A_h \Big] \Big]$$
$$= \mathbb{E}_\mu^{\pi^\theta} \Big[ \sum_{h=0}^{H-1} \nabla \log(\pi^\theta(A_h|S_h)) \, Q_h^{\pi^\theta}(S_h, A_h) \Big].$$

In the forth equation we have used that for every $k < h$ it holds

$$\mathbb{E}_\mu^{\pi^\theta} \Big[ \nabla \log(\pi^\theta(A_h|S_h))r(S_k, A_k) \Big] = \mathbb{E}_\mu^{\pi^\theta} \Big[ \mathbb{E}_\mu^{\pi^\theta} \Big[ \nabla \log(\pi^\theta(A_h|S_h)) \big| S_0, A_0, \ldots S_{h-1}, A_{h-1}, S_h \Big] r(S_k, A_k) \Big]$$

and furthermore

$$\mathbb{E}_\mu^{\pi^\theta} \Big[ \nabla \log(\pi^\theta(A_h|S_h)) \big| S_0, A_0, \ldots S_{h-1}, A_{h-1}, S_h \Big]$$
$$= \mathbb{E}_\mu^{\pi^\theta} \Big[ \nabla \log(\pi^\theta(A_h|S_h)) \big| S_h \Big]$$
$$= \sum_{a \in \mathcal{A}_{S_h}} \pi^\theta(a|S_h) \nabla \log(\pi^\theta(A_h|S_h))$$
$$= \nabla \Big( \sum_{a \in \mathcal{A}_{S_h}} \pi^\theta(a|S_h) \Big) = 0.$$

$\square$

**Theorem A.6** (Policy Gradient Theorem for the dynamic approach). *For a fixed policy $\tilde{\pi}$ and $h \in \mathcal{H}$ the gradient of $J_h(\theta_h, \tilde{\pi}_{(h+1)}, \delta_s)$ defined in Equation 5 is given by*

$$\nabla J_h(\theta_h, \tilde{\pi}_{(h+1)}, \delta_s) = \mathbb{E}_{S_h=s, A_h \sim \pi^{\theta_h}(\cdot|s)}[\nabla \log(\pi^\theta(A_h|S_h)) Q_h^{\tilde{\pi}}(S_h, A_h)].$$

*Proof.* The probability of a trajectory $\tau = (s_h, a_h, \ldots, s_{H-1}, a_{H-1})$ under the policy $(\pi^\theta, \tilde{\pi}_{(h+1)}) = (\pi^\theta, \tilde{\pi}_{h+1}, \ldots, \tilde{\pi}_{H-1})$ and initial state distribution $\delta_s$ is given by

$$p_s^{(\pi^\theta, \tilde{\pi}_{(h+1)})}(\tau) = \delta_s(s_h)\pi^\theta(a_h|s_h) \prod_{k=h+1}^{H-1} p(s_k|s_{k-1}, a_{k-1})\tilde{\pi}_k(a_k|s_k).$$

Then,

$$\begin{aligned}
\nabla \log(p_s^{(\pi^\theta, \tilde{\pi}_{(h+1)})}(\tau)) &= \nabla\Big( \log(\delta_s(s_h)) + \log(\pi^\theta(a_h|s_h)) \\
&\quad + \sum_{k=h+1}^{H-1} \log(p(s_k|s_{k-1}, a_{k-1})) + \log(\tilde{\pi}_k(a_k|s_k)) \Big) \\
&= \nabla \log(\pi^\theta(a_h|s_h)),
\end{aligned}$$

which is known as the log-trick. Let $\mathcal{W}$ be the set of all trajectories from $h$ to $H-1$. Note that $\mathcal{W}$ is finite due to the assumption that state and action space is finite. Then for $s \in \mathcal{S}_h$

$$\begin{aligned}
\nabla J_h(\theta_h, \tilde{\pi}_{(h+1)}, \delta_s) &= \nabla \sum_{\tau \in \mathcal{W}} p_s^{(\pi^\theta, \tilde{\pi}_{(h+1)})}(\tau) \sum_{k=h}^{H-1} r(s_k, a_k) \\
&= \sum_{\tau \in \mathcal{W}} p_s^{(\pi^\theta, \tilde{\pi}_{(h+1)})}(\tau)\nabla \log(p_s^{(\pi^\theta, \tilde{\pi}_{(h+1)})}) \sum_{k=h}^{H-1} r(s_k, a_k) \\
&= \sum_{\tau \in \mathcal{W}} p_s^{(\pi^\theta, \tilde{\pi}_{(h+1)})}(\tau)\nabla \log(\pi^\theta(a_h|s_h)) \sum_{k=h}^{H-1} r(s_k, a_k) \\
&= \mathbb{E}_{S_h=s}^{(\pi^\theta, \tilde{\pi}_{(h+1)})}\Big[\nabla \log(\pi^\theta(A_h|S_h)) \sum_{k=h}^{H-1} r(S_k, A_k)\Big] \\
&= \mathbb{E}_{S_h=s}^{(\pi^\theta, \tilde{\pi}_{(h+1)})}\Big[\nabla \log(\pi^\theta(A_h|S_h))\mathbb{E}_{\mathcal{S}_h}^{\tilde{\pi}}\Big[ \sum_{k=h}^{H-1} r(S_k, A_k)\big|S_h, A_h\Big]\Big] \\
&= \mathbb{E}_{S_h=s, A_h \sim \pi^\theta(\cdot|s)}\Big[\nabla \log(\pi^\theta(A_h|S_h)) Q_h^{\tilde{\pi}}(S_h, A_h)\Big].
\end{aligned}$$

$\square$

Using these two theorems we can explicitly derive the derivatives of our objective functions under the softmax parametrisation. First, we compute the derivative of the softmax policy for every $s \in \mathcal{S}_h$ and $a \in \mathcal{A}_s$,

$$\pi^\theta(a|s) = \frac{e^{\theta(s,a)}}{\sum_{a' \in \mathcal{A}} e^{\theta(s,a')}},$$

with parameter $\theta \in \mathbb{R}^{d_h}$:

$$\frac{\partial \log(\pi^\theta(a|s))}{\partial \theta(a', s')} = \mathbf{1}_{\{s=s'\}}(\mathbf{1}_{\{a=a'\}} - \pi^\theta(a'|s')).$$

Hence,

$$\nabla \log(\pi^\theta(a|s)) = \Big(\mathbf{1}_{\{s=s'\}}(\mathbf{1}_{\{a=a'\}} - \pi^\theta(a'|s'))\Big)_{s' \in \mathcal{S}_h, a' \in \mathcal{A}_{s'}} \quad \in \mathbb{R}^{d_h}.$$

**Lemma A.7.** *The partial derivative of the objective defined in Equation 3 is given by:*

$$\frac{\partial J(\theta, \mu)}{\partial \theta(s_h, a)} = \tilde{\rho}_\mu^{\pi^\theta}(s_h) \pi^\theta(a|s_h) A_h^{\pi^\theta}(s_h, a),$$

*for every $s_h \in \mathcal{S}^{[\mathcal{H}]}$ and $a \in \mathcal{A}_{s_h}$.*

*Proof.* Let $s_h \in \mathcal{S}^{[\mathcal{H}]}$ and $a \in \mathcal{A}_{s_h}$. Using Theorem A.5, it holds that

$$\frac{\partial J(\theta, \mu)}{\partial \theta(s_h, a)} = \mathbb{E}_\mu^{\pi^\theta} \Big[ \sum_{h=0}^{H-1} \frac{\partial}{\partial \theta(s_h, a)} \log(\pi^\theta(A_h|S_h)) Q_h^{\pi^\theta}(S_h, A_h) \Big]$$

$$= \mathbb{E}_\mu^{\pi^\theta} \Big[ \sum_{h=0}^{H-1} \mathbf{1}_{\{S_h = s_h\}} \big( \mathbf{1}_{\{A_h = a\}} - \pi^\theta(a|s_h) \big) Q_h^{\pi^\theta}(S_h, A_h) \Big]$$

$$= \mathbb{P}_\mu^{\pi^\theta}(S_h = s_h) \sum_{a'} \pi^\theta(a'|s_h) \big( \mathbf{1}_{\{a' = a\}} - \pi^\theta(a|s_h) \big) Q_h^{\pi^\theta}(s_h, a')$$

$$= \tilde{\rho}_\mu^{\pi^\theta}(s_h) \Big( \pi^\theta(a|s_h) Q^{\pi^\theta}(s_h, a) - \sum_{a'} \pi^\theta(a'|s_h) \pi^\theta(a|s_h) Q_h^{\pi^\theta}(s_h, a') \Big)$$

$$= \tilde{\rho}_\mu^{\pi^\theta}(s_h) \pi^\theta(a|s_h) A_h^{\pi^\theta}(s_h, a).$$

$\square$

**Lemma A.8.** *For fix $h \in \mathcal{H}$, the partial derivative of the objective defined in Equation 5 is given by:*

$$\frac{\partial J_h(\theta, \tilde{\pi}_{(h+1)}, \mu_h)}{\partial \theta(s, a)} = \mu_h(s) \pi^\theta(a|s) A_h^{(\pi^\theta, \tilde{\pi}_{(h+1)})}(s, a),$$

*for every $s \in \mathcal{S}_h$ and $a \in \mathcal{A}_s$*

*Proof.* By the policy gradient Theorem A.6,

$$\nabla J_h(\theta, \tilde{\pi}_{(h+1)}, \mu_h) = \nabla \mathbb{E}_{s \sim \mu_h} [J_h(\theta, \tilde{\pi}_{(h+1)}, \delta_s)]$$

$$= \sum_{s \in \mathcal{S}} \mu_h(s) \nabla J_h(\theta, \tilde{\pi}_{(h+1)}, \delta_s)$$

$$= \sum_{s \in \mathcal{S}} \mu_h(s) \mathbb{E}_{S_h = s, A_h \sim \pi^\theta(\cdot|s)} [\nabla \log(\pi^\theta(A_h|S_h)) Q_h^{\tilde{\pi}}(S_h, A_h)].$$

Next we plug in the derivative of the softmax parametrisation and obtain

$$\nabla J_h(\theta, \tilde{\pi}_{(h+1)}, \mu_h)$$

$$= \sum_{s \in \mathcal{S}} \mu_h(s) \mathbb{E}_{S_h = s, A_h \sim \pi^\theta(\cdot|s)} \Big[ \Big( \mathbf{1}_{\{S_h = s'\}} (\mathbf{1}_{\{A_h = a'\}} - \pi^\theta(a'|s')) \Big)_{s' \in \mathcal{S}_h, a' \in \mathcal{A}_{s'}} Q_h^{\tilde{\pi}}(S_h, A_h) \Big]$$

$$= \Big( \sum_{s \in \mathcal{S}} \mu_h(s) \sum_{a \in \mathcal{A}_s} \pi^\theta(a|s) \mathbf{1}_{\{s = s'\}} (\mathbf{1}_{\{a = a'\}} - \pi^\theta(a'|s')) Q_h^{\tilde{\pi}}(s, a) \Big)_{s' \in \mathcal{S}_h, a' \in \mathcal{A}_{s'}}$$

$$= \Big( \mu_h(s') \pi^\theta(a'|s') Q_h^{\tilde{\pi}}(s', a') - \mu_h(s') \pi^\theta(a'|s') \sum_{a \in \mathcal{A}_s} \pi^\theta(a|s') Q_h^{\tilde{\pi}}(s', a) \Big)_{s' \in \mathcal{S}_h, a' \in \mathcal{A}_{s'}}$$

$$= \Big( \mu_h(s') \pi^\theta(a'|s') (Q_h^{\tilde{\pi}}(s', a') - V_h^{(\pi^\theta, \tilde{\pi}_{(h+1)})}(s')) \Big)_{s' \in \mathcal{S}_h, a' \in \mathcal{A}_{s'}}$$

$$= \Big( \mu_h(s') \pi^\theta(a'|s') A_h^{(\pi^\theta, \tilde{\pi}_{(h+1)})}(s', a') \Big)_{s' \in \mathcal{S}_h, a' \in \mathcal{A}_{s'}},$$

where we used that $\sum_{a \in \mathcal{A}_s} \pi^\theta(a|s') Q_h^{\tilde{\pi}}(s', a) = J_h(\theta, \tilde{\pi}_{(h+1)}, \delta_{s'}) = V_h^{(\pi^\theta, \tilde{\pi}_{(h+1)})}(s')$. $\square$

# B  PROOFS OF SECTION 3

## B.1  PROOFS OF SECTION 3.1

**Lemma B.1.** *The objective $J(\theta, \mu)$ from Equation 3 is smooth in $\theta$ with parameter $\beta = H^2 R^* (2 - \frac{1}{|\mathcal{A}|})$.*

Comparing this result to Lemma E.1. in Yuan et al. (2022) where the smoothness constant of a discounted MDP under softmax parametrisation is given by $\frac{R^*}{(1-\gamma)^2} \left( 2 - \frac{1}{|\mathcal{A}|} \right)$, we can see that $\frac{1}{1-\gamma}$, the expectation of a geometric r.v. and the expected length of a discounted MDP, is replaced by $H$, the expected length of the finite-time MDP.

*Proof.* We are going to bound the norm of the hessian. Therefore, we first calculate the first and second derivative if $J$ for finite-time horizon stationary MDPs. So, let $\tau = (s_0, a_0, s_1, \ldots, s_{H-1}, a_{H-1})$ be a trajectory of the MDP under policy $\pi^\theta$ and denote by $p_\mu^\theta$ the discrete probability density. Then,

$$
\nabla J(\theta, \mu) = \nabla \Big( \sum_\tau p_\mu^\theta(\tau) \sum_{h=0}^{H-1} r(s_h, a_h) \Big)
$$

$$
= \sum_\tau p_\mu^\theta(\tau) \Big( \sum_{h=0}^{H-1} \nabla \log(\pi^\theta(a_h|s_h)) \sum_{h=0}^{H-1} r(s_h, a_h) \Big)
$$

$$
= \mathbb{E}_\mu^{\pi^\theta} \Big[ \sum_{h=0}^{H-1} \nabla \log(\pi^\theta(a_h|s_h)) \sum_{h=0}^{H-1} r(s_h, a_h) \Big].
$$

For the second derivative we have

$$
\nabla^2 J(\theta, \mu) = \nabla \Big( \sum_\tau p_\mu^\theta(\tau) \Big( \sum_{h=0}^{H-1} \nabla \log(\pi^\theta(a_h|s_h)) \sum_{h=0}^{H-1} r(s_h, a_h) \Big) \Big)
$$

$$
= \underbrace{\sum_\tau p_\mu^\theta(\tau) \Big( \Big( \sum_{h=0}^{H-1} \nabla \log(\pi^\theta(a_h|s_h)) \Big) \Big( \sum_{h=0}^{H-1} \nabla \log(\pi^\theta(a_h|s_h)) \Big)^T \sum_{h=0}^{H-1} r(s_h, a_h) \Big)}_{(1)}
$$

$$
+ \underbrace{\sum_\tau p_\mu^\theta(\tau) \Big( \sum_{h=0}^{H-1} \nabla^2 \log(\pi^\theta(a_h|s_h)) \sum_{h=0}^{H-1} r(s_h, a_h) \Big)}_{(2)}.
$$

Using the bounded reward assumption we get for the second term, that

$$
||(2)|| \leq \mathbb{E}_\mu^{\pi^\theta} \Big[ \sum_{h=0}^{H-1} ||\nabla^2 \log(\pi^\theta(a_h|s_h))|| \Big] H R^*
$$

$$
= H R^* \sum_{h=0}^{H-1} \mathbb{E}_\mu^{\pi^\theta} \Big[ ||\nabla^2 \log(\pi^\theta(a_h|s_h))|| \Big].
$$

By Lemma 4.8 in Yuan et al. (2022), we have for the softmax parametrisation that $\mathbb{E}_\mu^{\pi^\theta} \Big[ ||\nabla^2 \log(\pi^\theta(a_h|s_h))|| \Big] \leq 1$. Hence,

$$
||(2)|| \leq H^2 R^*.
$$

Next for the first term,

$$\|(1)\| \leq \mathbb{E}_\mu^{\pi^\theta}\Big[\|\sum_{h=0}^{H-1} \nabla \log(\pi^\theta(a_h|s_h))\|^2\Big] H R^*$$

$$= H R^* \sum_{h=0}^{H-1} \mathbb{E}_\mu^{\pi^\theta}\Big[\|\nabla \log(\pi^\theta(a_h|s_h))\|^2\Big]$$

$$\leq H^2 R^* \big(1 - \frac{1}{|\mathcal{A}|}\big),$$

where we first used the bounded reward assumption, then Lemma 3.6 and again Lemma 4.8 from Yuan et al. (2022). Finally, we obtain that

$$\|\nabla^2 J(\theta, \mu)\| \leq H^2 R^* \big(2 - \frac{1}{|\mathcal{A}|}\big).$$

$\square$

**Lemma B.2.** *It holds that*

$$\|\nabla J(\theta, \mu)\|_2 \geq \frac{\min_{s_h \in \mathcal{S}^{[\mathcal{H}]}} \pi^\theta(a^*(s_h)|s_h)}{\sqrt{|\mathcal{S}^{[\mathcal{H}]}|}} \Big\|\frac{d_\mu^{\pi^*}}{d_\mu^{\pi^\theta}}\Big\|_\infty^{-1} (J^*(\mu) - J(\theta, \mu)).$$

*Proof.* The idea of the proof follows the outline of Mei et al. (2020, Lem. 8) from the discounted setting. It holds

$$\|\nabla J(\theta, \mu)\|_2 = \Big[\sum_{s_h \in \mathcal{S}^{[\mathcal{H}]}} \sum_a \Big(\frac{\partial V_0^{\pi^\theta}(\mu)}{\partial \theta(s_h, a)}\Big)^2\Big]^{1/2}$$

$$\geq \Big[\sum_{s_h \in \mathcal{S}^{[\mathcal{H}]}} \Big(\frac{\partial V_0^{\pi^\theta}(\mu)}{\partial \theta(s_h, a^*(s_h))}\Big)^2\Big]^{1/2}$$

$$\geq \frac{1}{\sqrt{|\mathcal{S}^{[\mathcal{H}]}|}} \sum_{s_h \in \mathcal{S}^{[\mathcal{H}]}} \Big|\frac{\partial V_0^{\pi^\theta}(\mu)}{\partial \theta(s_h, a^*(s_h))}\Big|$$

$$= \frac{1}{\sqrt{|\mathcal{S}^{[\mathcal{H}]}|}} \sum_{s_h \in \mathcal{S}^{[\mathcal{H}]}} \tilde{\rho}_\mu^{\pi^\theta}(s_h) \pi^\theta(a^*(s_h)|s_h) |A^{\pi^\theta}(s_h, a^*(s_h))|$$

$$\geq \frac{\min_{s_h \in \mathcal{S}^{[\mathcal{H}]}} \pi^\theta(a^*(s_h)|s_h)}{\sqrt{|\mathcal{S}^{[\mathcal{H}]}|}} \sum_{s_h \in \mathcal{S}^{[\mathcal{H}]}} \tilde{\rho}_\mu^{\pi^*}(s_h) \Big\|\frac{d_\mu^{\pi^*}}{d_\mu^{\pi^\theta}}\Big\|_\infty^{-1} A^{\pi^\theta}(s_h, a^*(s_h))$$

$$= \frac{\min_{s_h \in \mathcal{S}^{[\mathcal{H}]}} \pi^\theta(a^*(s_h)|s_h)}{\sqrt{|\mathcal{S}^{[\mathcal{H}]}|}} \Big\|\frac{d_\mu^{\pi^*}}{d_\mu^{\pi^\theta}}\Big\|_\infty^{-1} \underbrace{\sum_{s_h \in \mathcal{S}^{[\mathcal{H}]}} \rho_\mu^{\pi^*}(s_h) A^{\pi^\theta}(s_h, a^*(s_h))}_{=\mathbb{E}_\mu^{\pi^*}[\sum_{h=0}^{H-1} A_h^{\pi^\theta}(S_h, A_h)]}$$

$$= \frac{\min_{s_h \in \mathcal{S}^{[\mathcal{H}]}} \pi^\theta(a^*(s_h)|s_h)}{\sqrt{||\mathcal{S}^{[\mathcal{H}]}|}} \Big\|\frac{d_\mu^{\pi^*}}{d_\mu^{\pi^\theta}}\Big\|_\infty^{-1} (J^*(\mu) - J(\theta, \mu)).$$

The third line is due to Cauchy-Schwarz, afterwards we used the derivative of the objective function from Lemma A.7. For the firths line, not that $\Big\|\frac{\tilde{\rho}_\mu^{\pi^*}}{\tilde{\rho}_\mu^{\pi^\theta}}\Big\|_\infty = \Big\|\frac{d_\mu^{\pi^*}}{d_\mu^{\pi^\theta}}\Big\|_\infty$ by definition of the state visitation measures and the distribution mismatch coefficient (see Equation 6). Finally, the last equation is due to Corollary A.4 from the performance difference lemma. $\square$

*Remark* B.3. Note that in order to use this weak PL-inequality uniformly we also have to bound the distribution mismatch coefficient uniform in $\theta$. Therefore, under Assumption 3.1 it holds $d_\mu^{\pi^\theta}(s) \geq \frac{1}{H}\mu(s)$ by definition for any $\theta$, since

$$d_\mu^{\pi^\theta}(s) = \frac{1}{H} \sum_{h=0}^{H-1} \mathbb{P}_\mu^{\pi^\theta}(S_h = s) \geq \frac{1}{H}\mu(s). \tag{14}$$

Hence, we obtain that

$$\|\nabla J(\theta, \mu)\|_2 \geq \frac{\min_{s_h \in \mathcal{S}^{[\mathcal{H}]}} \pi^\theta(a^*(s_h)|s_h)}{H\sqrt{|\mathcal{S}|H}} \left\|\frac{d_\mu^{\pi^*}}{\mu}\right\|_\infty^{-1} (J^*(\mu) - J(\theta, \mu)).$$

*Remark* B.4. Without Assumption 3.1 the expression

$$\sum_{s \in \mathcal{S}} \sum_{h=0}^{H-1} \mathbb{P}_\mu^{\pi^\theta}(S_h = s) = \sum_{s_h \in \mathcal{S}_h} \mathbb{P}(S_h = s_h) \tag{15}$$

cannot be bounded from below by $\mu$, since the probability to visit states in later epochs depends crucially on $\theta$. This cannot be covered by $\mu$ as the state $s_h$ might not belong to $\mathcal{S}_0$.

**Lemma B.5.** *Let $\mu$ be a probability measure such that $\mu(s) > 0$ for all $s \in \mathcal{S}$ and let $0 < \eta \leq \frac{1}{5H^2R^*}$. Consider the sequence $(\theta^{(n)})$ generated by Algorithm 1 for arbitrary $\theta^{(0)} \in \mathcal{R}^{\sum_h d_h}$. Then, $c = c(\theta^{(0)}) = \inf_n \min_{s_h \in \mathcal{S}^{[\mathcal{H}]}} \pi^{\theta^{(n)}}(a^*(s_h)|s_h) > 0$.*

The proof is adapted to the finite-time horizon from Mei et al. (2020, Lem. 9).

*Proof.* We will drop the $\mu$ in $J(\theta, \mu)$ for the rest of the proof. Define for all $s_h \in \mathcal{S}^{[\mathcal{H}]}$,

$$\Delta^*(s_h) = Q^\infty(s_h, a_h^*(s)) - \max_{a \neq a^*(s_h)} Q^\infty(s_h, a) > 0, \quad \text{and} \quad \Delta^* = \min_{s_h \in \mathcal{S}^{[\mathcal{H}]}} \Delta^*(s_h) > 0,$$

where $Q^\infty$ is the optimal $Q$-function from Lemma C.2.

Now consider for any $s_h \in \mathcal{S}^{[\mathcal{H}]}$ the following sets

$$\mathcal{R}_1(s_h) = \left\{\theta : \frac{\partial J(\theta)}{\partial \theta(s_h, a^*(s_h))} \geq \frac{\partial J(\theta)}{\partial \theta(s_h, a)}, \text{ for all } a \neq a^*(s_h)\right\},$$

$$\mathcal{R}_2(s_h) = \left\{\theta : Q^{\pi^\theta}(s_h, a^*(s_h)) \geq Q^\infty(s_h, a^*(s_h)) - \frac{\Delta^*(s_h)}{2}\right\},$$

$$\mathcal{R}_3(s_h) = \left\{\theta^{(n)} : V^{\pi^{\theta^{(n)}}}(s_h) \geq Q^{\pi^{\theta^{(n)}}}(s_h, a^*(s_h)) - \frac{\Delta^*(s_h)}{2}, \text{ for all } n \geq 1 \text{ large enough}\right\}.$$

Furthermore, we define $c(s_h) = \frac{|\mathcal{A}|HR^*}{\Delta^*(s_h)} - 1$ and

$$N_c(s_h) = \left\{\theta : \pi^\theta(a_h^*(s_h)|s_h) \geq \frac{c(s_h)}{c(s_h)+1}\right\}.$$

We divide the proof into the following Claims:

Claim 1. $\mathcal{R}(s_h) = \mathcal{R}_1(s_h) \cap \mathcal{R}_2(s_h) \cap \mathcal{R}_3(s_h)$ is a *nice* region, i.e.

(i) $\theta^{(n)} \in \mathcal{R}(s_h) \Rightarrow \theta^{(n+1)} \in \mathcal{R}(s_h)$.
(ii) $\pi^{\theta^{(n+1)}}(a^*(s_h)|s_h) \geq \pi^{\theta^{(n)}}(a^*(s_h)|s_h)$.

Claim 2. $\mathcal{N}_c(s_h) \cap \mathcal{R}_2(s_h) \cap \mathcal{R}_3(s_h) \subseteq \mathcal{R}_1(s_h) \cap \mathcal{R}_2(s_h) \cap \mathcal{R}_3(s_h)$.

Claim 3. For every $s_h \in \mathcal{S}^{[\mathcal{H}]}$, there exists a finite-time $n_0(s_h) \geq 1$, such that

$$\theta^{(n_0(s_h))} \in \mathcal{N}_c(s_h) \cap \mathcal{R}_2(s_h) \cap \mathcal{R}_3(s_h) \subseteq \mathcal{R}_1(s_h) \cap \mathcal{R}_2(s_h) \cap \mathcal{R}_3(s_h)$$

and thus

$$\inf_{n \geq 1} \pi^{\theta^{(n)}}(a^*(s_h)|s_h) = \min_{1 \leq n \leq n_0(s_h)} \pi^{\theta^{(n)}}(a^*(s_h)|s_h)$$

.

If all three claims hold true, we can finally define $n_0 = \max_{s_h \in \mathcal{S}^{[\mathcal{H}]}} n_0(s_h)$, such that

$$\inf_{n \geq 1} \min_{s \in \mathcal{S}^{[\mathcal{H}]}} \pi^{\theta^{(n)}}(a^*(s_h)|s_h) = \min_{1 \leq n \leq n_0} \min_{s_h \in \mathcal{S}^{[\mathcal{H}]}} \pi^{\theta^{(n)}}(a^*(s_h)|s_h) > 0.$$

Due to the positiveness of the softmax parametrisation the assertion follows.

**Claim 1.** We first prove (i). Let $\theta^{(n)} \in \mathcal{R}(s_h)$ and $a \neq a^*(s_h)$. Then $\theta^{(n+1)} \in \mathcal{R}_3(s_h)$ by definition of $\mathcal{R}_3(s_h)$. To see that $\theta^{(n+1)} \in \mathcal{R}_2(s_h)$ note that

$$
\begin{aligned}
Q^{\pi^{\theta^{(n+1)}}}(s_h, a^*(s_h)) &= Q_h^{\pi^{\theta^{(n+1)}}}(s_h, a^*(s_h)) \\
&= Q_h^{\pi^{\theta^{(n)}}}(s_h, a^*(s_h)) + Q_h^{\pi^{\theta^{(n+1)}}}(s_h, a^*(s_h)) - Q_h^{\pi^{\theta^{(n)}}}(s_h, a^*(s_h)) \\
&= Q_h^{\pi^{\theta^{(n)}}}(s_h, a^*(s_h)) + r(s_h, a^*(s_h)) + \sum_{s' \mathcal{S}^{[\mathcal{H}]}} p(s'|s_h, a^*(s_h)) V_{h+1}^{\pi^{\theta^{(n+1)}}}(s') \\
&\quad - r(s_h, a^*(s_h)) - \sum_{s' \mathcal{S}^{[\mathcal{H}]}} p(s'|s_h, a^*(s_h)) V_{h+1}^{\pi^{\theta^{(n)}}}(s') \\
&= Q_h^{\pi^{\theta^{(n)}}}(s_h, a^*(s_h)) + \sum_{s' \mathcal{S}^{[\mathcal{H}]}} p(s'|s_h, a^*(s_h)) \left( V_{h+1}^{\pi^{\theta^{(n+1)}}}(s') - V_{h+1}^{\pi^{\theta^{(n)}}}(s') \right) \\
&\geq Q_h^{\pi^{\theta^{(n)}}}(s_h, a^*(s_h)) = Q^{\pi^{\theta^{(n)}}}(s_h, a^*(s_h)) \\
&\geq Q^\infty(s_h, a^*(s_h)) - \frac{\Delta^*(s_h)}{2},
\end{aligned}
$$

where the first inequality is due to monotonicity of $V^{\pi^{\theta^{(n+1)}}}(s')$ in $n$ for every $s' \in \mathcal{S}^{[\mathcal{H}]}$ and the last inequality follows from $\theta^{(n)} \in \mathcal{R}_2(s_h)$.

Next we show $\theta^{(n+1)} \in \mathcal{R}_1(s_h)$. Therefore we first show that

$$
Q^{\pi^{\theta^{(n)}}}(s_h, a^*(s_h)) - Q^{\pi^{\theta^{(n)}}}(s_h, a) \geq \frac{\Delta^*(s_h)}{2}, \tag{16}
$$

for all $a \neq a^*(s_h)$. This holds true, because

$$
\begin{aligned}
&Q^{\pi^{\theta^{(n)}}}(s_h, a^*(s_h)) - Q^{\pi^{\theta^{(n)}}}(s_h, a) \\
&= Q^{\pi^{\theta^{(n)}}}(s_h, a^*(s_h)) - Q^\infty(s_h, a^*(s_h)) + Q^\infty(s_h, a^*(s_h)) - Q^{\pi^{\theta^{(n)}}}(s_h, a) \\
&\geq -\frac{\Delta^*(s_h)}{2} + Q^\infty(s_h, a^*(s_h)) - Q^\infty(s_h, a) + Q^\infty(s_h, a) - Q^{\pi^{\theta^{(n)}}}(s_h, a) \\
&\geq -\frac{\Delta^*(s_h)}{2} + \Delta^*(s_h) + \sum_{s' \in \mathcal{S}^{[\mathcal{H}]}} p(s'|s_h, a)(V^\infty(s') - V^{\pi^{\theta^{(n)}}}(s')) \\
&\geq \frac{\Delta^*(s_h)}{2}.
\end{aligned}
$$

The first inequality follows from $\theta^{(n)} \in \mathcal{R}_2(s)$, second by the definition of $\Delta^*(s_h)$ and the last from mononicity of $V^{\pi^{\theta^{(n)}}}(s')$ for every $s'$ and $V^\infty$ beeing the limit. Using Lemma A.7 we obtain for any $a \neq a^*(s_h)$ that

$$
\frac{\partial J(\theta^{(n)})}{\partial \theta(s_h, a^*(s_h))} \geq \frac{\partial J(\theta^{(n)})}{\partial \theta(s_h, a)}
$$
$$
\Leftrightarrow \pi^{\theta^{(n)}}(a^*(s_h)|s_h)\left(Q_h^{\pi^{\theta^{(n)}}}(s_h, a^*(s_h)) - V_h^{\pi^{\theta^{(n)}}}(s_h)\right) \geq \pi^{\theta^{(n)}}(a|s)\left(Q_h^{\pi^{\theta^{(n)}}}(s_h, a) - V_h^{\pi^{\theta^{(n)}}}(s_h)\right). \tag{17}
$$

We divide into two cases:

  a) $\pi^{\theta^{(n)}}(a^*(s_h)|s_h) \geq \pi^{\theta^{(n)}}(a|s_h)$,

  b) $\pi^{\theta^{(n)}}(a^*(s_h)|s_h) < \pi^{\theta^{(n)}}(a|s_h)$.

In $a)$ the assumption $\pi^{\theta^{(n)}}(a^*(s_h)|s_h) \geq \pi^{\theta^{(n)}}(a|s_h)$ implies $\theta^{(n)}(s_h, a^*(s_h)) \geq \theta^{(n)}(s_h, a)$. Thus,

$$\theta^{(n+1)}(s, a^*(s_h)) = \theta^{(n)}(s, a^*(s_h)) + \eta \frac{\partial J(\theta^{(n)})}{\partial \theta^{(n)}(s_h, a^*(s_h))}$$

$$\geq \theta^{(n)}(s, a) + \eta \frac{\partial J(\theta^{(n)})}{\partial \theta^{(n)}(s_h, a)}$$

$$= \theta^{(n+1)}(s, a),$$

which implies $\pi^{\theta^{(n+1)}}(a^*(s_h)|s_h) \geq \pi^{\theta^{(n+1)}}(a|s_h)$. Moreover, we have

$$Q_h^{\pi^{\theta^{(n+1)}}}(s_h, a^*(s_h)) - Q_h^{\pi^{\theta^{(n+1)}}}(s_h, a) \geq \frac{\Delta^*(s_h)}{2} \geq 0,$$

$$Q_h^{\pi^{\theta^{(n+1)}}}(s_h, a^*(s_h)) - V_h^{\pi^{\theta^{(n+1)}}}(s_h) \geq Q_h^{\pi^{\theta^{(n+1)}}}(s_h, a) - V_h^{\pi^{\theta^{(n+1)}}}(s_h).$$

Thus, both together yields

$$\pi^{\theta^{(n+1)}}(a^*(s_h)|s_h)\big(Q_h^{\pi^{\theta^{(n+1)}}}(s_h, a^*(s_h)) - V_h^{\pi^{\theta^{(n+1)}}}(s_h)\big) \geq \pi_t^{\theta^{(n+1)}}(a|s_h)\big(Q_h^{\pi^{\theta^{(n+1)}}}(s_h, a) - V_h^{\pi^{\theta^{(n+1)}}}(s_h)\big),$$

which is by Equation 17 equivalent to

$$\frac{\partial J(\theta^{(n+1)})}{\partial \theta^{(n+1)}(s, a^*(s_h))} \geq \frac{\partial J(\theta^{(n+1)})}{\partial \theta^{(n+1)}(s_h, a)}.$$

Hence, $\theta^{(n+1)} \in \mathcal{R}_1(s_h)$.

In $b)$ assume now that $\pi^{\theta^{(n)}}(a^*(s_h)|s_h) < \pi^{\theta^{(n)}}(a|s_h)$. As $\theta^{(n)} \in \mathcal{R}_1(s_h)$ Equation 17 is also true in this case and rearranging of terms gives

$$\frac{\partial J(\theta^{(n)})}{\partial \theta^{(n)}(s_h, a^*(s_h))} \geq \frac{\partial J(\theta^{(n)})}{\partial \theta^{(n)}(s_h, a)}$$

$$\Leftrightarrow Q_h^{\pi^{\theta^{(n)}}}(s_h, a^*(s_h)) - Q_h^{\pi^{\theta^{(n)}}}(s_h, a) \geq \Big(1 - \frac{\pi^{\theta^{(n)}}(a^*(s_h)|s_h)}{\pi^{\theta^{(n)}}(a|s_h)}\Big)\big(Q_h^{\pi^{\theta^{(n)}}}(s_h, a^*(s_h)) - V_h^{\pi^{\theta^{(n)}}}(s_h)\big)$$

$$\Leftrightarrow Q_h^{\pi^{\theta^{(n)}}}(s_h, a^*(s_h)) - Q_h^{\pi^{\theta^{(n)}}}(s_h, a) \geq \big(1 - \exp(\theta^{(n)}(s_h, a^*(s_h)) - \theta^{(n)}(s_h, a))\big)\big(Q_h^{\pi^{\theta^{(n)}}}(s_h, a^*(s_h)) - V_h^{\pi^{\theta^{(n)}}}(s_h)\big).$$
$$\tag{18}$$

Note next that by $\theta^{(n)} \in \mathcal{R}_1(s_h)$ and definition of $\mathcal{R}_1(s_h)$ we have

$$\theta^{(n+1)}(s_h, a^*(s_h)) - \theta^{(n+1)}(s_h, a)$$

$$= \theta^{(n)}(s_h, a^*(s_h)) + \eta \frac{\partial J(\theta^{(n)})}{\partial \theta^{(n)}(s_h, a^*(s_h))} - \theta^{(n)}(s_h, a) - \eta \frac{\partial J(\theta^{(n)})}{\partial \theta^{(n)}(s_h, a)}$$

$$\geq \theta^{(n)}(s_h, a^*(s_h)) - \theta^{(n)}(s_h, a)$$

and is follows $\big(1 - \exp(\theta^{(n+1)}(s_h, a^*(s_h)) - \theta^{(n+1)}(s_h, a))\big) \leq \big(1 - \exp(\theta^{(n)}(s_h, a^*(s_h)) - \theta^{(n)}(s_h, a))\big) < 1$ by assumption $b)$. We already know $\theta^{(n+1)} \in \mathcal{R}_3(s_h)$ and therefore $V_h^{\pi^{\theta^{(n+1)}}}(s_h) \geq Q_h^{\pi^{\theta^{(n+1)}}}(s_h, a^*(s_h)) - \frac{\Delta^*(s)}{2}$. This leads to

$$Q_h^{\pi^{\theta^{(n+1)}}}(s_h, a^*(s_h)) - V_h^{\pi^{\theta^{(n+1)}}}(s_h) \leq \frac{\Delta^*(s)}{2} \leq Q_h^{\pi^{\theta^{(n+1)}}}(s_h, a^*(s_h)) - Q_h^{\pi^{\theta^{(n+1)}}}(s_h, a),$$

where the last inequality is due to Equation 16. Combining everything leads to

$$\big(1 - \exp(\theta^{(n+1)}(s, a_h^*(s)) - \theta^{(n+1)}(s, a))\big)\Big[Q_h^{\pi^{\theta^{(n+1)}}}(s_h, a^*(s_h)) - V_h^{\pi^{\theta^{(n+1)}}}(s_h)\Big]$$

$$\leq Q_h^{\pi^{\theta^{(n+1)}}}(s_h, a^*(s_h)) - Q_h^{\pi^{\theta^{(n+1)}}}(s_h, a),$$

which is by Equation 18 equivalent to $\theta^{(n+1)} \in \mathcal{R}_1(s_h)$.

Now we come to Claim (ii).

$$\pi^{\theta^{(n+1)}}(a^*(s_h)|s_h)$$

$$= \frac{\exp(\theta^{(n+1)}(s_h, a^*(s_h)))}{\sum\limits_{a \in \mathcal{A}} \exp(\theta^{(n+1)}(s_h, a))}$$

$$= \frac{\exp(\theta^{(n)}(s_h, a^*(s_h)) + \eta \frac{\partial J(\theta^{(n)})}{\partial \theta^{(n)}(s_h, a^*(s_h))})}{\sum\limits_{a \in \mathcal{A}} \exp(\theta^{(n)}(s_h, a) + \eta \frac{\partial J(\theta^{(n)})}{\partial \theta^{(n)}(s_h, a)})}$$

$$\geq \frac{\exp(\theta^{(n)}(s_h, a^*(s_h))) \exp(\eta \frac{\partial J(\theta^{(n)})}{\partial \theta^{(n)}(s_h, a^*(s_h))})}{\sum\limits_{a \in \mathcal{A}} \exp(\theta^{(n)}(s_h, a)) \exp(\eta \frac{\partial J(\theta^{(n)})}{\partial \theta^{(n)}(s_h, a^*(s_h))})}$$

$$= \pi^{\theta^{(n)}}(a^*(s_h)|s_h),$$

where the inequality follows by $\theta^{(n)} \in \mathcal{R}_1(s_h)$.

**Claim 2.** Assume $\theta \in \mathcal{N}_c(s_h) \cap \mathcal{R}_2(s_h) \cap \mathcal{R}_3(s_h)$ and divide again in two cases. If $a)$ $\pi^\theta(a^*(s_h)|s_h) \geq \max\limits_{a \in \mathcal{A}} \pi^\theta(a|s_h)$, then for all $a \neq a^*(s_h)$ we have

$$\frac{\partial J(\theta)}{\partial \theta(s_h, a^*(s_h))}$$

$$= \tilde{\rho}_\mu^{\pi^\theta}(s_h) \pi^\theta(a^*(s_h)|s_h) A^{\pi^\theta}(s_h, a^*(s_h))$$

$$\geq \tilde{\rho}_\mu^{\pi^\theta}(s_h) \pi^\theta(a|s_h) A^{\pi^\theta}(s_h, a)$$

$$= \frac{\partial J(\theta)}{\partial \theta(s_h, a)}.$$

Where the inequality follows from $A^{\pi^\theta}(s_h, a^*(s_h)) - A^{\pi^\theta}(s_h, a) = Q^{\pi^\theta}(s_h, a^*(s_h)) - Q^{\pi^\theta}(s_h, a) \geq \frac{\Delta^*(s_h)}{2} > 0$ by Equation 16. Hence, $\theta \in \mathcal{R}_1(s_h)$.

The case $b)$ where $\pi^\theta(a^*(s_h)|s_h) < \max\limits_{a \in \mathcal{A}} \pi^\theta(a|s_h)$ is not possible for $\theta \in \mathcal{N}_c(s_h)$. Assume there exists $a \neq a^*(s_h)$ such that $\pi^\theta(a^*(s_h)|s_h) < \pi^\theta(a|s_h)$. Then

$$\pi^\theta(a^*(s_h)|s_h) + \pi^\theta(a|s_h) > \frac{2c(s_h)}{c(s_h) + 1} = \frac{\frac{2|\mathcal{A}|HR^*}{\Delta^*(s)} - 2}{\frac{|\mathcal{A}|HR^*}{\Delta^*(s)}} = 2 - \frac{2\Delta^*(s)}{|\mathcal{A}|HR^*} \geq 2 - \frac{2}{|\mathcal{A}|} \geq 1,$$

because $\Delta^*(s) \leq HR^*$ by definition and $|\mathcal{A}| \geq 2$. This is a contradiction as $\pi^\theta$ is a probability distribution and Claim 2 is proven.

**Claim 3.** By the asymptotic convergence of Theorem C.1, we have that $\pi^{\theta^{(n)}}(a^*(s_h)|s_h) \to 1$ for $n \to \infty$. Thus, there exists an $N_0(s_h) > 0$, such that $\pi^{\theta^{(n)}}(a^*(s_h)|s_h) \geq \frac{c(s_h)}{c(s_h)+1}$ for all $n \geq N_0(s_h)$, i.e. $\theta^{(n)} \in N_c(s_h)$ for all $n \geq N_0(s_h)$.

Furthermore, as $Q^{\pi^{\theta^{(n)}}}(s_h, a^*(s_h)) \to Q^\infty(s_h, a^*(s_h))$ for $n \to \infty$ there exists $N_1(s_h)$ such that $\theta^{(n)} \in \mathcal{R}_2(s_h)$ for all $n \geq N_1(s_h)$.

Moreover, as $Q^{\pi^{\theta^{(n)}}}(s_h, a^*(s_h)) \to Q^\infty(s_h, a^*(s_h)) = V^\infty(s_h)$ and $V^{\pi^{\theta^{(n)}}}(s_h) \to V^\infty(s_h)$ for $n \to \infty$ there exists $N_2(s_h)$ such that $\theta^{(n)} \in \mathcal{R}_3(s_h)$ for all $n \geq N_2(s_h)$.

We choose $n_0(s_h) = \max\{N_0(s_h), N_1(s_h), N_2(s_h)\}$ which proves Claim 3.

$\square$

**Theorem 3.2.** *Under Assumption 3.1, let $\mu$ be a probability measure such that $\mu(s) > 0$ for all $s \in \mathcal{S}$, let $\eta = \frac{1}{5H^2R^*}$ and consider the sequence $(\theta^{(n)})$ generated by Algorithm 1 with arbitrary initialisation $\theta^{(0)}$. For $\epsilon > 0$ choose the number of training steps as $N = \frac{10H^5R^*|\mathcal{S}|}{c^2\epsilon} \left\| \frac{d_\mu^{\pi^*}}{\mu} \right\|_\infty^2$. Then it holds that*

$$V_0^*(\mu) - V_0^{\pi^{\theta^{(N)}}}(\mu) \leq \epsilon.$$

*Proof.* We will show that

$$J^*(\mu) - J(\theta^{(n)}, \mu) = V_0^*(\mu) - V_0^{\pi^{\theta^{(n)}}}(\mu) \leq \frac{10H^5 R^* |\mathcal{S}|}{c^2 n} \left\| \frac{d_\mu^{\pi^*}}{\mu} \right\|_\infty^2,$$

then the claim follows immediately from this.

For any $\beta$-smooth function $f : \mathbb{R}^d \to \mathbb{R}$ the descent lemma gives (see Beck, 2017, Lemma 5.7)

$$f(y) \leq f(x) + \nabla f(x)^T (y - x) + \frac{\beta}{2} \|y - x\|^2.$$

As $-f$ is also $\beta$-smooth we follow

$$-f(y) \leq -f(x) - \nabla f(x)^T (y - x) + \frac{\beta}{2} \|y - x\|^2,$$

which is equivalent to

$$f(y) \geq f(x) + \nabla f(x)^T (y - x) - \frac{\beta}{2} \|y - x\|^2. \tag{19}$$

Now for gradient ascent updates

$$x_{k+1} = x_k + \alpha \nabla f(x_k)$$

we have that

$$f(x_{k+1}) \geq f(x_k) + \nabla f(x_k)^T (x_{k+1} - x_k) - \frac{\beta}{2} \|x_{k+1} - x_k\|^2$$

$$= f(x_k) + \alpha \|\nabla f(x_k)\|^2 - \frac{\beta \alpha^2}{2} \|\nabla f(x_k)\|^2$$

$$= f(x_k) + \left( \alpha - \frac{\beta \alpha^2}{2} \right) \|\nabla f(x_k)\|^2.$$

It follows for the maximum $f^*$ of $f$ that

$$f^* - f(x_{k+1}) \leq f^* - f(x_k) - \left( \alpha - \frac{\beta \alpha^2}{2} \right) \|\nabla f(x_k)\|^2.$$

Now assume that there exists a $b > 0$ such that $\|\nabla f(x_k)\| > b(f^* - f(x_k))$ for all $k \geq 0$, then

$$f^* - f(x_{k+1}) \leq f^* - f(x_k) - \left( \alpha - \frac{\beta \alpha^2}{2} \right) b^2 (f^* - f(x_k))^2.$$

We choose the step size $\alpha \leq \frac{1}{\beta}$, then

$$f^* - f(x_{k+1}) \leq f^* - f(x_k) - \frac{\alpha c^2}{2} (f^* - f(x_k))^2.$$

When $f^* - f(x_1) \leq \frac{2}{\alpha b^2}$, then $f^* - f(x_n) \leq \frac{2}{\alpha b^2 n}$ (see Lemma B.7).

We apply this to our objective $J(\theta, \mu)$ with $\alpha = \eta = 5H^2 R^*$ and $b = \frac{c}{\sqrt{|\mathcal{S}|HH}} \left\| \frac{d_\mu^{\pi^*}}{\mu} \right\|^{-1}$. Note for $b$, that $d_\mu^{\pi^\theta}(s) \geq \frac{1}{H} \mu(s)$ by definition for any $\theta$ (see also Remark B.3. So, we only need to check that

$$J^*(\mu) - J(\theta^{(0)}, \mu) \leq \frac{2H^2 R^* 5H^2 H |\mathcal{S}|}{c^2} \left\| \frac{d_\mu^{\pi^*}}{\mu} \right\|_\infty^2.$$

This is directly given by the bounded reward assumption and the fact that $c < 1$ and $\left\| \frac{d_\mu^{\pi^*}}{\mu} \right\|_\infty^2 > 1$. Then, we yield the claim

$$J^*(\mu) - J(\theta^{(n)}, \mu) \leq \frac{10H^5 R^* |\mathcal{S}|}{c^2 n} \left\| \frac{d_\mu^{\pi^*}}{\mu} \right\|_\infty^2.$$

$\square$

*Remark* B.6. In the discounted setting Mei et al. (2020) obtain the factor $(1 - \gamma)^{-6}$, where a power of 3 is due to their smoothness constant $\frac{8}{(1-\gamma)^3}^2$, a power of 2 is due to the distribution mismatch coefficient and the additional power is due to comparing value functions with a different start distribution $\rho$ instead of $\mu$. Comparing to our results, directly using $\mu$ leads also to a factor $(1 - \gamma)^{-5}$. For the simultaneous PG the smoothness of order $H^2 R^*$ leads to a $H^2$ in the convergence rate, then the distribution mismatch coefficient adds another $H^2$ and the additional $H$ comes from the PL-inequality, as the cardinality of the enlarged state space under Assumption 3.1 is $|\mathcal{S}^{[\mathcal{H}]}| = |\mathcal{S}|H$. As mentioned in the article, it cannot be proven that $c$ is independent of $H$. We omitted this dependency when we compare to the discounted case because the model dependent constant there could also depend on $\gamma$ in the same sense.

**Lemma B.7.** *Let $(d_n)_{n \in \mathbb{N}_0}$ be a positive sequence, such that $d_{n+1} \leq d_n - qd_n^2$ for some $q > 0$ and $d_0 < \frac{1}{q}$, then $d_n \leq \frac{1}{qn}$.*

*Proof.* We use an argument similar to Nesterov (2013, Thm. 2.1.14). It holds

$$\frac{1}{d_{n+1}} \geq \frac{1}{d_n} + \frac{qd_n}{d_{n+1}} \geq \frac{1}{d_n} + q,$$

where the first inequality is due to dividing by $d_n d_{n+1}$ and the second inequality follows by monotonicity. Using a telescope-sum argument we obtain

$$\frac{1}{d_n} = \frac{1}{d_0} + \sum_{k=0}^{n-1} \frac{1}{d_{k+1}} - \frac{1}{d_k} \geq \frac{1}{d_0} + nq.$$

Finally,

$$d_n \leq \frac{1}{nq + \frac{1}{d_0}} \leq \frac{1}{q(n+1)} \leq \frac{1}{qn}.$$

$\square$

## B.2 PROOFS OF SECTION 3.2

**Lemma B.8.** *Let $h \in \mathcal{H}$, then the objective $J_h(\theta_h, \tilde{\pi}_{(h+1)}, \mu_h)$ from Equation 5 is smooth in $\theta_h$ with parameter $\beta_h = 2(H - h)R^*$.*

*Proof.* Note that we can interpret the objective function $J_h(\theta_h, \tilde{\pi}_{(h+1)}, \mu_h)$ as a value function of a one-step discounted MDP with $\gamma = 0$ and bounded rewards between $[0, R^*(H - h)]$. Hence, we can use Yuan et al. (2022, Lem 4.4 and 4.8) to obtain that the softmax policy $\pi^{\theta_h}$ fulfills the desired properties with

$$\mathbb{E}_{A \sim \pi^{\theta_h}}\left[||\nabla \log \pi^{\theta_h}(A|s)||_2^2\right] \leq 1 - \frac{1}{|\mathcal{A}_s|} \leq 1 \quad \forall s \in \mathcal{S}$$

$$\mathbb{E}_{A \sim \pi^{\theta_h}}\left[||\nabla^2 \log \pi^{\theta_h}(A|s)||_2\right] \leq 1,$$

which leads to a smoothness constant $\beta_h = 2(H - h)R^*$ for the objective function $J_h$. $\square$

**Lemma B.9.** *It holds that*

$$\|\nabla J_h(\theta_h, \tilde{\pi}_{(h+1)}, \mu_h)\|_2 \geq \min_{s \in \mathcal{S}_h} \pi^{\theta_h}(a_h^*(s)|s)(J_h^*(\tilde{\pi}_{(h+1)}, \mu_h) - J_h(\theta_h, \tilde{\pi}_{(h+1)}, \mu_h)).$$

*Proof.* First note that by the definition of $\pi_h^*$, we have $J_h^*(\tilde{\pi}_{(h+1)}, \mu_h) = V_h^{(\pi_h^*, \tilde{\pi}_{(h+1)})}(\mu_h)$, because the tabular softmax parametrisation can approximate any deterministic policy arbitrarily well. Using

---

[2]Choosing the learning rate $(1 - \gamma)^3/8$ leads to the $(1 - \gamma)^{-3}$ factor in the convergence rate. Using recent results in (Yuan et al., 2022, Lem 4) one can improve the smoothness constant for discounted MDPs. Still, the global convergence result in (Agarwal et al., 2021, Thm 5) holds only for a learning rate $(1 - \gamma)^3/8$ and hence does not lead to direct improvement in the convergence rate.

the performance difference lemma in the dynamic setting from Corollary A.4 and the derivative of the objective given in Lemma A.8, we obtain

$$
\left\| \frac{\partial J_h(\theta_h, \tilde{\pi}_{(h+1)}, \mu_h)}{\partial \theta_h} \right\|_2
$$

$$
= \left\| \sum_{s \in \mathcal{S}_h} \mu_h(s) \frac{\partial J_h(\theta_h, \tilde{\pi}_{(h+1)}, \delta_s)}{\partial \theta_h} \right\|_2
$$

$$
= \left[ \sum_{s' \in \mathcal{S}_h} \sum_{a' \in \mathcal{A}_{s'}} \left( \sum_{s \in \mathcal{S}_h} \mu_h(s) \frac{\partial J_h(\theta_h, \tilde{\pi}_{(h+1)}, \delta_s)}{\partial \theta_h(s', a')} \right)^2 \right]^{\frac{1}{2}}
$$

$$
\geq \sum_{s \in \mathcal{S}_h} \mu_h(s) \left| \frac{\partial J_h(\theta_h, \tilde{\pi}_{(h+1)}, \delta_s)}{\partial \theta_h(s, a_h^*(s))} \right|
$$

$$
= \sum_{s \in \mathcal{S}_h} \mu_h(s) \pi^{\theta_h}(a_h^*(s)|s) A_h^{(\pi^{\theta_h}, \tilde{\pi}_{(h+1)})}(s, a_h^*(s))
$$

$$
= \sum_{s \in \mathcal{S}_h} \mu_h(s) \pi^{\theta_h}(a_h^*(s)|s) \left( J_h^*(\tilde{\pi}_{(h+1)}, \delta_s) - J_h(\theta_h, \tilde{\pi}_{(h+1)}, \delta_s) \right)
$$

$$
\geq \min_{s \in \mathcal{S}_h} \pi^{\theta_h}(a_h^*(s)|s) \left( J_h^*(\tilde{\pi}_{(h+1)}, \mu_h) - J_h(\theta_h, \tilde{\pi}_{(h+1)}, \mu_h) \right).
$$

The first inequality is due to the non-negativity of all other terms, and we just drop them. □

**Lemma B.10.** *Let $\mu_h$ be a probability measure such that $\mu_h(s) > 0$ for all $s \in \mathcal{S}_h$ and let $0 < \eta_h \leq \frac{1}{2(H-h)R^*}$. Consider the sequence $(\theta_h^{(n)})$ generated by Algorithm 2 for arbitrary $\theta_h^{(0)} \in \mathcal{R}^{d_h}$ and $\tilde{\pi}$. Then, $c_h = c_h(\theta_h^{(n)}) = \inf_{n \geq 0} \min_{s \in \mathcal{S}_h} \pi^{\theta_h^{(n)}}(a_h^*(s)|s) > 0$.*

The idea of the proof is based on Mei et al. (2020, Lemma 5) for bandits and extended to the contextual bandit case.

*Proof.* Throughout the proof we abuse notation as follows:

- As we consider a fixed time point $h$ we will only write $\theta_n$ instead of $\theta_h^{(n)}$.

- We denote the objective function by $J_h(\theta)$ instead of $J_h(\theta, \tilde{\pi}_{(h+1)}, \mu_h)$ for a fixed policy $\tilde{\pi}$ and start distribution $\mu_h$. Furthermore, we will just write $J_h^*$ instead of $J_h^*(\tilde{\pi}_{(h+1)}, \mu_h)$.

- We will write $J_{h,s}(\theta)$ for the objective function which starts almost surly in $s \in \mathcal{S}_h$, i.e. $J_{h,s}(\theta) = J_h(\theta, \tilde{\pi}_{(h+1)}, \delta_s)$.

First note that

$$
J_{h,s}(\theta_h) = \sum_{a \in \mathcal{A}_s} \pi^{\theta_h}(a|s) Q_h^{\tilde{\pi}}(s, a),
$$

where $Q_h^{\tilde{\pi}}(s, a)$ is independent of $\theta$. We will drop the subscript $\tilde{\pi}$ in $Q_h$ for the rest of the proof and define for all $s \in \mathcal{S}_h$,

$$
\Delta^*(s) = Q_h(s, a_h^*(s)) - \max_{a \neq a_h^*(s)} Q_h(s, a) > 0, \quad \text{and} \quad \Delta^* = \min_{s \in \mathcal{S}_h} \Delta^*(s) > 0.
$$

Consider the following sets

$$
\mathcal{R}_h^1(s) = \{\theta : \frac{\partial J_{h,s}(\theta)}{\partial \theta(s, a_h^*(s))} \geq \frac{\partial J_{h,s}(\theta)}{\partial \theta(s, a)} \forall a \neq a_h^*(s)\}
$$

$$
\mathcal{R}_h^2(s) = \{\theta : \pi^\theta(a_h^*(s)|s) \geq \pi^\theta(a|s) \forall a \neq a_h^*(s)\}
$$

$$
\mathcal{N}_h(s) = \{\theta : \pi^\theta(a_h^*(s)|s) \geq \frac{c_h(s)}{c_h(s) + 1}\},
$$

for $c_h(s) = \frac{|\mathcal{A}|(H-h)R^*}{\Delta_h^*(s)} - 1$ and $\Delta_h^*(s) = Q_h(s, a^*(s)) - \max_{a \neq a^*} Q_h(s, a)$. Then consider the following Claims:

1. $\theta_n \in \mathcal{R}_h^1(s) \Rightarrow \theta_{n+1} \in \mathcal{R}_h^1(s)$,

2. If $\theta_n \in \mathcal{R}_h^1(s)$, then $\pi^{\theta_{n+1}}(a_h^*(s)|s) \geq \pi^{\theta_n}(a_h^*(s)|s)$,

3. $\mathcal{N}_h(s) \subseteq \mathcal{R}_h^2(s) \subseteq \mathcal{R}_h^1(s)$.

**Claim 1.** Let $\theta_n \in \mathcal{R}_h^1(s)$ and $a \neq a_h^*(s)$. Using the derivative of the value function we obtain

$$\frac{\partial J_{h,s}(\theta_n)}{\partial \theta(s, a_h^*(s))} \geq \frac{\partial J_{h,s}(\theta_n)}{\partial \theta(s, a)} \tag{20}$$
$$\Leftrightarrow \pi^{\theta_n}(a_h^*(s)|s)\big(Q_h(s, a_h^*(s)) - J_{h,s}(\theta_n)\big) \geq \pi^{\theta_n}(a|s)\big(Q_h(s, a) - J_{h,s}(\theta_n)\big).$$

We divide into two cases:

a) $\pi^{\theta_n}(a_h^*(s)|s) \geq \pi^{\theta_n}(a|s)$,

b) $\pi^{\theta_n}(a_h^*(s)|s) < \pi^{\theta_n}(a|s)$.

In $a)$ the assumption $\pi^{\theta_n}(a_h^*(s)|s) \geq \pi^{\theta_n}(a|s)$ implies $\theta_n(s, a_h^*(s)) \geq \theta_n(s, a)$. Thus,

$$\theta_{n+1}(s, a_h^*(s)) = \theta_n(s, a_h^*(s)) + \eta_h \mu_h(s) \frac{\partial J_{h,s}(\theta_n)}{\partial \theta_n(s, a_h^*(s))}$$
$$\geq \theta_n(s, a) + \eta_h \mu_h(s) \frac{\partial J_{h,s}(\theta_n)}{\partial \theta_n(s, a)}$$
$$= \theta_{n+1}(s, a),$$

which implies $\pi^{\theta_{n+1}}(a_h^*(s)|s) \geq \pi^{\theta_{n+1}}(a|s)$. By the optimality of $a_h^*(s)$ we follow

$$\pi_t^{\theta_{n+1}}(a_h^*(s)|s)\big(Q_h(s, a_h^*(s)) - J_{h,s}(\theta_{n+1})\big) \geq \pi_t^{\theta_{n+1}}(a|s)\big(Q_h(s, a) - J_{h,s}(\theta_{n+1})\big),$$

which is by Equation 20 equivalent to

$$\frac{\partial J_{h,s}(\theta_{n+1})}{\partial \theta_{n+1}(s, a_h^*(s))} \geq \frac{\partial J_{h,s}(\theta_{n+1})}{\partial \theta_{n+1}(s, a)}.$$

Hence, $\theta_{n+1} \in \mathcal{R}_h^1(s)$.
In $b)$ assume now that $\pi^{\theta_n}(a_h^*(s)|s) < \pi^{\theta_n}(a|s)$. As $\theta_n \in \mathcal{R}_h^1(s)$ Equation 20 is also true in this case and rearranging of terms gives

$$\frac{\partial J_{h,s}(\theta_n)}{\partial \theta_n(s, a_h^*(s))} \geq \frac{\partial J_{h,s}(\theta_n)}{\partial \theta_n(s, a)}$$
$$\Leftrightarrow Q_h(s, a_h^*(s)) - Q_h(s, a) \geq \Big(1 - \frac{\pi^{\theta_n}(a_h^*(s)|s)}{\pi^{\theta_n}(a|s)}\Big)\big(Q_h(s, a_h^*(s)) - J_{h,s}(\theta_n)\big)$$
$$\Leftrightarrow Q_h(s, a_h^*(s)) - Q_h(s, a) \geq \big(1 - \exp(\theta_n(s, a_h^*(s)) - \theta_n(s, a))\big)\big(Q_h(s, a_h^*(s)) - J_{h,s}(\theta_n)\big). \tag{21}$$

Note next that by $\theta^{(n)} \in \mathcal{R}_h^1(s)$ and definition of $\mathcal{R}_h^1(s)$ we have

$$\theta_{n+1}(s, a_h^*(s)) - \theta_{n+1}(s, a)$$
$$= \theta_n(s, a_h^*(s)) + \eta_h \mu_h(s) \frac{\partial J_{h,s}(\theta_n)}{\partial \theta_n(s, a_h^*(s))} - \theta_n(s, a) - \eta_h \mu_h(s) \frac{\partial J_{h,s}(\theta_n)}{\partial \theta_n(s, a)}$$
$$\geq \theta_n(s, a_h^*(s)) - \theta_n(s, a)$$

and is follows $\big(1 - \exp(\theta_{n+1}(s, a_h^*(s)) - \theta_{n+1}(s, a))\big) \leq \big(1 - \exp(\theta_n(s, a_h^*(s)) - \theta_n(s, a))\big) < 1$ by assumption $b)$. By the ascent lemma for smooth functions we get monotonicity in the objective function, so

$$Q_h(s, a_h^*(s)) - J_{h,s}(\theta_{n+1}) \leq Q_h(s, a_h^*(s)) - J_{h,s}(\theta_n),$$

where the last inequality is due to the definition of $\Delta^*(s)$. Combining everything leads to

$$\big(1 - \exp(\theta_{n+1}(s, a_h^*(s)) - \theta_{n+1}(s, a)))\big[Q_h(s, a_h^*(s)) - J_{h,s}(\theta_{n+1})\big]$$

$$\leq \big(1 - \exp(\theta_n(s, a_h^*(s)) - \theta_n(s, a)))\big[Q_h(s, a_h^*(s)) - J_{h,s}(\theta_n)\big]$$

$$\leq Q_h(s, a_h^*(s)) - Q_h(s, a),$$

which is by Equation 21 equivalent to $\theta_{n+1} \in \mathcal{R}_1(s)$.

**Claim 2.** If $\theta_n \in \mathcal{R}_h^1(s)$, then

$$\pi^{\theta_{n+1}}(a_h^*(s)|s)$$

$$= \frac{\exp(\theta_{n+1}(s, a_h^*(s)))}{\sum\limits_{a \in \mathcal{A}} \exp(\theta_{n+1}(s, a))}$$

$$= \frac{\exp(\theta_n(s, a_h^*(s)) + \eta_h \mu_h(s)\frac{\partial J_{h,s}(\theta_n)}{\partial \theta_n(s, a_h^*(s))})}{\sum\limits_{a \in \mathcal{A}_s} \exp(\theta_n(s, a) + \eta_h \mu_h(s)\frac{\partial J_{h,s}(\theta_n)}{\partial \theta_n(s, a)})}$$

$$\geq \frac{\exp(\theta_n(s, a_h^*(s))) \exp(\eta_h \mu_h(s)\frac{\partial J_{h,s}(\theta_n)}{\partial \theta_n(s, a_h^*(s))})}{\sum\limits_{a \in \mathcal{A}_s} \exp(\theta_n(s, a)) \exp(\eta_h \mu_h(s)\frac{\partial J_{h,s}(\theta_n)}{\partial \theta_n(s, a_h^*(s))})}$$

$$= \pi^{\theta_n}(a_h^*(s)|s),$$

where the inequality follows by $\theta_n \in \mathcal{R}_h^1(s)$.

**Claim 3.** Let $\theta_n \in \mathcal{R}_h^2(s)$, then by the optimality of $a^*(s)$,

$$\pi^{\theta_n}(a^*(s)|s)(Q_h(s, a_h^*(s)) - J_{h,s}(\theta_n)) \geq \pi^{\theta_n}(a|s)(Q_h(s, a) - J_{h,s}(\theta_n)) \tag{22}$$

$$\Leftrightarrow \frac{\partial J_{h,s}(\theta_n)}{\partial \theta_n(s, a^*(s))} \geq \frac{\partial J_{h,s}(\theta_n)}{\partial \theta_n(s, a)}. \tag{23}$$

Hence, $\theta_n \in \mathcal{R}_h^1(s)$.

On the other hand, let $\theta_n \in \mathcal{N}_h(s)$, then assume there exists $a \neq a_h^*(s)$ such that $\pi^\theta(a_h^*(s)|s) < \pi^\theta(a|s)$. Then

$$\pi^\theta(a_h^*(s)|s) + \pi^\theta(a|s) > \frac{2c(s)}{c(s)+1} = \frac{\frac{2|\mathcal{A}|(H-h)R^*}{\Delta_h^*(s)} - 2}{\frac{|\mathcal{A}|(H-h)R^*}{\Delta_h^*(s)}} = 2 - \frac{2\Delta_h^*(s)}{|\mathcal{A}|(H-h)R^*} \geq 2 - \frac{2}{|\mathcal{A}|} \geq 1,$$

because $\Delta^*(s) \leq (H-h)R^*$ by definition and $|\mathcal{A}| \geq 2$. This is a contradiction as $\pi^\theta$ is a probability distribution and Claim 3 is proven.

To follow the claim of the lemma from the claims 1 to 3, we need asymptotic convergence to the global optimum. This is given by Agarwal et al. (2021, Theorem 5), since we can interpret the objective $J_h$ as a one-step MDP with $\gamma = 0$. Then, assuring that the step size is smaller than one over the smoothness parameter is enough to use the same proof as provided in Agarwal et al. (2021).

So, there exists a time $t_0$ such that $\theta \in \mathcal{N}_h(s)$ for all $s \in \mathcal{S}$. Finally,

$$\inf_n \min_s \pi^{\theta_n}(a^*(s)|s) = \min_{0 \leq n \leq t_0} \min_s \pi^{\theta_n}(a^*(s)|s) > 0.$$

$\square$

**Proposition 3.3.** *For fixed $h \in \mathcal{H}$, let $\mu_h$ be a probability measure such that $\mu_h(s) > 0$ for all $s \in \mathcal{S}_h$ and let $0 < \eta_h \leq \frac{1}{2(H-h)R^*}$. Let $\theta_h^{(0)} \in \mathcal{R}^{d_h}$ be an initialisation such that the initial policy is a uniform distribution, then $c_h = \frac{1}{|\mathcal{A}|} > 0$.*

*Proof.* If we initialise uniformly, then $\theta_0 \in \mathcal{R}_h^2(s)$ for all $s \in \mathcal{S}_h$ from the proof of the previous Lemma B.10. Therefore, $\theta_n \in \mathcal{R}_h^1(s)$ for all $n \geq 0$ and from Claim 2 we have

$$c_h = \inf_n \min_s \pi^{\theta_h^{(n)}}(a^*(s)|s) = \min_s \pi^{\theta_h^{(0)}}(a^*(s)|s) = \frac{1}{|\mathcal{A}|}.$$

$\square$

*Remark* B.11. Let us shortly discuss why the constant $c_h$ in the dynamic case can be bound explicitly, but $c$ in the simultaneous case can not. In the dynamic case, the future policy is fixed and the optimal action $a^*(s)$ is chosen with respect to the best policy $\pi_h^*$ dependent on the fixed future policy $\tilde{\pi}_{(h+1)}$. This leads to monotone improvements in epoch $h$ (due to smoothness), and so the probability to choose the best action increases over the training steps. In the simultaneous approach the future policy changes in ever training step, as $\theta$ changes after every update. Smoothness just guarantees improvement in the value function, i.e. improvement in expectation over the time horizon. As $\pi^*$ is the final best policy, we have to bound $\min_{s_h \in \mathcal{S}^{[\mathcal{H}]}} \pi^\theta(a^*(s_h)|s_h)$ for every $s_h$ in the enlarged state space. This the cannot be followed from improvement in expectation.

**Lemma 3.4.** *For fixed $h \in \mathcal{H}$, let $\mu_h$ be a probability measure such that $\mu_h(s) > 0$ for all $s \in \mathcal{S}_h$, let $\eta_h = \frac{1}{2(H-h)R^*}$ and consider the sequence $(\theta_h^{(n)})$ generated by Algorithm 2 with arbitrary initialisation $\theta_h^{(0)}$ and $\tilde{\pi}$. For $\epsilon > 0$ choose the number of training steps as $N_h = \frac{4(H-h)R^*}{c_h^2 \epsilon}$. Then it holds that*

$$V_h^{(\pi_h^*, \tilde{\pi}_{(h+1)})}(\mu_h) - V_h^{(\pi^{\theta_h^{(N_h)}}, \tilde{\pi}_{(h+1)})}(\mu_h) \leq \epsilon$$

*Moreover, if $\theta_h^{(0)}$ initialises the uniform distribution the constants $c_h$ can be replaced by $\frac{1}{|\mathcal{A}|}$.*

*Proof.* First, note that $V_h^{(\pi_h^*, \tilde{\pi}_{(h+1)})}(\mu_h) = J_h^*(\tilde{\pi}_{(h+1)}, \mu_h)$ and $V_h^{(\pi^{\theta_h^{(n)}}, \tilde{\pi}_{(h+1)})}(\mu_h) = J_h(\theta_h^{(n)}, \tilde{\pi}_{(h+1)}, \mu_h)$ by definition of $J_h$ and choice of $\pi_h^*$. We will proof

$$J_h^*(\tilde{\pi}_{(h+1)}, \mu_h) - J_h(\theta_h^{(n)}, \tilde{\pi}_{(h+1)}, \mu_h) \leq \frac{4(H-h)R^*}{c_h^2 n},$$

Then the claim follows directly from this.

We use the same arguments as in the proof of Theorem 3.2 for our objective function $J_h(\theta_h, \tilde{\pi}_{(h+1)}, \mu_h)$. Thus, we only need to assure, that

$$J_h^*(\tilde{\pi}_{(h+1)}, \mu_h) - J_h(\theta_h^{(0)}, \tilde{\pi}_{(h+1)}, \mu_h) \leq \frac{1}{q}.$$

for $q = \frac{\alpha b^2}{2}$, with $\alpha = \eta_h = \frac{1}{\beta_h}$ and $b = c_h$. It holds that

$$J_h^*(\tilde{\pi}_{(h+1)}, \mu_h) - J_h(\theta_h^{(0)}, \tilde{\pi}_{(h+1)}, \mu_h) \leq (H-h)R^* \leq \frac{4(H-h)R^*}{c_h^2} = \frac{2\beta_h}{c_h^2} = \frac{1}{q}$$

and the claim follows as in the proof of Theorem 3.2.

$\square$

**Theorem 3.5.** *For all $h \in \mathcal{H}$, let $\mu_h$ be probability measures such that $\mu_h(s) > 0$ for all $s \in \mathcal{S}_h$, let $\eta_h = \frac{1}{2(H-h)R^*}$. For $\epsilon > 0$ choose the number of training steps as $N_h = \frac{4(H-h)HR^*}{c_h^2 \epsilon} \left\| \frac{1}{\mu_h} \right\|_\infty$. Then for the final policy from Algorithm 2, $\hat{\pi}^* = (\pi^{\theta_0^{(N_0)}}, \ldots, \pi^{\theta_{H-1}^{(N_{H-1})}})$, it holds for all $s \in \mathcal{S}_0$ that*

$$V_0^*(s) - V_0^{\hat{\pi}^*}(s) \leq \epsilon.$$

*If $\theta_h^{(0)}$ initialises the uniform distribution the constants $c_h$ can be replaced by $\frac{1}{|\mathcal{A}|}$.*

*Proof.* First note that by our choice of the future policy $\tilde{\pi} = \hat{\pi}^*$ we have

$$J_h(\theta_h^{(N_h)}, \tilde{\pi}_{(h+1)}, \delta_s) = V_h^{\hat{\pi}^*}(s). \tag{24}$$

By Lemma 3.4 we obtain

$$J_h^*(\tilde{\pi}_{(h+1)}, \mu_h) - J_h(\theta_h^{(N_h)}, \tilde{\pi}_{(h+1)}, \mu_h) \leq \frac{4(H-h)R^*}{c_h^2 N_h}.$$

For every $s \in \mathcal{S}_h$,

$$
\begin{aligned}
J_h^*(\tilde{\pi}_{(h+1)}, \delta_s) - J_h(\theta_h^{(N_h)}, \tilde{\pi}_{(h+1)}, \delta_s) &= \sum_{s' \in \mathcal{S}_h} \mu_h(s') \frac{\delta_s(s')}{\mu_h(s')} J_h^*(\tilde{\pi}_{(h+1)}, \delta_s) - J_{h,s}(\theta_h^{(N_h)}, \tilde{\pi}_{(h+1)}, \delta_s) \\
&\leq \left\| \frac{1}{\mu_h} \right\|_\infty \left( J_h^*(\tilde{\pi}_{(h+1)}, \mu_h) - J_h(\theta_h^{(N_h)}, \tilde{\pi}_{(h+1)}, \mu_h) \right) \\
&\leq \frac{4(H-h)R^*}{c_h^2 N_h} \left\| \frac{1}{\mu_h} \right\|_\infty,
\end{aligned}
$$
(25)

where $\left\| \frac{1}{\mu_h} \right\|_\infty = \max_{s \in \mathcal{S}_h} \frac{1}{\mu_h(s)} > 0$ by assumption. As $N_h = \frac{4(H-h)HR^*}{c_h^2 \epsilon} \left\| \frac{1}{\mu_h} \right\|_\infty$, it holds that

$$J_h^*(\tilde{\pi}_{(h+1)}, \delta_s) - J_h(\theta_h^{(N_h)}, \tilde{\pi}_{(h+1)}, \delta_s) \leq \frac{\epsilon}{H}$$
(26)

for every $s \in \mathcal{S}_h$. For $h = H-1$ it follows directly by Equation 24 and the specialty of the last time point that for all $s \in \mathcal{S}_{H-1}$,

$$V_{H-1}^*(s) - V_{H-1}^{\hat{\pi}^*}(s) = J_{H-1}^*(\delta_s) - J_{H-1}(\theta_{H-1}^{(N_{H-1})}, \delta_s) \leq \frac{\epsilon}{H}.$$

Note that the last epoch is independent of $\tilde{\pi}$. Assume now that for all $s \in \mathcal{S}_h$,

$$V_h^*(s) - V_h^{\hat{\pi}^*}(s) \leq \frac{\epsilon(H-h)}{H}.$$
(27)

Then it holds for all $s \in \mathcal{S}_{h-1}$ that,

$$
\begin{aligned}
J_{h-1}^*(\tilde{\pi}_{(h)}, \delta_s) &= \max_{a \in \mathcal{A}_s} \left( r(s,a) + \sum_{s' \in \mathcal{S}_h} p(s'|s,a) V_h^*(s) - \sum_{s' \in \mathcal{S}_h} p(s'|s,a)(V_h^*(s) - V_h^{\hat{\pi}^*}(s)) \right) \\
&\geq \max_{a \in \mathcal{A}_s} \left( r(s,a) + \sum_{s' \in \mathcal{S}_h} p(s'|s,a) V_h^*(s) \right) - \frac{\epsilon(H-h)}{H} \\
&= V_{h-1}^*(s) - \frac{\epsilon(H-h)}{H},
\end{aligned}
$$
(28)

by the Bellman expectation equation for finite-time MDPs (Puterman (2005)). We close the backward induction using Equation 24 such that for all $s \in \mathcal{S}_{h-1}$,

$$
\begin{aligned}
V_{h-1}^*(s) - V_{h-1}^{\hat{\pi}^*}(s) &= V_{h-1}^*(s) - J_{h-1}^*(\tilde{\pi}_{(h)}, \delta_s) + J_{h-1}^*(\tilde{\pi}_{(h)}, \delta_s) - V_{h-1}^{\hat{\pi}^*}(s) \\
&\leq \frac{\epsilon(H-h)}{H} + \frac{\epsilon}{H} \\
&= \frac{\epsilon(H-(h-1))}{H}.
\end{aligned}
$$
(29)

Finally, it holds for $h = 0$ and all $s \in \mathcal{S}_0$ that

$$V_0^*(s) - V_0^{\hat{\pi}^*}(s) \leq \epsilon.$$

$\square$

## C  ASYMPTOTIC CONVERGENCE FOR SIMULTANEOUS PG

In this section we will proof asymptotic convergence of simultaneous softmax PG towards the global optimum. Therefore, we use the extended notation of the state value, state-action value and advantage function introduced in Remark A.2. For the rest of the section we will write $\theta_n = \theta^{(n)}$ to save notation.

**Theorem C.1.** *Let $\mu$ be a probability measure such that $\mu(s) > 0$ for all $s \in \mathcal{S}$ and let $0 < \eta \le \frac{1}{5H^2R^*}$. Consider the sequence $(\theta^{(n)})$ generated by Algorithm 1 for arbitrary $\theta^{(0)} \in \mathcal{R}^{\sum_h d_h}$. Then, for all $s_h \in \mathcal{S}^{[\mathcal{H}]}$ we have $V^{\pi^{\theta^{(n)}}}(s_h) \to V^*(s_h)$ as $n \to \infty$. Especially we have $V_0^{\pi^{\theta^{(n)}}}(s) \to V_0^*(s)$ as $n \to \infty$ for all $s \in \mathcal{S}_0$.*

Before we can proof this result we have to proof a row of lemmata. The outline follows the proof of Agarwal et al. (2021, Theorem 5). For the rest of this section we will just write $J(\theta)$ or $J^*$ instead of $J(\theta, \mu)$ or $J^*(\mu)$.

**Lemma C.2** (Monotonicity). *If the learning rate satisfies $0 < \eta \le \frac{1}{H^2R^*5} \le \frac{1}{H^2R^*\left(2-\frac{1}{|\mathcal{A}|}\right)} = \frac{1}{\beta}$*

*then $V^{\pi^{\theta_{n+1}}}(s_h) \ge V^{\pi^{\theta_n}}(s_h)$ and $Q^{\pi^{\theta_{n+1}}}(s_h, a) \ge Q^{\pi^{\theta_n}}(s_h, a)$ for all $s_h \in \mathcal{S}^{[\mathcal{H}]}$ and all $a \in \mathcal{A}$. Furthermore, there exist limits $V^\infty(s)$ and $Q^\infty(s, a)$ such that*

$$\lim_{n\to\infty} V^{\pi^{\theta_n}}(s) = V^\infty(s) < \infty.$$

$$\lim_{n\to\infty} Q^{\pi^{\theta_n}}(s, a) = Q^\infty(s, a) < \infty.$$

*Proof.* We will show that $V_h^{\pi^{\theta_n}}(s) \le V_h^{\pi^{\theta_{n+1}}}(s)$ for each state $s \in \mathcal{S}$ (in the not enlarged state space) and each epoch $h$. Then by the bounded reward assumption there exists $V_h^\infty(s)$ such that $V_h^{\pi^{\theta_n}}(s) \to V_h^\infty$ for $n \to \infty$. If this holds true we see the mononicity and convergence of the Q-functions from the relation

$$Q_h^{\pi^\theta}(s, a) = r(s, a) + \sum_{s' \in \mathcal{S}} p(s'|s, a)V_{h+1}^{\pi^\theta}(s'),$$

with $V_H \equiv 0$.

In order to show the claim we first see from the performance difference lemma, that

$$V_h^{\pi^{\theta_{n+1}}}(s) - V_h^{\pi^{\theta_n}}(s) = \mathbb{E}_{S_h=s}^{\pi^{\theta_{n+1}}}\left[\sum_{t=h}^{H-1} A_t^{\pi^{\theta_n}}(S_t, A_t)\right]$$

$$= \sum_{s_l \in \mathcal{S}^{[\mathcal{H}]}} \tilde{\rho}_{s,h}^{\pi^{\theta_{n+1}}}(s_l) \sum_{a \in \mathcal{A}} \pi^{\theta_{n+1}}(a|s_l)A^{\pi^{\theta_n}}(s_l, a),$$

where $\tilde{\rho}_{s,h}^{\pi^{\theta_{n+1}}}(s_l) := \sum_{t=h}^{H-1} \mathbb{P}_{S_h=s}^{\pi^{\theta_{n+1}}}(S_t = s_l)$ the state visitation measure from epoch $h$ to $H-1$ on the enlarged state space $\mathcal{S}^{[\mathcal{H}]}$. Note that $\tilde{\rho}_{s,h}^{\pi^{\theta_{n+1}}}(s_l) = 0$ for $l < h$, as we cannot visit states from previous epochs.

We will prove that $\sum_{a\in\mathcal{A}} \pi^{\theta_{n+1}}(a|s_h)A^{\pi^{\theta_n}}(s_h, a) \ge \sum_{a\in\mathcal{A}} \pi^{\theta_n}(a|s_h)A^{\pi^{\theta_n}}(s_h, a)$, for any $s_h \in \mathcal{S}^{[\mathcal{H}]}$. Then the fact that $\sum_{a\in\mathcal{A}} \pi^{\theta_n}(a|s)A^{\pi^{\theta_n}}(s, a) = 0$ leads to the desired result.

Therefore, we consider the function

$$F_{s_h}(\theta^{s_h}) := \sum_{a \in \mathcal{A}} \pi^{\theta^{s_h}}(a|s_h)c(s_h, a) \quad s_h \in \mathcal{S}^{[\mathcal{H}]},$$

for $\theta^{s_h} = (\theta(s_h, a))_{a\in\mathcal{A}} \in \mathbb{R}^{|\mathcal{A}|}$. We will set $c(s_h, a) = A_h^{\pi^{\theta_n}}(s_h, a)$, but for $\theta_n$ fix, i.e. the following derivatives with respect to $\theta^{s_h}$ of $F$ are independent of $A^{\pi^{\theta_n}}$. From Agarwal Lemma C.2 we know that

$$\frac{\partial F_{s_h}(\theta^{s_h})}{\partial \theta(s_h, a)}\bigg|_{\theta_n^{s_h}} = \pi^{\theta_{s_h}^{(n)}}(a|s_h)A_h^{\pi^{\theta_n}}(s_h, a). \tag{30}$$

Furthermore, $F_{s_h}(\theta_{s_h})$ is $5HR^*$-smooth for every $s_h$ by Lemma D.1 in Agarwal and the bounded reward assumption. Considering our gradient ascent updates from simultaneous training we get

$$\theta_{n+1}(s_h, a) = \theta_n(s_h, a) + \eta \frac{\partial V^{\pi^{\theta_n}}(\mu)}{\partial \theta_n(s_h, a)} \tag{31}$$

$$= \theta_n(s_h, a) + \eta \tilde{\rho}_\mu^{\pi^{\theta_n}}(s_h)\pi^{\theta_n}(a|s_h)A^{\pi^{\theta_n}}(s_h, a) \tag{32}$$

$$= \theta_n(s_h, a) + \eta \tilde{\rho}_\mu^{\pi^{\theta_n}}(s_h)\frac{\partial F_{s_h}(\theta_{s_h})}{\partial \theta_n(s_h, a)}\bigg|_{\theta_n^{s_h}}. \tag{33}$$

As $\eta\tilde{\rho}_\mu^{\pi^{\theta_n}}(s_h) = \eta H d_\mu^{\pi^{\theta_n}}(s_h)$ and $d_\mu^{\pi^{\theta_n}}(s_h)$ a probability measure we see that $\eta\tilde{\rho}_\mu^{\pi^{\theta_n}}(s_h) \leq \frac{1}{5HR^*}$ by our choice of $\eta \leq \frac{1}{H^2R^*5}$. Then the descent lemma for the $5HR^*$-smooth function $F_{s_h}$ gives the desired inequality

$$\sum_{a\in\mathcal{A}}\pi^{\theta_{n+1}}(a|s_h)A^{\pi^{\theta_n}}(s_h,a) \geq \sum_{a\in\mathcal{A}}\pi^{\theta_n}(a|s_h)A^{\pi^{\theta_n}}(s_h,a).$$

$\square$

*Remark* C.3. We want to point out that the proof of Lemma C.2 is crucial for the choice of the step size in the convergence analysis of the simultaneous PG algorithm. As we can only use the descent lemma for a step size $0 < \eta \leq \frac{1}{5H^2R^*}$, we can only achieve asymptotic convergence towards global minima under this assumption. Hence, we also need this step size requirement in the convergence analysis.

We introduce the following definitions:

$$\Delta = \min_{\{(s_h,a)\in(\mathcal{S}H)\times\mathcal{A}\,:\,A^\infty(s_h,a)\neq 0\}} |A^\infty(s_h,a)|$$

where $A^\infty(s_h,a) = Q^\infty(s_h,a) - V^\infty(s_h)$.

We define the sets for each $s_h \in \mathcal{S}^{[\mathcal{H}]}$:

$$\begin{aligned}
I_0^{s_h} &= \{a \in \mathcal{A} \,|\, Q^\infty(s_h,a) = V^\infty(s_h)\}, \\
I_+^{s_h} &= \{a \in \mathcal{A} \,|\, Q^\infty(s_h,a) > V^\infty(s_h)\}, \\
I_-^{s_h} &= \{a \in \mathcal{A} \,|\, Q^\infty(s_h,a) < V^\infty(s_h)\}.
\end{aligned}$$

We aim to prove that $I_+^{s_h}$ is an empty set, then $V^\infty(s_h) = V^*(s_h)$ the optimal value function (epoch wise true).

**Lemma C.4.** *There exists a time $N_1 > 0$ such that for all $n > N_1$, and $s_h \in \mathcal{S}^{[\mathcal{H}]}$, we have*

$$A^{\theta_n}(s_h,a) < -\frac{\Delta}{4} \text{ for } a \in I_-^{s_h}; \quad A^{\theta_n}(s_h,a) > \frac{\Delta}{4} \text{ for } a \in I_+^{s_h}.$$

*Proof.* Fix $s_h \in \mathcal{S}^{[\mathcal{H}]}$ arbitrarily. As $V^{\pi^{\theta_n}}(s_h) \to V^\infty(s_h)$ for $n \to \infty$ and $\mathcal{S}$ is finite, we have that there exists $N_1 > 0$ such that for all $n > N_1$ and $s_h \in \mathcal{S}^{[\mathcal{H}]}$,

$$V^{\pi^{\theta_n}}(s_h) > V^\infty(s_h) - \frac{\Delta}{4}.$$

It follows for all $n > N_1$, $s_h \in \mathcal{S}^{[\mathcal{H}]}$ and $a \in I_-^{s_h}$ by the definition of $\Delta$:

$$A^{\theta_n}(s_h,a) = Q^{\theta_n}(s_h,a) - V^{\pi^{\theta_n}}(s_h) \leq Q^\infty(s_h,a) - V^\infty(s_h) + \frac{\Delta}{4} \leq -\Delta + \frac{\Delta}{4} < -\frac{\Delta}{4}.$$

Similarly, for all $n > N_1$, $s_h \in \mathcal{S}^{[\mathcal{H}]}$ and $a \in I_+^{s_h}$ we obtain from monotonicity Lemma C.2 and the definition of $\Delta$,

$$A_h^{\theta_n}(s,a) = Q^{\theta_n}(s_h,a) - V^{\pi^{\theta_n}}(s_h) \geq Q^\infty(s,a) - \frac{\Delta}{4} - V^\infty(s_h) \geq \Delta - \frac{\Delta}{4} > \frac{\Delta}{4}.$$

$\square$

**Lemma C.5.** *It holds that $\frac{\partial J(\theta_n)}{\partial \theta_n(s_h,a)} \to 0$ as $n \to \infty$ for all $s_h \in \mathcal{S}^{[\mathcal{H}]}$, $a \in \mathcal{A}_s$. This implies that for $a \in I_+^{s_h} \cup I_-^{s_h}$, $\pi^{\theta_n}(a|s_h) \to 0$ and that $\sum_{a\in I_0^{s_h}} \pi^{\theta_n}(a|s_h) \to 1$ for $n \to \infty$.*

*Proof.* From (Beck, 2017, Theorem 10.15) we deduce for any $\beta$-smooth function $f : \mathbb{R}^d \to \mathbb{R}$, that $\|\nabla f(x^k)\| \to 0$ for $k \to \infty$, if $x^{k+1} = x^k - \eta\nabla f(x^k)$, when $\eta < \frac{1}{\beta}$. By Lemma B.1 $J(\cdot)$ is $H^2R^*(2 - \frac{1}{|\mathcal{A}|})$-smooth. It follows by our choice of $\eta < \frac{1}{5H^2R^*}$ that $\frac{\partial J(\theta_n)}{\partial\theta_n(s_h,a)} \to 0$ as $n \to \infty$

for all $s_h \in \mathcal{S}^{[\mathcal{H}]}$, $a \in \mathcal{A}_s$. Now remember the derivative of the softmax parametrisation in the stationary case

$$\frac{\partial J(\theta_n)}{\partial \theta_n(s_h, a)} = \tilde{\rho}_\mu^{\pi^{\theta_n}}(s_h) \pi^{\theta_n}(a|s_h) A^{\theta_n}(s_h, a),$$

and by Lemma C.4 $|A^{\theta_n}(s_h, a)| > \frac{\Delta}{4}$ for all $n > N_1$ and $a \in I_+^{s_h} \cup I_-^{s_h}$. As $\tilde{\rho}_\mu^{\pi^{\theta_n}}(s_h) > 0$ by assumption on $\mu$ and the positivity of the softmax parametrisation. It follows that $\pi^{\theta_n}(a|s) \to 0$ for $n \to \infty$ for all $a \in I_+^{s_h} \cup I_-^{s_h}$ from $\frac{\partial J(\theta_n)}{\partial \theta_n(s_h, a)} \to 0$ as $n \to \infty$.

The last claim, $\sum_{a \in I_0^s} \pi^{\theta_n}(a|s_h) \to 1$ for $n \to \infty$, follows immediately from $\sum_{a \in \mathcal{A}_s} \pi^{\theta_n}(a|s_h) = 1$ by:

$$\lim_{n \to \infty} \sum_{a \in I_0^{s_h}} \pi^{\theta_n}(a|s_h) = \lim_{n \to \infty} \left( \sum_{a \in \mathcal{A}} \pi^{\theta_n}(a|s_h) - \sum_{a \in I_+^{s_h} \cup I_-^{s_h}} \pi^{\theta_n}(a|s_h) \right)$$

$$= 1 - \sum_{a \in I_+^{s_h} \cup I_-^{s_h}} \lim_{n \to \infty} \pi^{\theta_n}(a|s_h)$$

$$= 1.$$

$\square$

**Lemma C.6.** *For $a \in I_+^{s_h}$, the sequence $(\theta_n(s_h, a))_{n \geq 0}$ is strictly increasing for $n > N_1$ and for $a \in I_-^{s_h}$, the sequence $(\theta_n(s_h, a))_{n \geq 0}$ is strictly decreasing for $n > N_1$.*

*Proof.* With Lemma C.4 we know that for $n > N_1$

$$A_h^{\theta_n}(s_h, a) > 0 \text{ for } a \in I_+^{s_h}; \quad A_h^{\theta_n}(s_h, a) < 0 \text{ for } a \in I_-^{s_h},$$

and by the derivative of the value function

$$\frac{\partial J(\theta_n)}{\partial \theta_n(s_h, a)} = \tilde{\rho}_\mu^{\pi^{\theta_n}}(s_h) \pi^{\theta_n}(a|s_h) A_h^{\theta_n}(s_h, a).$$

As $\tilde{\rho}_\mu^{\pi^{\theta_n}}(s_h) > 0$ by the assumption $\mu(s) > 0$ and the positivity of the softmax parametrisation, we have for all $n > N_1$

$$\frac{\partial J(\theta_n)}{\partial \theta_n(s_h, a)} > 0 \text{ for } a \in I_+^{s_h}; \quad \frac{\partial J(\theta_n)}{\partial \theta_n(s_h, a)} < 0 \text{ for } a \in I_-^{s_h}.$$

This implies for $a \in I_+^{s_h}$,

$$\theta_{n+1}(s_h, a) - \theta_n(s_h, a) = \eta \frac{\partial J(\theta_n)}{\partial \theta(s_h, a)} > 0,$$

i.e. $(\theta_n(s_h, a))_{n \geq 0}$ is strictly increasing for $n > N_1$ and similar for $a \in I_-^{s_h}$,

$$\theta_{n+1}(s_h, a) - \theta_n(s_h, a) = \eta \frac{\partial J(\theta_n)}{\partial \theta_n(s_h, a)} < 0,$$

i.e. $(\theta_n(s_h, a))_{n \geq 0}$ is strictly decreasing for $n > N_1$. $\square$

**Lemma C.7.** *For all $s_h \in \mathcal{S}^{[\mathcal{H}]}$ where $I_+^{s_h} \neq \emptyset$, we have that*

$$\max_{a \in I_0^{s_h}} \theta_n(s_h, a) \to \infty \quad \text{and} \quad \min_{a \in \mathcal{A}} \theta_n(s_h, a) \to -\infty \quad \text{for } n \to \infty.$$

*Proof.* By assumption $I_+^{s_h} \neq \emptyset$ there exists an $a_+ \in I_+^{s_h}$ and by Lemma C.5 we have $\pi^{\theta_n}(a_+|s_h) \to 0$, as $n \to \infty$. Hence, by softmax parametrisation this is equivalent to

$$\frac{\exp(\theta_n(s_h, a_+))}{\sum_{a \in \mathcal{A}} \exp(\theta_n(s_h, a))} \to 0, \text{ for } n \to \infty.$$

Using Lemma C.6, i.e. $\theta_n(s_h, a_+)$ is strictly increasing for $n > N_1$, we imply that $\exp(\theta_n(s_h, a_+))$ is strictly increasing for $n > N_1$. This implies that

$$\sum_{a \in \mathcal{A}} \exp(\theta_n(s_h, a)) \to \infty, \text{ for } n \to \infty.$$

Again by Lemma C.5 we know that

$$\sum_{a \in I_0^{s_h}} \pi^{\theta_n}(a|s_h) \to 1, \text{ for } n \to \infty,$$

i.e. by definition

$$\sum_{a \in I_0^{s_h}} \frac{\exp(\theta_n(s_h, a))}{\sum_{a' \in \mathcal{A}} \exp(\theta_n(s_h, a'))} \to 1, \text{ for } n \to \infty.$$

As $\sum_{a' \in \mathcal{A}} \exp(\theta_n(s_h, a')) \to \infty$ it follows that

$$\sum_{a \in I_0^{s_h}} \exp(\theta_n(s_h, a)) \to \infty, \text{ for } n \to \infty$$

implying

$$\max_{a \in I_0^{s_h}} \theta_n(s_h, a) \to \infty, \text{ for } n \to \infty.$$

For the second claim it holds that

$$\begin{aligned}
\sum_{a \in \mathcal{A}} \frac{\partial J(\theta_n)}{\partial \theta_n(s_h, a)} &= \tilde{\rho}_\mu^{\pi^{\theta_n}}(s_h) \sum_{a \in \mathcal{A}} \pi^{\theta_n}(a|s_h)(Q_h^{\pi^{\theta_n}}(s_h, a) - V_h^{\pi^{\theta_n}}(s_h)) \\
&= \tilde{\rho}_\mu^{\pi^{\theta_n}}(s_h)(\mathbb{E}_{S_h=s}^{\pi^{\theta_n}}[Q_h^{\pi^{\theta_n}}(s_h, a)] - V_h^{\pi^{\theta_n}}(s_h)) \\
&= \tilde{\rho}_\mu^{\pi^{\theta_n}}(s_h)(V_h^{\pi^{\theta_n}}(s_h) - V_h^{\pi^{\theta_n}}(s_h)) \\
&= 0.
\end{aligned}$$

By induction, we obtain $\sum_{a \in \mathcal{A}} \theta_n(s_h, a) = \sum_{a \in \mathcal{A}} \theta_0(s_h, a) := c$ for every $n > 0$ and hence

$$\min_{a \in \mathcal{A}} \theta_n(s_h, a) < \sum_{a \in \mathcal{A}} \theta_n(s_h, a) - \max_{a \in \mathcal{A}} \theta_n(s_h, a) = -\max_{a \in \mathcal{A}} \theta_n(s_h, a) + c.$$

Since $\max_{a \in \mathcal{A}} \theta_n(s_h, a) \to \infty$, because $\max_{a \in I_0^{s_h}} \theta_n(s_h, a) \to \infty$, we conclude $\min_{a \in \mathcal{A}} \theta_n(s_h, a) \to -\infty$ for $n \to \infty$. $\qquad\square$

**Lemma C.8.** *Suppose $a_+ \in I_+^{s_h}$. If there exists $a \in I_0^{s_h}$ such that for some $n > N_1$, $\pi^{\theta_n}(a|s_h) \leq \pi^{\theta_n}(a_+|s_h)$, then for all $m > n$ it holds that $\pi^{\theta_m}(a|s_h) \leq \pi^{\theta_m}(a_+|s_h)$.*

*Proof.* Suppose there exists $a \in I_0^s$ such that for an $n > 0$, $\pi^{\theta_n}(a|s_h) \leq \pi^{\theta_n}(a_+|s_h)$. We show that $\pi^{\theta_{n+1}}(a|s_h) \leq \pi^{\theta_{n+1}}(a_+|s_h)$, then the claim follows by induction. We have

$$\begin{aligned}
\frac{\partial J_h(\theta_n)}{\partial \theta_n(s_h, a)} &= \tilde{\rho}_\mu^{\pi^{\theta_n}}(s_h)\pi^{\theta_n}(a|s_h)(Q_h^{\pi^{\theta_n}}(s_h, a) - V_h^{\pi^{\theta_n}}(s_h)) \\
&\leq \tilde{\rho}_\mu^{\pi^{\theta_n}}(s_h)\pi^{\theta_n}(a_+|s_h)(Q_h^{\pi^{\theta_n}}(s_h, a_+) - V_h^{\pi^{\theta_n}}(s_h)) \\
&= \frac{\partial J(\theta_n)}{\partial \theta_n(s_h, a_+)},
\end{aligned}$$

where the inequality follows with

$$\begin{aligned}
Q_h^{\pi^{\theta_n}}(s_h, a_+) &\geq Q_h^\infty(s_h, a_+) - \frac{\Delta}{4} \\
&\geq Q_h^\infty(s_h, a) + \Delta - \frac{\Delta}{4} \\
&> Q_h^{\pi^{\theta_n}}(s_h, a).
\end{aligned}$$

The first inequaility is due to Lemma C.4 and the second by the definition of $\Delta$ and $a \in I_0^{s_h}$. Now by assumption we have $\pi^{\theta_n}(a|s_h) \leq \pi^{\theta_n}(a_+|s_h)$ and thus $\theta_n(s_h, a) \leq \theta_n(s_h, a_+)$. It follows

$$\theta_{n+1}(s_h, a) = \theta_n(s_h, a) + \eta \frac{\partial J(\theta_n)}{\partial \theta_n(s_h, a)} \leq \theta(s_h, a_+) + \eta \frac{\partial J(\theta_n)}{\partial \theta_n(s_h, a_+)} = \theta_{n+1}(s_h, a_+).$$

$\square$

Now define for every $a_+ \in I_+^{s_h}$ the set

$$B_0^{s_h}(a_+) = \{a \in I_0^{s_h} | \pi^{\theta_n}(a_+|s_h) \leq \pi^{\theta_n}(a|s_h) \text{ for all } l > 0\}$$

and denote its complement in $I_0^{s_h}$ as $\bar{B}_0^{s_h}(a_+) = I_0^{s_h} \setminus B_0^{s_h}(a_+)$.

**Lemma C.9.** *Suppose $I_+^{s_h} \neq \emptyset$. For all $a_+ \in I_+^{s_h}$, we have that $B_0^{s_h}(a_+) \neq \emptyset$ and*

$$\sum_{a \in B_0^{s_h}(a_+)} \pi^{\theta_n}(a|s_h) \to 1, \ as \ n \to \infty.$$

*This implies:*

$$\max_{a \in B_0^{s_h}(a_+)} \theta_n(s_h, a) \to \infty, \ for \ n \to \infty.$$

*Proof.* Let $a_+ \in I_+^{s_h}$ and consider $a \in \bar{B}_0^{s_h}(a_+)$. Then by definition of $\bar{B}_0^{s_h}(a_+)$ there exists $n' > N_1$ such that $\pi^{\theta_{n'}}(a_+|s_h) \geq \pi^{\theta_{n'}}(a|s_h)$. Hence, by Lemma C.8 for all $n \geq n'$ we have $\pi^{\theta_n}(a_+|s_h) \geq \pi^{\theta_n}(a|s_h)$. As $\pi^{\theta_n}(a_+|s_h) \to 0$ for $n \to \infty$. We obtain $\pi^{\theta_n}(a|s_h) \to 0$ for $n \to \infty$, for all $a \in \bar{B}_0^{s_h}(a_+)$. Since by Lemma C.5 $\sum_{a \in I_0^{s_h}} \pi^{\theta_n}(a|s_h) \to 1$ for $n \to \infty$, we have that $B_0^{s_h}(a_+) \neq \emptyset$ and that $\sum_{a \in B_0^{s_h}(a_+)} \pi^{\theta_n}(a|s_h) \to 1$, as $n \to \infty$. The second claim follows from this as in Lemma C.7. $\square$

**Lemma C.10.** *Consider $s_h \in \mathcal{S} \times \mathcal{H}$ such that $I_+^{s_h} \neq \emptyset$. Then, for any $a_+ \in I_+^{s_h}$, there exists an $N_{a_+}$ such that for all $n > N_{a_+}$ we have*

$$\pi^{\theta_n}(a_+|s_h) > \pi^{\theta_n}(a|s_h) \text{ for all } a \in \bar{B}_0^{s_h}(a_+).$$

*Proof.* For every $a \in \bar{B}_0^{s_h}(a_+)$ exists time $n_a$ such that

$$\pi^{\theta_n}(a_+|s_h) > \pi^{\theta_n}(a|s_h) \text{ for all } a \in \bar{B}_0^{s_h}(a_+)$$

for all $n > n_a$ by definition. Set $N_{a_+} = \max_{a \in \bar{B}_0^{s_h}(a_+)} n_a$ and the proof is completed. $\square$

**Lemma C.11.** *Assume again $I_+^{s_h} \neq \emptyset$. For all actions $a \in I_+^{s_h}$, we have that $\theta_n(s_h, a)$ is bounded from below as $n \to \infty$. And for all $a \in I_-^{s_h}$, we have that $\theta_n(s_h, a) \to -\infty$ as $n \to \infty$.*

*Proof.* The first claim follows directly with Lemma C.6 as $\theta_n(s_h, a)$ is strictly increasing for all $a \in I_+^{s_h}$, $n > N_1$, and thus for all $n > N_1$ we have $\theta_n(s_h, a) \geq \theta_{N_1}(s_h, a)$. Now suppose $a \in I_-^{s_h}$, then by Lemma C.6 we have that $\theta_n(s_h, a)$ is strictly decreasing for $n > N_1$. Assume there exists $b$ such that $\lim_{n \to \infty} \theta_n(s_h, a) = b$, then $\theta_n(s_h, a) > b$ for all $n > N_1$. By Lemma C.7 there exists an action $a' \in \mathcal{A}$ such that $\theta_n(s_h, a') \to -\infty$ for $n \to \infty$. Consider $\delta > 0$ such that $\theta_{N_1}(s_h, a') \geq b - \delta$. Define for all $n > N_1$

$$\tau(n) = \max\{k \in (N_1, n] : \theta_k(s_h, a') \geq b - \delta\}.$$

Define also

$$\mathcal{T}^{(n)} = \left\{\tau(n) < n' < n : \frac{\partial J(\theta_{n'})}{\partial \theta_{n'}(s_h, a')} \leq 0\right\},$$

as the set of all indices $n'$ in $(\tau(n), n)$, where $\theta_{n'}(s_h, a')$ is decreasing. Next we define $Z_n := \sum_{n' \in \mathcal{T}^{(n)}} \frac{\partial J(\theta_{n'})}{\partial \theta_{n'}(s_h, a')}$, then it holds that

$$
\begin{aligned}
Z_n &= \sum_{n' \in \mathcal{T}^{(n)}} \frac{\partial J(\theta_{n'})}{\partial \theta_{n'}(s_h, a')} \\
&\leq \sum_{n'=\tau(n)+1}^{n-1} \frac{\partial J(\theta_{n'})}{\partial \theta_{n'}(s_h, a')} \\
&\leq \sum_{n'=\tau(n)}^{n-1} \frac{\partial J(\theta_{n'})}{\partial \theta_{n'}(s_h, a')} + \left| \frac{\partial J(\theta_{\tau(n)})}{\partial \theta_{\tau(n)}(s_h, a')} \right|.
\end{aligned}
$$

By Lemma A.7 and the bounded reward assumption we have

$$
\left| \frac{\partial J(\theta_{\tau(n)})}{\partial \theta_{\tau(n)}(s_h, a')} \right| = \tilde{\rho}_\mu^{\pi^{\theta_{\tau(n)}}}(s_h) \pi^{\theta_{\tau(n)}}(a'|s_h) |A_h^{\theta_{\tau(n)}}(s_h, a')| \leq H^2 R^*.
$$

Hence,

$$
\begin{aligned}
Z_n &\leq \sum_{n'=\tau(n)}^{n-1} \frac{\partial J(\theta_{n'})}{\partial \theta_{n'}(s_h, a')} + H^2 R^* \\
&= \frac{1}{\eta}(\theta_n(s_h, a') - \theta_{\tau(n)}(s_h, a')) + H^2 R^* \\
&\leq \frac{1}{\eta}(\theta_n(s_h, a') - b + \delta) + H^2 R^*.
\end{aligned}
$$

Then $\theta_n(s_h, a') \to -\infty$ for $n \to \infty$ implies that $Z_n \to -\infty$ for $n \to \infty$. As we chose $a \in I_-^{s_h}$ it holds that $|A_h^{\theta_n}(s_h, a)| \geq \frac{\Delta}{4}$ for $n > N_1$ with Lemma C.4 and so for all $n' \in \mathcal{T}^{(n)}$:

$$
\begin{aligned}
\left| \frac{\frac{\partial J(\theta_{n'})}{\partial \theta_{n'}(s_h, a)}}{\frac{\partial J(\theta_{n'})}{\partial \theta_{n'}(s_h, a')}} \right| &= \left| \frac{\pi^{\theta_{n'}}(a|s_h) A_h^{\theta_{n'}}(s_h, a)}{\pi^{\theta_{n'}}(a'|s_h) A_h^{\theta_{n'}}(s_h, a')} \right| \\
&\geq \frac{\pi^{\theta_{n'}}(a|s_h)}{\pi^{\theta_{n'}}(a'|s_h)} \frac{\Delta}{4HR^*} \\
&= \exp(\theta_{n'}(s_h, a) - \theta_{n'}(s_h, a')) \frac{\Delta}{4HR^*} \\
&\geq \exp(b - (b - \delta)) \frac{\Delta}{4HR^*} \\
&= \exp(\delta) \frac{\Delta}{4HR^*},
\end{aligned}
$$

where we used in the last inequality that $\theta_{n'}(s_h, a') \leq b - \delta$ for all $n' > \tau(n)$ and $\theta_{n'}(s_h, a) > b$ for all $n' > N_1$. By the definition of $\mathcal{T}^{(n)}$ these inequalities holds especially for all $n' \in \mathcal{T}^{(n)}$. Using this we can imply that for all $n > N_1$ with $\mathcal{T}^{(n)} \neq \emptyset$,

$$
\begin{aligned}
\frac{1}{\eta} \left( \theta_{N_1}(s_h, a) - \theta_n(s_h, a) \right) &= \sum_{n'=N_1+1}^{n-1} \frac{\partial J(\theta_{n'})}{\partial \theta_{n'}(s_h, a)} \\
&\leq \sum_{n' \in \mathcal{T}^{(n)}} \frac{\partial J(\theta_{n'})}{\partial \theta_{n'}(s_h, a)} \\
&\leq \exp(\delta) \frac{\Delta}{4HR^*} \sum_{n' \in \mathcal{T}^{(n)}} \frac{\partial J(\theta_{n'})}{\partial \theta_{n'}(s_h, a')} \\
&= \exp(\delta) \frac{\Delta}{4HR^*} Z_n,
\end{aligned}
$$

where the first inequality holds because $\theta_{n'}(s_h, a)$ is strictly decreasing for $n' > N_1$, i.e. $\frac{\partial J(\theta_{n'})}{\partial \theta_{n'}(s_h, a)} < 0$ for all $n' \in \{N_1 + 1, \ldots, n-1\}$. In the second inequality we used

$$\left| \frac{\frac{\partial J(\theta_{n'})}{\partial \theta_{n'}(s_h, a)}}{\frac{\partial J(\theta_{n'})}{\partial \theta_{n'}(s_h, a')}} \right| \geq \exp(\delta) \frac{\Delta}{4HR^*}.$$

Note that $\frac{\partial J(\theta_{n'})}{\partial \theta_{n'}(s_h, a)} < 0$ and $\frac{\partial J(\theta_{n'})}{\partial \theta_{n'}(s_h, a')} < 0$ for $n' \in \mathcal{T}^{(n)}$ so that the sign of the inequality reverses. Finally, we deduce from $Z_n \to -\infty$ that $\theta_n(s_h, a) \to \infty$ for $n \to \infty$, which is a contradiction to $\theta_n(s_h, a)$ strictly decreasing for all $n > N_1$. $\qquad\square$

**Lemma C.12.** *Consider* $s \in \mathcal{S}^{[\mathcal{H}]}$ *such that* $I_+^{s_h} \neq \emptyset$. *Then for any* $a_+ \in I_+^{s_h}$ *it holds that*

$$\sum_{a \in B_0^{s_h}(a_+)} \theta_n(s_h, a) \to \infty, \quad \text{for } n \to \infty.$$

*Proof.* Let $a_+ \in I_+^{s_h}$ and $a \in B_0^{s_h}(a_+)$. Then by definition of $B_0^{s_h}(a_+)$ we have

$$\pi^{\theta_n}(a_+|s_h) \leq \pi^{\theta_n}(a|s_h)$$

for all $n > 0$ and hence by softmax parametrisation $\theta_n(s_h, a_+) \leq \theta_n(s_h, a)$ for all $n > 0$. By Lemma C.11 we have that $\theta_n(s_h, a_+)$ and thus also $\theta_n(s_h, a)$ is bounded from below for $n \to \infty$. Together with

$$\max_{\{a \in B_0^{s_h}(a_+)\}} \theta_n(s_h, a) \to \infty, \quad \text{for } n \to \infty$$

by Lemma C.9 we deduce the claim. $\qquad\square$

Finally, we are ready to prove the asymptotic convergence of simultaneous PG with tabular softmax parametrisation.

*Proof of Theorem C.1.* We have to show that $I_+^{s_h} = \emptyset$ for all $s_h \in \mathcal{S}^{[\mathcal{H}]}$. So assume there exists $s_h \in \mathcal{S}^{[\mathcal{H}]}$ such that $I_+^{s_h} \neq \emptyset$ and let $a_+ \in I_+^{s_h}$. Then by Lemma C.12 we have

$$\sum_{a \in B_0^{s_h}(a_+)} \theta_n(s_h, a) \to \infty, \quad \text{for } n \to \infty. \tag{34}$$

For any $a \in I_-^{s_h}$ we have by Lemma C.11 that

$$\frac{\pi^{\theta_n}(a|s_h)}{\pi^{\theta_n}(a_+|s_h)} = \exp(\underbrace{\theta_n(s_h, a)}_{\to -\infty} - \underbrace{\theta_n(s_h, a_+)}_{\text{bounded from below}}) \to 0, \quad n \to \infty.$$

Hence, there exists $N_2 > N_1$ such that for all $n > N_2$

$$\frac{\pi^{\theta_n}(a|s_h)}{\pi^{\theta_n}(a_+|s_h)} < \frac{\Delta}{16|\mathcal{A}|HR^*},$$

which leads for $n > N_2$ to

$$-HR^* \sum_{a \in I_-^{s_h}} \pi^{\theta_n}(a|s_h) > -\frac{\Delta}{16} \pi^{\theta_n}(a_+|s_h). \tag{35}$$

Note that if $I_-^{s_h} = \emptyset$ we can just ignore this sum later on.

Next consider $a \in \bar{B}_0^{s_h}(a_+) \subseteq I_0^{s_h}$. By the definition of $I_0^{s_h}$ we have that $A_h^{\theta_n}(s_h, a) \to A_h^\infty(s_h, a) = 0$ for $n \to \infty$. By Lemma C.10 we have for $n \geq N_{a_+}$

$$1 < \frac{\pi^{\theta_n}(a_+|s_h)}{\pi^{\theta_n}(a|s_h)}.$$

Thus, there exists $N_3 > \max\{N_2, N_{a_+}\}$ such that for all $n \geq N_3$

$$|A_h^{\theta_n}(s_h, a)| < \frac{\pi^{\theta_n}(a_+|s_h)}{\pi^{\theta_n}(a|s_h)} \frac{\Delta}{16|\mathcal{A}|}.$$

This implies

$$\sum_{a \in \bar{B}_0^{s_h}(a_+)} \pi^{\theta_n}(a|s_h)|A_h^{\theta_n}(s_h, a)| < \pi^{\theta_n}(a_+|s_h)\frac{\Delta}{16}$$

and so

$$-\pi^{\theta_n}(a_+|s_h)\frac{\Delta}{16} < \sum_{a \in \bar{B}_0^{s_h}(a_+)} \pi^{\theta_n}(a|s)A_h^{\theta_n}(s_h, a) < \pi^{\theta_n}(a_+|s_h)\frac{\Delta}{16}, \quad (36)$$

for all $n > N_3$. We can conclude again for $n > N_3$,

$$0 = \sum_{a \in \mathcal{A}} \pi^{\theta_n}(a|s_h)A_h^{\theta_n}(s_h, a)$$

$$= \sum_{a \in B_0^{s_h}(a_+)} \pi^{\theta_n}(a|s_h)A_h^{\theta_n}(s_h, a) + \sum_{a \in \bar{B}_0^{s_h}(a_+)} \pi^{\theta_n}(a|s_h)A_h^{\theta_n}(s_h, a)$$

$$+ \sum_{a \in I_+^{s_h}} \pi^{\theta_n}(a|s_h)A_h^{\theta_n}(s_h, a) + \sum_{a \in I_-^{s_h}} \pi^{\theta_n}(a|s_h)A_h^{\theta_n}(s_h, a)$$

$$> \sum_{a \in B_0^{s_h}(a_+)} \pi^{\theta_n}(a|s_h)A_h^{\theta_n}(s_h, a) - \pi^{\theta_n}(a_+|s_h)\frac{\Delta}{16} + \pi^{\theta_n}(a_+|s_h)\frac{\Delta}{4} - HR^* \sum_{a \in I_-^{s_h}} \pi^{\theta_n}(a|s_h)$$

$$\geq \sum_{a \in B_0^{s_h}(a_+)} \pi^{\theta_n}(a|s_h)A_h^{\theta_n}(s_h, a) - \pi^{\theta_n}(a_+|s_h)\frac{\Delta}{16} + \pi^{\theta_n}(a_+|s)\frac{\Delta}{4} - \frac{\Delta}{16}\pi^{\theta_n}(a_+|s_h)$$

$$> \sum_{a \in B_0^{s_h}(a_+)} \pi^{\theta_n}(a|s_h)A_h^{\theta_n}(s_h, a),$$

where we used Equation Equation 36 and Lemma C.4 in the first inequality and Equation Equation 35 in the second inequality. Finally, by our assumption and Equation Equation 34 for $n > N_3$,

$$\infty \overset{n \to \infty}{\longleftarrow} \sum_{a \in B_0^{s_h}(a_+)} (\theta_n(s, a) - \theta_{N_3}(s_h, a))$$

$$= \eta \sum_{n'=N_3}^{n} \sum_{a \in B_0^{s_h}(a_+)} \frac{\partial J(\theta_{n'})}{\partial \theta_{n'}(s_h, a)}$$

$$= \eta \sum_{n'=N_3}^{n} \tilde{\rho}_\mu^{\pi^{\theta_{n'}}}(s_h) \sum_{a \in B_0^{s_h}(a_+)} \pi^{\theta_{n'}}(a|s_h)A_h^{\theta_{n'}}(s_h, a),$$

which contradicts $\sum_{a \in B_0^s(a_+)} \pi^{\theta_n}(a|s)A_h^{\theta_n}(s, a) < 0$ for all $n > N_3$. $\square$

# D  PROOFS OF SECTION 4

## D.1  SIMULTANEOUS APPROACH

We first proof that the gradient estimator is unbiased and has bounded variance.

**Lemma D.1.** *Consider the estimator from Equation 7. For any $K > 0$ it holds that*

$$\mathbb{E}_\mu^{\pi^\theta}[\widehat{\nabla} J^K(\theta, \mu)] = \nabla J(\theta, \mu)$$

*and*

$$\mathbb{E}_\mu^{\pi^\theta}[\|\widehat{\nabla} J^K(\theta, \mu) - \nabla J(\theta, \mu)\|^2] \leq \frac{3H^4 \max\{R^*, 1\}^4}{K} =: \frac{\xi}{K}$$

*Proof.* By the definition of $\widehat{\nabla} J^K$ we have

$$\mathbb{E}_\mu^{\pi^\theta}[\widehat{\nabla} J^K(\theta, \mu)]$$

$$= \mathbb{E}_\mu^{\pi^\theta}\Big[\frac{1}{K}\sum_{i=1}^K \sum_{h=0}^{H-1} \nabla \log(\pi^\theta(A_h^i|S_h^i))\hat{R}_h^i\Big]$$

$$= \mathbb{E}_\mu^{\pi^\theta}\Big[\sum_{h=0}^{H-1} \nabla \log(\pi^\theta(A_h|S_h))\hat{R}_h\Big]$$

$$= \mathbb{E}_\mu^{\pi^\theta}\Big[\sum_{h=0}^{H-1} \nabla \log(\pi^\theta(A_h|S_h))\sum_{k=h}^{H-1} r(S_k, A_k)\Big],$$

where we used that we consider independent samples for $i = 1, \ldots, K$. From the proof of the policy gradient Theorem, we obtain that

$$\mathbb{E}_\mu^{\pi^\theta}[\widehat{\nabla} J^K(\theta, \mu)] = \nabla J(\theta, \mu).$$

For the second claim we first see that

$$\|\nabla J(\theta, \mu)\| = \Big(\sum_{s\in\mathcal{S}^{[\mathcal{H}]}}\sum_{a\in\mathcal{A}}(Hd_\mu^{\pi^\theta}(s)\pi^\theta(a|s)A^{\pi^\theta}(s,a))^2\Big)^{\frac{1}{2}}$$

$$\leq H^2 R^*\Big(\sum_{s\in\mathcal{S}^{[\mathcal{H}]}}\sum_{a\in\mathcal{A}}(d_\mu^{\pi^\theta}(s)\pi^\theta(a|s))^2\Big)^{\frac{1}{2}}$$

$$\leq H^2(R^*)^2,$$

because $\pi^\theta(\cdot|s) \leq 1$, $d_\mu^{\pi^\theta}(s) \leq 1$ and both are probability distributions.

Next we have that

$$\mathbb{E}_\mu^{\pi^\theta}[\|\widehat{\nabla} J^1(\theta, \mu)\|] \leq \mathbb{E}_\mu^{\pi^\theta}\Big[\sum_{h=0}^{H-1}\|\nabla \log(\pi^\theta(A_h|S_h))\|\|\hat{R}_h\|\Big]$$

$$\leq H^2 R^* \mathbb{E}_\mu^{\pi^\theta}\Big[\|\nabla \log(\pi^\theta(A_h|S_h))\|\Big]$$

$$\leq H^2 R^*,$$

where the last inequality follows with by Yuan et al. (2022, Lem 4.8) and Jensen's inequality.

Thus,

$$\mathbb{E}_\mu^{\pi^\theta}[\|\widehat{\nabla} J^K(\theta, \mu) - \nabla J(\theta, \mu)\|^2]$$

$$\leq \frac{1}{K}\mathbb{E}_\mu^{\pi^\theta}\Big[\|\widehat{\nabla} J^1(\theta, \mu) - \nabla J(\theta, \mu)\|^2\Big]$$

$$\leq \frac{1}{K}\mathbb{E}_\mu^{\pi^\theta}\Big[\|\widehat{\nabla} J^1(\theta)\|^2 + 2\|\widehat{\nabla} J^1(\theta, \mu)\|\|\nabla J(\theta)\| + \|\nabla J(\theta, \mu)\|^2\Big]$$

$$\leq \frac{1}{K}\Big[H^4(R^*)^2 + H^4(R^*)^2 + H^4(R^*)^4\Big].$$

Define $\xi = 3H^4 \max\{R^*, 1\}^4 \geq H^4(R^*)^2 + H^4(R^*)^2 + H^4(R^*)^4$ proves the claim. $\square$

Recall the stochastic PG updates for training the softmax parameter from Equation 8

$$\bar{\theta}^{(n+1)} = \bar{\theta}^{(n)} + \eta\widehat{\nabla} J^K(\bar{\theta}^{(n)}, \mu).$$

In the following denote by $(\theta^{(n)})_{n\geq 0}$ the deterministic sequence generated by Algorithm 1 such that the initial parameter agree, $\theta^{(0)} = \bar{\theta}^{(0)}$, and the step size $\eta$ is the same for both processes. The natural filtration of $(\bar{\theta}_h^{(n)})_{n\geq 0}$ is denoted by $(\mathcal{F}^{(n)})_{n\geq 0}$. Recall that for the deterministic scheme we could assure $c = \inf_n \min_{s_h\in\mathcal{S}^{[\mathcal{H}]}} \pi^{\theta_n}(a^*(s_h)|s_h)$ is bounded away from 0 by Lemma B.5.

This cannot be guaranteed for the stochastic trajectory. The idea of the convergence analysis for stochastic softmax PG is now to define the following stopping time

$$\tau := \min\{n \geq 0 : \|\theta^{(n)} - \bar{\theta}^{(n)}\|_2 \geq \frac{c}{4}\}.$$

This means, $\tau$ is the first time when the stochastic process $(\bar{\theta}^{(n)})_{n \geq 0}$ is *too far away* from the PG trajectory $(\theta^{(n)})_{n \geq 0}$. Hence, all challenges encountered in the deterministic case transfer to the stochastic context, indicating that the model dependent constant $c$ naturally appears in the error bounds of the stochastic case. We emphasise that $\tau$ is a stopping time with respect to the filtration $(\mathcal{F}^{(n)})_{n \geq 0}$ by construction.

First, consider the event $\{n \leq \tau\}$, i.e. $\|\theta^{(n)} - \bar{\theta}^{(n)}\|_2 \leq \frac{c_h}{4}$. Then, it follows from the $\sqrt{2}$-Lipschitz continuity of $\theta \mapsto \pi^\theta(a^*(s)|s)$ that $\min_{0 \leq n \leq \tau} \min_{s \in \mathcal{S}} \pi^{\bar{\theta}^{(n)}}(a^*(s)|s) \geq \frac{c}{2} > 0$.

**Lemma D.2.** *The softmax policy $\pi^\theta(a|s)$ is $\sqrt{2}$-Lipschitz with respect to $\theta \in \mathbb{R}^d$ for every $s, a$.*

*Proof.* The derivative of the softmax function is

$$\frac{\partial \pi^\theta(a|s)}{\partial \theta(s', a')} = \mathbf{1}_{s'=s} \left[ \frac{\mathbf{1}_{a'=a} \exp(\theta(s,a)) \left( \sum_{\tilde{a} \in \mathcal{A}_s} \exp(\theta(s, \tilde{a})) \right) - \exp(\theta(s,a)) \exp(\theta(s,a'))}{\left( \sum_{\tilde{a} \in \mathcal{A}_s} \exp(\theta(s, \tilde{a})) \right)^2} \right]$$

$$= \mathbf{1}_{s'=s} \left[ \mathbf{1}_{a'=a} \pi^\theta(a|s) - \pi^\theta(a|s) \pi^\theta(a'|s) \right].$$

Therefore,

$$\|\nabla \pi^\theta(a|s)\|_2 = \sqrt{\sum_{\tilde{a} \in \mathcal{A}_s} \left( \mathbf{1}_{a'=a} \pi^\theta(a|s) - \pi^\theta(a|s) \pi^\theta(a'|s) \right)^2}$$

$$\leq \sqrt{\pi^\theta(a|s)^2 - 2\pi^\theta(a|s)^3 + \sum_{\tilde{a} \in \mathcal{A}_s} \pi^\theta(a'|s)^2 \pi^\theta(a|s)^2}$$

$$\leq \sqrt{2}.$$

$\square$

**Lemma D.3.** *Let $\mu$ be a probability measure such that $\mu(s) > 0$ for all $s \in \mathcal{S}$ and consider the sequence $(\bar{\theta}^{(n)})_{n \geq 0}$ generated by Equation 8. Then, it holds almost surely that $\min_{0 \leq n \leq \tau} \min_{s \in \mathcal{S}_h} \pi^{\bar{\theta}^{(n)}}(a^*(s)|s) \geq \frac{c}{2}$ is strictly positive.*

*Proof.* For every $n \leq \tau$ we obtain by the $\sqrt{2}$-Lipschitz continuity in Lemma D.2 that

$$\pi^{\bar{\theta}^{(n)}}(a^*(s)|s) \geq \pi^{\theta^{(n)}}(a^*(s)|s) - |\pi^{\theta^{(n)}}(a^*(s)|s) - \pi^{\bar{\theta}^{(n)}}(a^*(s)|s)|$$

$$\geq \pi^{\theta^{(n)}}(a^*(s)|s) - \sqrt{2}\|\bar{\theta}^{(n)} - \theta^{(n)}\|_2$$

$$> \frac{c}{2} > 0,$$

holds almost surely. The claim follows directly. $\square$

This allows us to use the weak PL-inequality of Lemma B.2 to derive a convergence rate on the event $\{n \leq \tau\}$ in the following sense:

**Lemma D.4.** *Under Assumption 3.1, let $\mu$ be a probability measure such that $\mu(s) > 0$ for all $s \in \mathcal{S}$ and consider the sequence $(\bar{\theta}^{(n)})_{n \geq 0}$ generated by Equation 8. Suppose that*

- *the batch size $K^{(n)} \geq \frac{9}{8} \frac{c^2 \max\{R^*, 1\}^2 (1 - \frac{1}{2\sqrt{N}})}{N^{3/2} |\mathcal{S}| H^{19}} \left\| \frac{d_\mu^{\pi^*}}{\mu} \right\|_\infty^{-2} n^2$ is increasing for fix $N \geq 1$,*

- *the step size $\eta = \frac{1}{5H^2 R^* \sqrt{N}}$.*

*Then,*

$$\mathbb{E}\big[(J^*(\mu) - J(\bar{\theta}^{(n)}, \mu))\mathbf{1}_{\{n \le \tau\}}\big] \le \frac{20|\mathcal{S}|H^5 R^*}{c^2 \frac{1}{\sqrt{N}}(1 - \frac{1}{2\sqrt{N}})n} \Big\| \frac{d_\mu^{\pi^*}}{\mu} \Big\|_\infty^2.$$

*Proof.* Throughout the proof we drop the $\mu$ in $J$ and $J^*$.

First, we deduce from the $\beta$-smoothness of $J$, as in the proof of Theorem 3.2 that almost surely

$$J(\bar{\theta}^{(n+1)}) \ge J(\bar{\theta}^{(n)}) + \big(\nabla J(\bar{\theta}^{(n)})\big)^T (\bar{\theta}^{(n+1)} - \bar{\theta}^{(n)}) - \frac{\beta}{2}\|\bar{\theta}^{(n+1)} - \bar{\theta}^{(n)}\|^2.$$

We continue with

$$
\begin{aligned}
J(\bar{\theta}^{(n+1)}) &\ge J(\bar{\theta}^{(n)}) + \eta\big(\nabla J(\bar{\theta}^{(n)})\big)^T \widehat{\nabla} J^K(\bar{\theta}^{(n)}) - \frac{\beta\eta^2}{2}\|\widehat{\nabla} J^K(\bar{\theta}^{(n)})\|^2 \\
&= J(\bar{\theta}^{(n)}) + \eta\big(\nabla J(\bar{\theta}^{(n)})\big)^T \nabla J(\bar{\theta}^{(n)}) + \eta\big(\nabla J(\bar{\theta}^{(n)})\big)^T \big(\widehat{\nabla} J^K(\bar{\theta}^{(n)}) - \nabla J(\bar{\theta}^{(n)})\big) \\
&\quad - \frac{\beta\eta^2}{2}\|\big(\widehat{\nabla} J^K(\bar{\theta}^{(n)}) - \nabla J(\bar{\theta}^{(n)})\big) + \nabla J(\bar{\theta}^{(n)})\|^2.
\end{aligned}
$$

Thus,

$$J(\bar{\theta}^{(n+1)}) \ge J(\bar{\theta}^{(n)},) + \big(\eta - \frac{\beta\eta^2}{2}\big)\|\nabla J(\bar{\theta}^{(n)})\|^2 + \big(\eta - \beta\eta^2\big)\langle \nabla J(\bar{\theta}^{(n)}), \phi_n\rangle - \frac{\beta\eta^2}{2}\|\phi_n\|^2,$$

where $\phi_n := \widehat{\nabla} J^K(\bar{\theta}^{(n)}) - \nabla J(\bar{\theta}^{(n)})$. Next we take the conditional expectation on $\mathcal{F}_n$. Then by Lemma D.1 we obtain

$$E\Big[J(\bar{\theta}^{(n+1)})|\mathcal{F}_n\Big] \ge J(\bar{\theta}^{(n)}) + \Big(\eta - \frac{\beta\eta^2}{2}\Big)\|\nabla J(\bar{\theta}^{(n)})\|^2 - \frac{\beta\eta^2\xi}{2K_n}.$$

Subtracting this equation form $J^*$ and taking the expectation under the event $\{n+1 \le \tau\}$ results in:

$$
\begin{aligned}
&\mathbb{E}\Big[(J^* - J(\bar{\theta}^{(n+1)}))\mathbf{1}_{\{n+1 \le \tau\}}\Big] \\
&= \mathbb{E}\Big[\mathbb{E}\Big[(J^* - J(\bar{\theta}^{(n+1)}))|\mathcal{F}_n\Big]\mathbf{1}_{\{n+1 \le \tau\}}\Big] \\
&\le \mathbb{E}\Big[\Big(J^* - \mathbb{E}\Big[J(\bar{\theta}^{(n+1)})|\mathcal{F}_n\Big]\Big)\mathbf{1}_{\{n \le \tau\}}\Big] \\
&\le \mathbb{E}\Big[(J^* - J(\bar{\theta}^{(n)}))\mathbf{1}_{\{n \le \tau\}}\Big] - \Big(\eta - \frac{\beta\eta^2}{2}\Big)\mathbb{E}\Big[\|\nabla J(\bar{\theta}^{(n)})\|^2\mathbf{1}_{\{n \le \tau\}}\Big] + \frac{\beta\eta^2\xi}{2K_n} \\
&\le \mathbb{E}\Big[(J^* - J(\bar{\theta}^{(n)}))\mathbf{1}_{\{n \le \tau\}}\Big] - \eta(1 - \frac{1}{2\sqrt{N}})\mathbb{E}\Big[\|\nabla J(\bar{\theta}^{(n)})\|^2\mathbf{1}_{\{n \le \tau\}}\Big] + \frac{\beta\eta^2\xi}{2K_n},
\end{aligned}
$$

where we used that $\{n+1 \le \tau_h\} = \{\tau_h \le n\}^C$ is $\mathcal{F}_n$-measurable and that $\mathbf{1}_{\{n+1 \le \tau_h\}} \le \mathbf{1}_{\{n \le \tau_h\}}$ a.s. With the PL-type inequality Lemma B.2 and $\min_{0 \le n \le \tau}\min_{s \in \mathcal{S}}\pi^{\bar{\theta}^{(n)}}(a^*(s)|s) \ge \frac{c}{2}$ by Lemma D.3 we have

$$
\begin{aligned}
&\mathbb{E}\Big[(J^* - J(\bar{\theta}^{(n+1)}))\mathbf{1}_{\{n+1 \le \tau\}}\Big] \\
&\le \mathbb{E}\Big[(J^* - J(\bar{\theta}^{(n)}))\mathbf{1}_{\{n \le \tau\}}\Big] - \eta(1 - \frac{1}{2\sqrt{N}})\frac{c^2}{|\mathcal{S}|H}\Big\|\frac{d_\mu^{\pi^*}}{d_\mu^{\pi^\theta}}\Big\|_\infty^{-2}\mathbb{E}\Big[(J^* - J(\bar{\theta}^{(n)}))\mathbf{1}_{\{n \le \tau\}}\Big]^2 + \frac{\beta\eta^2\xi}{2K_n} \\
&\le \mathbb{E}\Big[(J^* - J(\bar{\theta}^{(n)}))\mathbf{1}_{\{n \le \tau\}}\Big] - \eta(1 - \frac{1}{2\sqrt{N}})\frac{c^2}{|\mathcal{S}|H^3}\Big\|\frac{d_\mu^{\pi^*}}{\mu}\Big\|_\infty^{-2}\mathbb{E}\Big[(J^* - J(\bar{\theta}^{(n)}))\mathbf{1}_{\{n \le \tau\}}\Big]^2 + \frac{\beta\eta^2\xi}{2K_n},
\end{aligned}
$$

where we used in the last inequality that under Assumption 3.1 we have $d_\mu^{\pi^\theta}(s) \geq \frac{1}{H}\mu(s)$ (see Remark B.3). For $d_n := \mathbb{E}\left[(J^* - J(\bar{\theta}^{(n)}))\mathbf{1}_{\{n \leq \tau\}}\right]$ we obtain the recursive inequality

$$d_{n+1} \leq d_n - \eta(1 - \frac{1}{2\sqrt{N}})\frac{c^2}{|\mathcal{S}|H^3}\left\|\frac{d_\mu^{\pi^*}}{\mu}\right\|_\infty^{-2}d_n^2 + \frac{\beta\eta^2\xi}{2K_n}.$$

We define $w := \eta(1 - \frac{1}{2\sqrt{N}})\frac{c^2}{|\mathcal{S}|H^3}\left\|\frac{d_\mu^{\pi^*}}{\mu}\right\|_\infty^{-2}$ and $B = \frac{\beta\eta^2\xi}{2} > 0$ such that

$$d_{n+1} \leq d_n(1 - wd_n) + \frac{B}{K_n}.$$

Note that $w > 0$ by the assumption $\mu(s) > 0$ for all $s \in \mathcal{S}$. Then by our choice of $K_n$ it holds that

$$\frac{9}{4}wBn^2 = \frac{9}{8}\frac{c^2\eta^3\beta(1 - \frac{1}{2\sqrt{N}})\xi}{|\mathcal{S}|H^3}\left\|\frac{d_\mu^{\pi^*}}{\mu}\right\|_\infty^{-2}n^2$$

$$\leq \frac{9}{8}\frac{c^2\eta^2(1 - \frac{1}{2\sqrt{N}})\xi}{\sqrt{N}|\mathcal{S}|H^3}\left\|\frac{d_\mu^{\pi^*}}{\mu}\right\|_\infty^{-2}n^2 \leq \frac{9}{8}\frac{c^2\max\{R^*,1\}^2(1 - \frac{1}{2\sqrt{N}})}{N^{3/2}|\mathcal{S}|H^{19}}\left\|\frac{d_\mu^{\pi^*}}{\mu}\right\|_\infty^{-2}n^2 \leq K_n.$$

Furthermore, we have for $\eta = \frac{1}{5H^2R^*\sqrt{N}}$ that

$$\frac{4}{3w} = \frac{4|\mathcal{S}|H^3}{3\eta(1 - \frac{1}{2\sqrt{N}})c^2}\left\|\frac{d_\mu^{\pi^*}}{\mu}\right\|_\infty^2 = \frac{20|\mathcal{S}|H^5R^*}{c^2\frac{1}{\sqrt{N}}(1 - \frac{1}{2\sqrt{N}})}\left\|\frac{d_\mu^{\pi^*}}{\mu}\right\|_\infty^2.$$

We obtain that

$$d_1 \leq HR^* \leq \frac{4}{3w} \leq \frac{4}{3w \cdot 1},$$

because $c \leq 1$, $\left\|\frac{d_\mu^{\pi^*}}{\mu}\right\|_\infty^2 \geq 1$ and $\frac{1}{\sqrt{N}}(1 - \frac{1}{2\sqrt{N}}) < 1$ for all $N \geq 1$.

Suppose the induction assumption $d_n \leq \frac{4}{3wn}$ holds true. First, recall the recursive inequality

$$d_{n+1} \leq d_n - wd_n^2 + \frac{B}{K_n}.$$

The function $f(x) = x - wx^2$ is monotonically increasing in $[0, \frac{1}{2w}]$, and by induction assumption $d_n \leq \frac{1}{4wn} \leq \frac{1}{2w}$. Thus,

$$d_{n+1} \leq d_n - wd_n^2 + \frac{B}{K_n}$$

$$\leq \frac{4}{3wn} - \frac{16}{9wn^2} + \frac{B}{K_n}$$

$$\leq \frac{4}{3wn} - \frac{16}{9wn^2} + \frac{4B}{9wBn^2}$$

$$= \frac{4}{3wn} - \frac{12}{9wn^2}$$

$$= \frac{4}{3w}\left(\frac{1}{n} - \frac{1}{n^2}\right)$$

$$\leq \frac{4}{3wn},$$

by the choice of $K_n \geq \frac{9}{4}wBn^2$. We deduce the claim

$$d_n \leq \frac{4}{3wn} = \frac{20|\mathcal{S}|H^5R^*}{c^2\frac{1}{\sqrt{N}}(1 - \frac{1}{2\sqrt{N}})n}\left\|\frac{d_\mu^{\pi^*}}{\mu}\right\|_\infty^2.$$

$\square$

Secondly, consider the complementary event $\{\tau \leq n\}$. We can bound the probability of this event by $\delta$ for a large enough batch size $K$. The proof is inspired by a similar result obtained by Ding et al. (2022, Lem. 6.3) for discounted MDPs.

**Lemma D.5.** *Let $\mu$ be a probability measure such that $\mu(s) > 0$ for all $s \in \mathcal{S}$ and consider the sequence $(\bar{\theta}^{(n)})_{n \geq 0}$ generated by Equation 8. For any $\delta > 0$, suppose that*

- *the batch size $K \geq \frac{10 \max\{R^*, 1\}^2 n^3}{c^2 \delta^2}$,*

- *the step size $\eta = \frac{1}{\sqrt{n} 5 H^2 R^*}$.*

*Then we have $\mathbb{P}(\tau \leq n) < \delta$.*

*Proof.* By the definition of $\tau$ we have

$$\mathbb{P}(\tau \leq n) = \mathbb{P}(\max_{0 \leq t \leq n} \|\theta^{(t)} - \bar{\theta}^{(t)}\| \geq \frac{c_h}{4}),$$

so we first study $\|\theta^{(t)} - \bar{\theta}^{(t)}\|$. We emphasise that (Ding et al., 2022, Lemma 6.3) established a similar recursive inequality.

$$\begin{aligned}
\|\bar{\theta}^{(t)} - \theta^{(t)}\| &= \|\bar{\theta}^{(0)} + \sum_{k=1}^{t-1} \eta \widehat{\nabla} J^K(\bar{\theta}^{(k)}, \mu) - (\theta^{(0)} + \sum_{k=1}^{t-1} \eta \nabla J(\theta^{(k)}, \mu))\| \\
&\leq \sum_{k=1}^{t-1} \eta \|\widehat{\nabla} J^K(\bar{\theta}^{(k)}, \mu) - \nabla J(\theta^{(k)}, \mu)\| \\
&\leq \eta \sum_{k=1}^{t-1} (\|\widehat{\nabla} J^K(\bar{\theta}^{(k)}, \mu) - \nabla J(\bar{\theta}^{(k)}, \mu)\| + \|\nabla J(\bar{\theta}^{(k)}, \mu) - \nabla J(\theta^{(k)}, \mu)\|).
\end{aligned}$$

We define again $\phi_k^K = \widehat{\nabla} J^K(\bar{\theta}^{(k)}, \mu) - \nabla J(\bar{\theta}^{(k)}, \mu)$ and continue using the $\beta$-lipschitz continuity of $\nabla J(\theta)$ such that

$$\begin{aligned}
\|\theta^{(t)} - \bar{\theta}^{(t)}\| &\leq \eta \sum_{k=1}^{t-1} (\|\phi_k^K\| + \beta \|\theta^{(k)} - \bar{\theta}^{(k)}\|) \\
&= \eta \sum_{k=1}^{t-1} \|\phi_k^K\| + \eta \beta \sum_{k=1}^{t-1} \|\theta^{(k)} - \bar{\theta}^{(k)}\|.
\end{aligned}$$

Using this inequality sequentially leads to

$$\begin{aligned}
\|\theta^{(t)} - \bar{\theta}^{(t)}\| &\leq \eta \sum_{k=1}^{t-1} \|\phi_k^K\| + \eta \beta \sum_{k=1}^{t-1} \|\theta^{(k)} - \bar{\theta}^{(k)}\| \\
&\leq \eta \sum_{k=1}^{t-1} \|\phi_k^K\| + \eta \beta \sum_{k=1}^{t-2} \|\theta^{(k)} - \bar{\theta}^{(k)}\| + \eta \beta \Big(\eta \sum_{k=1}^{t-2} \|\phi_k^K\| + \eta \beta \sum_{k=1}^{t-2} \|\theta^{(k)} - \bar{\theta}^{(k)}\|\Big) \\
&= \eta \sum_{k=1}^{t-1} \|\phi_k^K\| + \eta^2 \beta \sum_{k=1}^{t-2} \|\phi_k^K\| + (1 + \eta\beta)\eta\beta \sum_{k=1}^{t-2} \|\theta^{(k)} - \bar{\theta}^{(k)}\| \\
&= \eta \|\phi_{t-1}^K\| + \eta(1 + \eta\beta) \sum_{k=1}^{t-2} \|\phi_k^K\| + (1 + \eta\beta)\eta\beta \sum_{k=1}^{t-2} \|\theta^{(k)} - \bar{\theta}^{(k)}\| \\
&\leq \sum_{k=1}^{t-1} \eta(1 + \eta\beta)^{t-k-1} \|\phi_k^K\|.
\end{aligned}$$

Applying Markov's inequality results in

$$\mathbb{P}(\tau \le n) = \mathbb{P}(\max_{0 \le t \le n} \|\theta^{(t)} - \bar{\theta}^{(t)}\| \ge \frac{c}{4})$$

$$\le \mathbb{P}(\sum_{k=1}^{n-1} \eta(1+\eta\beta)^{n-k-1} \|\phi_k^K\| \ge \frac{c_h}{4})$$

$$\le \frac{4 \sum_{k=1}^{n-1} \eta(1+\eta\beta)^{n-k-1} \mathbb{E}[\|\phi_k^K\|]}{c}$$

$$\le \frac{4n\eta(1+\eta\beta)^{n-1} \sqrt{\frac{\xi}{K}}}{c},$$

where in the last inequality $\mathbb{E}[\|\phi_k^K\|] \le \sqrt{\mathbb{E}[\|\phi_k^K\|^2]} \le \sqrt{\frac{\xi}{K}}$ by Jensen's inequality and Lemma D.1. Now we plug in the choice of $\eta = \frac{1}{\sqrt{n}5H^2R^*} < \frac{1}{\sqrt{n}\beta}$,

$$\mathbb{P}(\tau \le n) \le \frac{4n\frac{1}{\sqrt{n}5H^2R^*}(1+\frac{1}{\sqrt{n}\beta}\beta)^{n-1}\sqrt{\frac{\xi}{K}}}{c}$$

$$= \frac{4\sqrt{n}(1+\frac{1}{\sqrt{n}})^{n-1}\sqrt{C_h}}{5H^2R^*c\sqrt{K}} \le \frac{4\sqrt{n}n\sqrt{\xi}}{5H^2R^*c\sqrt{K}},$$

where the last step is due to $f(x) = (1+\frac{1}{\sqrt{x}})^{x-1} \le x$ for all $x \ge 1$. We follow that $\mathbb{P}(\tau < n) < \delta$ if

$$\frac{16n^3\xi}{25H^4(R^*)^2c^2\delta^2} = \frac{16n^3H^4\max\{R^*,1\}^43}{25H^4(R^*)^2c^2\delta^2} \le \frac{48\max\{R^*,1\}^2n^3}{5c^2\delta^2} \le \frac{10\max\{R^*,1\}^2n^3}{c^2\delta^2} = K.$$

$\square$

**Theorem 4.1.** *Under Assumption 3.1, let $\mu$ be a probability measure such that $\mu(s) > 0$ for all $s \in \mathcal{S}$. Consider the final policy using Algorithm 1 with stochastic updates from Equation 8 denoted by $\hat{\pi}^* = \pi^{\bar{\theta}^{(N)}}$. Moreover, for any $\delta, \epsilon > 0$ assume that the number of training steps satisfies $N \ge \left(\frac{21|\mathcal{S}|H^5R^*}{\epsilon\delta c^2}\right)^2 \left\|\frac{d_\mu^{\pi^*}}{\mu}\right\|_\infty^4$, let $\eta = \frac{1}{5H^2R^*\sqrt{N}}$ and $K \ge \frac{10\max\{R^*,1\}^2N^3}{c^2\delta^2}$. Then it holds true that*

$$\mathbb{P}(V_0^*(\mu) - V_0^{\hat{\pi}^*}(\mu) < \epsilon) > 1 - \delta.$$

*Proof.* First note again, that by definition $J^*(\mu) = V_0^*(\mu)$ and $J(\bar{\theta}^{(N)}, \mu) = V_0^{\pi^{\bar{\theta}^{(N)}}}(\mu)$. We separate the probability using the stopping time $\tau$ and obtain

$$\mathbb{P}\left((J^*(\mu) - J(\bar{\theta}^{(N)}, \mu)) \ge \epsilon\right) \le \mathbb{P}\left(\{\tau \ge N\} \cap \{(J^*(\mu) - J(\bar{\theta}^{(N)}, \mu)) \ge \epsilon\}\right)$$

$$+ \mathbb{P}\left(\{\tau \le N\} \cap \{(J^*(\mu) - J(\bar{\theta}^{(N)}, \mu)) \ge \epsilon\}\right)$$

$$\le \frac{\mathbb{E}\left[(J^*(\mu) - J(\bar{\theta}^{(N)}, \mu))\mathbf{1}_{\{\tau \ge N\}}\right]}{\epsilon} + \mathbb{P}(\tau \le N)$$

$$\le \frac{1}{\epsilon}\frac{20|\mathcal{S}|H^5R^*}{c^2\frac{1}{\sqrt{N}}(1-\frac{1}{2\sqrt{N}})N}\left\|\frac{d_\mu^{\pi^*}}{\mu}\right\|_\infty^2 + \frac{\delta}{2}$$

$$\le \frac{\delta}{2} + \frac{\delta}{2}$$

$$= \delta,$$

where the second inequality holds due to Lemma D.4 and Lemma D.5. The last inequality follows by our choice of $N$:

$$\frac{20|\mathcal{S}|H^5 R^*}{c^2 \sqrt{N}(1 - \frac{1}{2\sqrt{N}})}\left\|\frac{d_\mu^{\pi^*}}{\mu}\right\|_\infty^2 \leq \frac{\delta}{2}$$

if and only if $N \geq \left(\frac{20|\mathcal{S}|H^5 R^*}{\epsilon \delta c^2}\left\|\frac{d_\mu^{\pi^*}}{\mu}\right\|_\infty^2 + \frac{1}{2}\right)^2$, which is satisfied if $N \geq \left(\frac{21|\mathcal{S}|H^5 R^*}{\epsilon \delta c^2}\right)^2 \left\|\frac{d_\mu^{\pi^*}}{\mu}\right\|_\infty^4$.
Note that we can use Lemma D.4 in the equation above with a constant batch size, because by our choice of $\eta$

$$\max\left\{\frac{9}{8}\frac{c^2 \max\{R^*, 1\}^2(1 - \frac{1}{2\sqrt{N}})}{N^{3/2}|\mathcal{S}|H^{19}}\left\|\frac{d_\mu^{\pi^*}}{\mu}\right\|_\infty^{-2} n^2, \frac{10 \max\{R^*, 1\}^2 N^3}{c^2 \delta^2}\right\} = \frac{10 \max\{R^*, 1\}^2 N^3}{c^2 \delta^2},$$

for all $n \leq N$. The last equality holds, as $c < 1$, $\left\|\frac{d_\mu^{\pi^*}}{\mu}\right\|_\infty^{-2} < 1$. $\qquad\square$

## D.2 DYNAMIC APPROACH

We start again by showing that the gradient estimator is unbiased and has bounded variance.

**Lemma D.6.** *For any $h \in \mathcal{H}$ and $K_h > 0$ it holds that*

$$\mathbb{E}_{\mu_h}^{(\pi^\theta, (\tilde{\pi})_{(h+1)})}[\widehat{\nabla} J_h^{K_h}(\theta, \tilde{\pi}_{(h+1)}, \mu_h)] = \nabla J_h(\theta, \tilde{\pi}_{(h+1)}, \mu_h)$$

*and*

$$\mathbb{E}_{\mu_h}^{(\pi^\theta, (\tilde{\pi})_{(h+1)})}[\|\widehat{\nabla} J_h^{K_h}(\theta, \tilde{\pi}_{(h+1)}, \mu_h) - \nabla J_h(\theta, \tilde{\pi}_{(h+1)}, \mu_h)\|^2] \leq \frac{5(H-h)^2(R^*)^2}{K_h} =: \frac{\psi_h}{K}.$$

*Proof.* By the definition of $\widehat{\nabla} J_h^K$ we have

$$\mathbb{E}_{\mu_h}^{(\pi^\theta, (\tilde{\pi})_{(h+1)})}[\widehat{\nabla} J_h^{K_h}(\theta, \tilde{\pi}_{(h+1)}, \mu_h)]$$

$$= \mathbb{E}_{\mu_h}^{(\pi^\theta, (\tilde{\pi})_{(h+1)})}\left[\frac{1}{K_h}\sum_{i=1}^{K_h}\nabla \log(\pi^\theta(A_t^i|S_t^i))\hat{R}_h^i\right]$$

$$= \mathbb{E}_{\mu_h}^{(\pi^\theta, (\tilde{\pi})_{(h+1)})}\left[\nabla \log(\pi^\theta(A_h^1|S_h^1))\hat{R}_h^1\right]$$

$$= \mathbb{E}_{\mu_h}^{(\pi^\theta, (\tilde{\pi})_{(h+1)})}\left[\nabla \log(\pi^\theta(A_h|S_h))\sum_{k=h}^{H-1} r(S_k, A_k)\right],$$

where we used that we consider independent samples for $i = 1, \ldots, K_h$. From the proof of the policy gradient Theorem A.6, we obtain that

$$\mathbb{E}_{\mu_h}^{(\pi^\theta, (\tilde{\pi})_{(h+1)})}[\widehat{\nabla} J_h^{K_h}(\theta, \tilde{\pi}_{(h+1)}, \mu)]$$

$$= \mathbb{E}_{\mu_h}^{(\pi^\theta, (\tilde{\pi})_{(h+1)})}\left[\nabla \log(\pi^\theta(A_1|S_h))\sum_{k=h}^{H-1} r(S_k, A_k)\right]$$

$$= \nabla J_h(\theta, \tilde{\pi}_{(h+1)}, \mu_h).$$

For the second claim we have

$$\mathbb{E}_{\mu_h}^{(\pi^\theta, (\tilde{\pi})_{(h+1)})}\left[\|\widehat{\nabla} J_h^{K_h}(\theta, \tilde{\pi}_{(h+1)}, \mu_h) - \nabla J_h(\theta, \tilde{\pi}_{(h+1)}, \mu_h)\|^2\right]$$

$$\leq \frac{1}{K_h}\mathbb{E}_{\mu_h}^{(\pi^\theta, (\tilde{\pi})_{(h+1)})}\left[\|\nabla \log(\pi^\theta(A_h|S_h))\hat{Q}_h(S_h, A_h) - \nabla J_h(\theta)\|^2\right]$$

$$= \frac{1}{K_h}\mathbb{E}_{\mu_h}^{(\pi^\theta, (\tilde{\pi})_{(h+1)})}\left[\Big(\sum_{s \in \mathcal{S}_h}\sum_{a \in \mathcal{A}_s}\Big(\mathbf{1}_{s=S_h}(\mathbf{1}_{a=A_h} - \pi^\theta(a|s))\sum_{k=h}^{H-1} r(S_k, A_k)\right.$$

$$\left.- \mu_h(s)\pi^\theta(a|s)A_h^{(\pi^\theta, (\tilde{\pi})_{(h+1)})}(s, a)\Big)^2\right],$$

by the definition of $\widehat{\nabla} J_h^{K_h}(\theta, \tilde{\pi}_{(h+1)}, \mu_h)$ and the derivative of $\nabla J_h(\theta, \tilde{\pi}_{(h+1)}, \mu_h)$ for the softmax parametrisation. Further,

$$\mathbb{E}_{\mu_h}^{(\pi^\theta, (\tilde{\pi})_{(h+1)})}\left[\|\widehat{\nabla} J_h^{K_h}(\theta, \tilde{\pi}_{(h+1)}, \mu_h) - \nabla J_h(\theta, \tilde{\pi}_{(h+1)}, \mu_h)\|^2\right]$$

$$\leq \frac{1}{K_h}\mathbb{E}_{\mu_h}^{(\pi^\theta, (\tilde{\pi})_{(h+1)})}\Big[\sum_{a\in\mathcal{A}_s}(\mathbf{1}_{a=A_h} - \pi^\theta(a|S_h))^2\Big(\sum_{k=h}^{H-1}r(S_k, A_k)\Big)^2$$

$$-2\sum_{a\in\mathcal{A}_s}(\mathbf{1}_{a=A_h} - \pi^\theta(a|S_h))\sum_{k=h}^{H-1}r(S_k, A_k)\mu_h(s)\pi^\theta(a|S_h)A_h^{(\pi^\theta, (\tilde{\pi})_{(h+1)})}(S_h, a)$$

$$+\sum_{s\in\mathcal{S}_h}\sum_{a\in\mathcal{A}_s}\mu_h(s)^2\pi^\theta(a|s)^2 A_h^{(\pi^\theta, (\tilde{\pi})_{(h+1)})}(s, a)^2\Big].$$

We consider all three terms separately. For the first term we have

$$\mathbb{E}_{\mu_h}^{(\pi^\theta, (\tilde{\pi})_{(h+1)})}\Big[\sum_{a\in\mathcal{A}_s}(\mathbf{1}_{a=A_h} - \pi^\theta(a|S_h))^2\Big(\sum_{k=h}^{H-1}r(S_k, A_k)\Big)^2\Big]$$

$$= \mathbb{E}_{\mu_h}^{(\pi^\theta, (\tilde{\pi})_{(h+1)})}\Big[\Big(\sum_{k=h}^{H-1}r(S_k, A_k)\Big)^2\Big] - 2\mathbb{E}_{\mu_h}^{(\pi^\theta, (\tilde{\pi})_{(h+1)})}\Big[\pi^\theta(A_h|S_h)\Big(\sum_{k=h}^{H-1}r(S_k, A_k)\Big)^2\Big]$$

$$+ \mathbb{E}_{\mu_h}^{(\pi^\theta, (\tilde{\pi})_{(h+1)})}\Big[\sum_{a\in\mathcal{A}_s}\pi^\theta(a|S_h)^2\Big(\sum_{k=h}^{H-1}r(S_k, A_k)\Big)^2\Big]$$

$$\leq ((H-h)R^*)^2 - 0 + ((H-h)R^*)^2$$

$$= 2((H-h)R^*)^2,$$

by bounded reward assumption and the fact that $\pi^\theta$ is a probability distribution. For the second term, we note that $A_h^{(\pi^\theta, (\tilde{\pi})_{(h+1)})}(S_h, a)$ can be negative, therefore we consider the absolute value and obtain

$$2\mathbb{E}_{\mu_h}^{(\pi^\theta, (\tilde{\pi})_{(h+1)})}\Big[\sum_{a\in\mathcal{A}_s}(\mathbf{1}_{a=A_h} - \pi^\theta(a|S_h))\sum_{k=h}^{H-1}r(S_k, A_k)\mu_h(s)\pi^\theta(a|S_h)\big|A_h^{(\pi^\theta, (\tilde{\pi})_{(h+1)})}(S_h, a)\big|\Big]$$

$$\leq 2\mathbb{E}_{\mu_h}^{(\pi^\theta, (\tilde{\pi})_{(h+1)})}\Big[\sum_{a\in\mathcal{A}_s}1\cdot(H-h)R^*\cdot 1\cdot\pi^\theta(a|S_h)\cdot(H-h)R^*\Big]$$

$$= 2((H-h)R^*)^2.$$

For the last term we have

$$\mathbb{E}_{\mu_h}^{(\pi^\theta, (\tilde{\pi})_{(h+1)})}\Big[\sum_{s\in\mathcal{S}_h}\sum_{a\in\mathcal{A}_s}\mu_h(s)^2\pi^\theta(a|s)^2 A_h^{(\pi^\theta, (\tilde{\pi})_{(h+1)})}(s, a)^2\Big] \leq ((H-h)R^*)^2.$$

In total, it holds that

$$\mathbb{E}_{\mu_h}^{(\pi^\theta, (\tilde{\pi})_{(h+1)})}\Big[\|\widehat{\nabla} J_h^{K_h}(\theta, \tilde{\pi}_{(h+1)}, \mu_h) - \nabla J_h(\theta, \tilde{\pi}_{(h+1)}, \mu_h)\|^2\Big] \leq \frac{5((H-h)R^*)^2}{K_h}.$$

$\square$

Recall $(\bar{\theta}_h^{(n)})_{n\geq 0}$ be the stochastic process from Equation 10 and let $(\theta_h^{(n)})_{n\geq 0}$ be the deterministic sequence generated by PG with exact gradients,

$$\theta_h^{(n+1)} = \theta_h^{(n)} + \eta_h\nabla J_h(\theta_h^{(n)}, \tilde{\pi}_{(h+1)}, \mu_h)$$

such that the initial parameter agree, $\theta_h^{(0)} = \bar{\theta}_h^{(0)}$, and the step size $\eta_h$ is the same for both processes. The natural filtration of $(\bar{\theta}_h^{(n)})_{n\geq 0}$ is denoted by $(\mathcal{F}_h^{(n)})_{n\geq 0}$.

For the deterministic scheme we could assure $c_h = \min_{n \geq 0} \min_{s \in \mathcal{S}} \pi^{\theta_h^{(n)}}(a^*(s)|s)$ is bounded away from 0 by Lemma B.10. As for the simultaneous PG this cannot be guaranteed for the stochastic trajectory. Define for every epoch the following stopping time

$$\tau_h := \min\{n \geq 0 : \|\theta_h^{(n)} - \bar{\theta}_h^{(n)}\|_2 \geq \frac{c_h}{4}\}.$$

We emphasise that $\tau_h$ is a stopping time with respect to the filtration $(\mathcal{F}_h^{(n)})_{n \geq 0}$ by construction.

It follows again by the $\sqrt{2}$-Lipschitz continuity of $\theta \mapsto \pi^\theta(a^*(s)|s)$ (Lemma D.2) that $\min_{0 \leq n \leq \tau_h} \min_{s \in \mathcal{S}} \pi^{\bar{\theta}_h^{(n)}}(a^*(s)|s) \geq \frac{c_h}{2} > 0$.

**Lemma D.7.** *Let $\mu_h$ be probability measures such that $\mu_h(s) > 0$ for all $s \in \mathcal{S}_h$ and consider the sequence $(\theta_h^{(n)})_{n \geq 0}$ generated by Equation 10. Then, it holds almost surely that $\min_{0 \leq n \leq \tau_h} \min_{s \in \mathcal{S}_h} \pi^{\bar{\theta}_h^{(n)}}(a^*(s)|s) \geq \frac{c_h}{2}$ is strictly positive.*

*Proof.* For every $n \leq \tau$ we obtain by the $\sqrt{2}$-Lipschitz continuity in Lemma D.2 that

$$
\begin{aligned}
\pi^{\bar{\theta}_h^{(n)}}(a^*(s)|s) &\geq \pi^{\theta_h^{(n)}}(a^*(s)|s) - |\pi^{\theta_h^{(n)}}(a^*(s)|s) - \pi^{\bar{\theta}_h^{(n)}}(a^*(s)|s)| \\
&\geq \pi^{\theta_h^{(n)}}(a^*(s)|s) - \sqrt{2}\|\bar{\theta}_h^{(n)} - \theta_h^{(n)}\|_2 \\
&> \frac{c_h}{2} > 0,
\end{aligned}
$$

holds almost surely. The claim follows directly. $\qquad\square$

We derive a convergence rate on the event $\{n \leq \tau_h\}$ in the following sense:

**Lemma D.8.** *Let $\mu_h$ be probability measures such that $\mu_h(s) > 0$ for all $s \in \mathcal{S}_h$ and consider the sequence $(\theta_h^{(n)})_{n \geq 0}$ generated by Equation 10. Suppose that*

- *the batch size $K_h^{(n)} \geq \frac{45 c_h^2}{64 N_h^{\frac{3}{2}}}(1 - \frac{1}{2\sqrt{N_h}})n^2$ is increasing for some $N_h \geq 1$*

- *the step size $\eta_h = \frac{1}{2(H-h)R^*\sqrt{N_h}}$.*

*Then,*

$$\mathbb{E}\big[(J_h^*(\tilde{\pi}_{(h+1)}, \mu_h) - J_h(\bar{\theta}_h^{(n)}, \tilde{\pi}_{(h+1)}, \mu_h))\mathbf{1}_{\{n \leq \tau_h\}}\big] \leq \frac{32\sqrt{N_h}(H-h)R^*}{3(1 - \frac{1}{2\sqrt{N_h}})c_h^2 n}.$$

*Proof.* As in the proof of Theorem 4.1 we deduce from the $\beta_h$-smoothness and Lemma D.6, that

$$
\begin{aligned}
&\mathbb{E}\Big[J(\bar{\theta}_h^{(n+1)}, \tilde{\pi}_{(h+1)}, \mu_h)|\mathcal{F}_h^{(n)}\Big] \\
&\geq J(\bar{\theta}_h^{(n)}, \tilde{\pi}_{(h+1)}, \mu_h) + \Big(\eta_h - \frac{\beta_h \eta_h^2}{2}\Big)\|\nabla J(\bar{\theta}_h^{(n)}, \tilde{\pi}_{(h+1)}, \mu_h)\|^2 - \frac{\beta_h \eta_h^2 \psi_h}{2K_h^{(n)}}.
\end{aligned}
$$

We take the expectation of this inequality on both sides under the event $\{n + 1 \leq \tau_h\}$. Note that $\{n + 1 \leq \tau_h\} = \{\tau_h \leq n\}^C$ is $\mathcal{F}_n$-measurable and that $\mathbf{1}_{\{n+1 \leq \tau_h\}} \leq \mathbf{1}_{\{n \leq \tau_h\}}$ a.s., thus

$$
\mathbb{E}\Big[(J_h^*(\tilde{\pi}_{(h+1)}, \mu_h) - J_h(\bar{\theta}_h^{(n+1)}, \tilde{\pi}_{(h+1)}, \mu_h))\mathbf{1}_{\{n+1 \leq \tau_h\}}\Big]
$$
$$
= \mathbb{E}\Big[\mathbb{E}\Big[(J_h^*(\tilde{\pi}_{(h+1)}, \mu_h) - J_h(\bar{\theta}_h^{(n+1)}, \tilde{\pi}_{(h+1)}, \mu_h))|\mathcal{F}_h^{(n)}\Big]\mathbf{1}_{\{n+1 \leq \tau_h\}}\Big]
$$
$$
\leq \mathbb{E}\Big[\Big(J_h^*(\tilde{\pi}_{(h+1)}, \mu_h) - \mathbb{E}\Big[J_h(\bar{\theta}_h^{(n+1)}, \tilde{\pi}_{(h+1)}, \mu_h)|\mathcal{F}_h^{(n)}\Big]\Big)\mathbf{1}_{\{n \leq \tau_h\}}\Big]
$$
$$
\leq \mathbb{E}\Big[(J_h^*(\tilde{\pi}_{(h+1)}, \mu_h) - J_h(\bar{\theta}_h^{(n)}, \tilde{\pi}_{(h+1)}, \mu_h))\mathbf{1}_{\{n \leq \tau_h\}}\Big]
$$
$$
- \Big(\eta_h - \frac{\beta_h \eta_h^2}{2}\Big)\mathbb{E}\Big[\|\nabla J_h(\bar{\theta}_h^{(n)}, \tilde{\pi}_{(h+1)}, \mu_h)\|^2 \mathbf{1}_{\{n \leq \tau_h\}}\Big] + \frac{\beta_h \eta_h^2 \psi_h}{2K_h^{(n)}}
$$
$$
= \mathbb{E}\Big[(J_h^*(\tilde{\pi}_{(h+1)}, \mu_h) - J_h(\bar{\theta}_h^{(n)}, \tilde{\pi}_{(h+1)}, \mu_h))\mathbf{1}_{\{n \leq \tau_h\}}\Big]
$$
$$
- \eta_h\Big(1 - \frac{1}{2\sqrt{N_h}}\Big)\mathbb{E}\Big[\|\nabla J_h(\bar{\theta}_h^{(n)}, \tilde{\pi}_{(h+1)}, \mu_h)\|^2 \mathbf{1}_{\{n \leq \tau_h\}}\Big] + \frac{5(H-h)R^*}{2K_h^{(n)}N_h}.
$$

By Lemma B.9 we have that

$$
\|\nabla J_h(\bar{\theta}_h^{(n)}, \tilde{\pi}_{(h+1)}, \mu_h)\|^2 \geq \min_{s \in \mathcal{S}} \pi^{\theta_n}(a^*(s|s))^2 (J_h^*(\tilde{\pi}_{(h+1)}, \mu_h) - J_h(\bar{\theta}_h^{(n)}, \tilde{\pi}_{(h+1)}, \mu_h))^2
$$

almost surely, and by Lemma D.7 we have that $\min_{0 \leq n \leq \tau_h} \min_{s \in \mathcal{S}} \pi^{\bar{\theta}_h^{(n)}}(a^*(s|s))^2 \geq \frac{c_h}{2} > 0$ almost surly. Therefore,

$$
\mathbb{E}\Big[(J_h^*(\tilde{\pi}_{(h+1)}, \mu_h) - J_h(\bar{\theta}_h^{(n+1)}, \tilde{\pi}_{(h+1)}, \mu_h)\mathbf{1}_{\{n+1 \leq \tau_h\}}\Big]
$$
$$
\leq \mathbb{E}\Big[(J_h^*(\tilde{\pi}_{(h+1)}, \mu_h) - J_h(\bar{\theta}_h^{(n)}, \tilde{\pi}_{(h+1)}, \mu_h))\mathbf{1}_{\{n \leq \tau_h\}}\Big]
$$
$$
- \eta_h\Big(1 - \frac{1}{2\sqrt{N_h}}\Big)\mathbb{E}\Big[\min_{s \in \mathcal{S}} \pi^{\theta_n}(a^*(s|s))^2 (J_h^*(\tilde{\pi}_{(h+1)}, \mu_h) - J_h(\theta_n))^2 \mathbf{1}_{\{n \leq \tau_h\}}\Big] + \frac{5(H-h)R^*}{2K_h^{(n)}N_h},
$$
$$
\leq \mathbb{E}\Big[(J_h^*(\tilde{\pi}_{(h+1)}, \mu_h) - J_h(\bar{\theta}_h^{(n)}, \tilde{\pi}_{(h+1)}, \mu_h))\mathbf{1}_{\{n \leq \tau_h\}}\Big]
$$
$$
- \eta_h\Big(1 - \frac{1}{2\sqrt{N_h}}\Big)\frac{c_h^2}{4}\mathbb{E}\Big[(J_h^*(\tilde{\pi}_{(h+1)}, \mu_h) - J_h(\bar{\theta}_h^{(n)}, \tilde{\pi}_{(h+1)}, \mu_h))\mathbf{1}_{\{n \leq \tau_h\}}\Big]^2 + \frac{5(H-h)R^*}{2K_h^{(n)}N_h},
$$

where we used Jensen's inequality in the last step.

For $d_n := \mathbb{E}\Big[(J_h^*(\tilde{\pi}_{(h+1)}, \mu_h) - J_h(\bar{\theta}_h^{(n)}, \tilde{\pi}_{(h+1)}, \mu_h))\mathbf{1}_{\{n \leq \tau_h\}}\Big]$ we imply the recursive inequality

$$
d_{n+1} \leq d_n - \eta_h\Big(1 - \frac{1}{2\sqrt{N_h}}\Big)\frac{c_h^2}{4}d_n^2 + \frac{5(H-h)R^*}{2K_h^{(n)}N_h}.
$$

Define $w := \eta_h\Big(1 - \frac{1}{2\sqrt{N_h}}\Big)\frac{c_h^2}{4} > 0$ and $B = \frac{5(H-h)R^*}{2N_h} > 0$, then

$$
d_{n+1} \leq d_n(1 - wd_n) + \frac{B}{K_h^{(n)}}
$$

and by our choice of $\eta_h$,

$$
K_h^{(n)} \geq \frac{45c_h^2}{64N_h^{\frac{3}{2}}}(1 - \frac{1}{2\sqrt{N_h}})n^2 = \frac{9}{4}wBn^2,
$$

Moreover, it holds that

$$
d_1 \leq (H-h)R^* \leq \frac{1}{\eta_h} \leq \frac{4}{3w} \leq \frac{4}{3w \cdot 1},
$$

because $c_h \leq 1$ and $\frac{1}{\sqrt{N_h}}(1 - \frac{1}{2\sqrt{N_h}}) < 1$ for all $N_h \geq 1$. Suppose the induction assumption $d_n \leq \frac{4}{3wn}$ holds true, then for $d_{n+1}$,

$$d_{n+1} \leq d_n - wd_n^2 + \frac{B}{K_h^{(n)}}.$$

The function $f(x) = x - wx^2$ is monotonically increasing in $[0, \frac{1}{2w}]$ and by induction assumption $d_n \leq \frac{1}{4wn} \leq \frac{1}{2w}$. So $d_n - wd_n^2 \leq \frac{4}{3wn}$ which implies

$$
\begin{aligned}
d_{n+1} &\leq d_n - wd_n^2 + \frac{B}{K_h^{(n)}} \\
&\leq \frac{4}{3wn} - \frac{16}{9wn^2} + \frac{B}{K_n} \\
&\leq \frac{4}{3wn} - \frac{16}{9wn^2} + \frac{4B}{9wBn^2} \\
&= \frac{4}{3wn} - \frac{12}{9wn^2} \\
&= \frac{4}{3w}\left(\frac{1}{n} - \frac{1}{n^2}\right) \\
&\leq \frac{4}{3w(n+1)},
\end{aligned}
$$

where we used that $K_h^{(n)} \geq \frac{9}{4}wBn^2$. We follow the claim

$$d_n \leq \frac{4}{3wn} = \frac{32\sqrt{N_h}(H-h)R^*}{3(1 - \frac{1}{2\sqrt{N_h}})c_h^2 n}.$$

$\square$

Secondly, consider the complementary event $\{\tau \leq n\}$. We can bound the probability of this event by $\delta$ for a large enough batch size $K_h$. The proof is again inspired by similar results obtained in Ding et al. (2022, Lem. 6.3) for discounted MDPs.

**Lemma D.9.** *Let $\mu_h$ be probability measures such that $\mu_h(s) > 0$ for all $s \in \mathcal{S}_h$ and consider the sequence $(\theta_h^{(n)})_{n \geq 0}$ generated by Equation 10. For any $\delta > 0$, suppose that*

- *the batch size $K_h \geq \frac{5n^3}{c_h^2\delta^2}$*

- *the step size $\eta_h = \frac{1}{\sqrt{n}\beta_h}$.*

*Then, we have $\mathbb{P}(\tau_h \leq n) < \delta$.*

*Proof.* The proof follows line by line the one of Lemma D.5.

One obtains

$$
\begin{aligned}
\mathbb{P}(\tau_h \leq n) &= \mathbb{P}\left(\max_{0 \leq t \leq n} \|\theta_h^{(n)} - \bar{\theta}_h^{(n)}\| \geq \frac{c_h}{4}\right) \\
&\leq \frac{4n\eta_h(1 + \eta_h\beta_h)^{n-1}\sqrt{\frac{\psi_h}{K_h}}}{c_h},
\end{aligned}
$$

where $\psi_h$ from Lemma D.6. Now we plug in the choice of $\eta_h = \frac{1}{\sqrt{n}\beta_h} = \frac{1}{2(H-h)R^*\sqrt{n}}$,

$$\mathbb{P}(\tau_h \leq n) \leq \frac{4n\frac{1}{\sqrt{n}\beta_h}(1 + \frac{1}{\sqrt{n}\beta_h}\beta_h)^{n-1}\sqrt{\frac{\xi_h}{K_h}}}{c_h}$$

$$= \frac{4\sqrt{n}(1 + \frac{1}{\sqrt{n}})^{n-1}\sqrt{\psi_h}}{\beta_h c_h \sqrt{K_h}}$$

$$\leq \frac{2n\sqrt{n}\sqrt{\psi_h}}{\beta_h c_h \sqrt{K_h}} = \frac{n\sqrt{5n}}{c_h\sqrt{K_h}},$$

where the last step is due to $f(x) = (1 + \frac{1}{\sqrt{x}})^{x-1} \leq x$ for all $x \geq 1$. We follow that $\mathbb{P}(\tau_h < n) < \delta$ if

$$K_h \geq \frac{5n^3}{c_h^2\delta^2}.$$

$\square$

We are now ready to proof the epoch wise statement.

**Lemma D.10.** *Let $\mu_h$ be probability measures such that $\mu_h(s) > 0$ for all $s \in \mathcal{S}_h$ and consider the sequence $(\theta_h^{(n)})_{n\geq 0}$ generated by Equation 10. Moreover, for any $\delta, \epsilon > 0$, assume that*

*(i) the number of training steps $N_h \geq \left(\frac{12(H-h)R^*}{\epsilon\delta c_h^2}\right)^2$,*

*(ii) the step size $\eta_h = \frac{1}{2(H-h)R^*\sqrt{N_h}}$ and the batch size $K_h = \frac{5N_h^3}{c_h^2\delta^2}$.*

*Then, it holds true that $\mathbb{P}\left(J_h^*(\tilde{\pi}_{(h+1)}, \mu_h) - J_h(\bar{\theta}_h^{(N_h)}, \tilde{\pi}_{(h+1)}, \mu_h) \geq \epsilon\right) \leq \delta$.*

*Proof.* We separate the probability using the stopping time $\tau_h$ and obtain

$$\mathbb{P}\left(J_h^*(\tilde{\pi}_{(h+1)}, \mu_h) - J_h(\bar{\theta}_h^{(N_h)}, \tilde{\pi}_{(h+1)}, \mu_h) \geq \epsilon\right)$$

$$\leq \mathbb{P}\left(\{\tau_h \geq N_h\} \cap \{J_h^*(\tilde{\pi}_{(h+1)}, \mu_h) - J_h(\bar{\theta}_h^{(N_h)}, \tilde{\pi}_{(h+1)}, \mu_h) \geq \epsilon\}\right)$$

$$+ \mathbb{P}\left(\{\tau_h \leq N_h\} \cap \{J_h^*(\tilde{\pi}_{(h+1)}, \mu_h) - J_h(\bar{\theta}_h^{(N_h)}, \tilde{\pi}_{(h+1)}, \mu_h) \geq \epsilon\}\right)$$

$$\leq \frac{\mathbb{E}\left[(J_h^*(\tilde{\pi}_{(h+1)}, \mu_h) - J_h(\bar{\theta}_h^{(N_h)}, \tilde{\pi}_{(h+1)}, \mu_h))\mathbf{1}_{\{\tau_h \geq N_h\}}\right]}{\epsilon} + \mathbb{P}(\tau_h \leq N_h)$$

$$\leq \frac{1}{\epsilon}\frac{32\sqrt{N_h}(H-h)R^*}{3(1 - \frac{1}{2\sqrt{N_h}})c_h^2 n} + \frac{\delta}{2}$$

$$\leq \frac{\delta}{2} + \frac{\delta}{2}$$

$$= \delta,$$

where the second inequality it due to Lemma D.8 and Lemma D.9. The last inequality follows by our choice of $N_h$:

$$\frac{32\sqrt{N_h}(H-h)R^*}{3\epsilon(1 - \frac{1}{2\sqrt{N_h}})c_h^2 n} \leq \frac{11\sqrt{N_h}(H-h)R^*}{\epsilon(1 - \frac{1}{2\sqrt{N_h}})c_h^2 n} \leq \frac{\delta}{2}$$

for $N_h \geq \left(\frac{11(H-h)R^*}{\epsilon\delta c_h^2} + \frac{1}{2}\right)^2$, which is satisfied for $N_h \geq \left(\frac{12(H-h)R^*}{\epsilon\delta c_h^2}\right)^2$. Note further that we could use Lemma D.8 in the equation above with a constant batch size $K_h$, because

$$\max\left\{\frac{45c_h^2}{64N_h^{\frac{3}{2}}}(1 - \frac{1}{2\sqrt{N_h}})n^2, \frac{5N_h^3}{c_h^2\delta^2}\right\} = \frac{5N_h^3}{c_h^2\delta^2},$$

for all $n \leq N_h$, as $(1 - \frac{1}{2\sqrt{N_h}}) < 1$ and $c_h < 1$.

$\square$

**Theorem 4.2.** *For all $h \in \mathcal{H}$, let $\mu_h$ be probability measures such that $\mu_h(s) > 0$ for all $h \in \mathcal{H}$, $s \in \mathcal{S}_h$. Consider the final policy using Algorithm 2 with stochastic updates from Equation 10 denoted by $\hat{\pi}^* = (\pi^{\bar{\theta}_0^{(N_0)}}, \dots, \pi^{\bar{\theta}_{H-1}^{(N_{H-1})}})$. Moreover, for any $\delta, \epsilon > 0$ assume that the numbers of training steps satisfy $N_h \geq \left( \frac{12(H-h)R^* H^2 \left\| \frac{1}{\mu_h} \right\|_\infty}{\delta c_h^2 \epsilon} \right)^2$, let $\eta_h = \frac{1}{2(H-h)R^* \sqrt{N_h}}$ and $K_h \geq \frac{5 N_h^3 H^2}{c_h^2 \delta^2}$. Then it holds true that*

$$\mathbb{P}\Big( \forall s \in \mathcal{S}_0 : V_0^*(s) - V_0^{\hat{\pi}^*}(s) < \epsilon \Big) > 1 - \delta.$$

*Proof.* As in the proof of the exact gradient case (Theorem 3.5) Equation 24 we have by our choice of the future policy $\tilde{\pi} = \hat{\pi}^*$ that

$$J_h(\bar{\theta}_h^{(N_h)}, \tilde{\pi}_{(h+1)}, \delta_s) = V_h^{\hat{\pi}^*}(s). \tag{37}$$

By Lemma D.10 we have that

$$\mathbb{P}\Big( J_h^*(\tilde{\pi}_{(h+1)}, \mu_h) - J_h(\bar{\theta}_h^{(N_h)}, \tilde{\pi}_{(h+1)}, \mu_h) \geq \frac{\epsilon}{H \left\| \frac{1}{\mu_h} \right\|_\infty} \Big) \leq \frac{\delta}{H},$$

by our choice of $N_h$, $\eta_h$ and $K_h$.

For every $s \in \mathcal{S}_h$, denote by $\delta_s$ the dirac measure on state $s$, then as in Equation 25

$$J_h^*(\tilde{\pi}_{(h+1)}, \delta_s) - J_h(\bar{\theta}_h^{(N_h)}, \tilde{\pi}_{(h+1)}, \delta_s) \leq \left\| \frac{1}{\mu_h} \right\|_\infty (J_h^*(\tilde{\pi}_{(h+1)}, \mu_h) - J_h(\theta_h^{(N_h)}, \tilde{\pi}_{(h+1)}, \mu_h)) \quad \text{a.s.}$$

Thus, for all $h \in \mathcal{H}$ it holds that

$$\mathbb{P}\Big( \exists s \in \mathcal{S}_h : J_h^*(\tilde{\pi}_{(h+1)}, \delta_s) - J_h(\bar{\theta}_h^{(N_h)}, \tilde{\pi}_{(h+1)}, \delta_s) \geq \frac{\epsilon}{H} \Big)$$
$$\leq \mathbb{P}\Big( J_h^*(\tilde{\pi}_{(h+1)}, \mu_h) - J_h(\theta_h^{(N_h)}, \tilde{\pi}_{(h+1)}, \mu_h) \geq \frac{\epsilon}{H \left\| \frac{1}{\mu_h} \right\|_\infty} \Big) \leq \frac{\delta}{H}. \tag{38}$$

Define the event $A_h := \{ J_h^*(\tilde{\pi}_{(h+1)}, \delta_s) - J_h(\bar{\theta}_h^{(N_h)}, \tilde{\pi}_{(h+1)}, \delta_s) < \frac{\epsilon}{H}, \forall s \in \mathcal{S}_h \}$. Then Equation 38 is equivalent to $\mathbb{P}(A_h^C) \leq \frac{\delta}{H}$. For $h = H - 1$ it follows directly with Equation 37 and the special property of the last time point that

$$\mathbb{P}\Big( \exists s \in \mathcal{S}_h : V_{H-1}^*(s) - V_{H-1}^{\hat{\pi}^*}(s) \geq \frac{\epsilon}{H} \Big)$$
$$= \mathbb{P}\Big( \exists s \in \mathcal{S}_h : J_{H-1}^*(\delta_s) - J_{H-1}(\bar{\theta}_h^{(N_h)}, \delta_s) \geq \frac{\epsilon}{H} \Big) \leq \frac{\delta}{H}.$$

We close the proof by induction. Assume for some $0 < h < H$ that

$$\mathbb{P}\Big( \exists s \in \mathcal{S}_h : V_h^*(s) - V_h^{\hat{\pi}^*}(s) \geq \frac{\epsilon(H-h)}{H} \Big) \leq \frac{\delta(H-h)}{H}. \tag{39}$$

Define $B_h := \{ V_h^*(s) - V_h^{\hat{\pi}^*}(s) < \frac{\epsilon(H-h)}{H}, \forall s \in \mathcal{S}_h \}$. Similar to Equation 28, on the event $B_h$ it holds that

$$J_{h-1}^*(\tilde{\pi}_{(h)}, \delta_s) = \max_{a \in \mathcal{A}_s} \Big( r(s,a) + \sum_{s' \in \mathcal{S}_h} p(s'|s,a) V_h^*(s) - \sum_{s' \in \mathcal{S}_h} p(s'|s,a)(V_h^*(s) - V_h^{\hat{\pi}^*}(s)) \Big)$$
$$> \max_{a \in \mathcal{A}_s} \Big( r(s,a) + \sum_{s' \in \mathcal{S}_h} p(s'|s,a) V_h^*(s) \Big) - \frac{\epsilon(H-h)}{H}$$
$$= V_{h-1}^*(s) - \frac{\epsilon(H-h)}{H}.$$

We obtain on the event $A_{h-1} \cap B_h$ that (compare to Equation 29)

$$V_{h-1}^*(s) - V_{h-1}^{\hat{\pi}^*}(s) = V_{h-1}^*(s) - J_{h-1}^*(\tilde{\pi}_{(h)}, \delta_s) + J_{h-1}^*(\tilde{\pi}_{(h)}, \delta_s) - V_{h-1}^{\hat{\pi}^*}(s)$$
$$< \frac{\epsilon(H-h)}{H} + \frac{\epsilon}{H}$$
$$= \frac{\epsilon(H-(h-1))}{H},$$

for every $s \in \mathcal{S}_{h-1}$. Hence, $A_{h-1} \cap B_h \subseteq B_{h-1}$. Finally, we close the induction by

$$
\begin{aligned}
\mathbb{P}&\left(\exists s \in \mathcal{S}_{h-1} : V_{h-1}^*(s) - V_{h-1}^{\hat{\pi}^*}(s) \geq \frac{\epsilon(H - (h-1))}{H}\right) \\
&= 1 - \mathbb{P}(B_{h-1}) \leq 1 - \mathbb{P}(A_{h-1} \cap B_h) = \mathbb{P}(A_{h-1}^C \cup B_h^C) \leq \mathbb{P}(A_{h-1}^C) + \mathbb{P}(B_h^C) \\
&= \mathbb{P}\left(\exists s \in \mathcal{S}_{h-1} : J_{h-1}^*(\tilde{\pi}_{(h)}, \delta_s) - J_{h-1}(\theta_{h-1}^{(N_h-1)}, \tilde{\pi}_{(h)}, \delta_s) \geq \frac{\epsilon}{H}\right) \\
&\quad + \mathbb{P}\left(\exists s \in \mathcal{S}_h : V_h^*(s) - V_h^{\hat{\pi}^*}(s) \geq \frac{\epsilon(H-h)}{H}\right) \\
&\leq \frac{\delta}{H} + \frac{\delta(H-h)}{H} \\
&= \frac{\delta(H - (h-1))}{H}.
\end{aligned}
$$

Finally, for $h = 0$ we have shown the assertion

$$
\mathbb{P}\left(\exists s \in \mathcal{S}_0 : V_0^*(s) - V_0^{\hat{\pi}^*}(s) \geq \epsilon\right) \leq \delta.
$$

$\square$

# E  EXAMPLE

We enclose a numerical toy example of a very simple MDP problem of optimally stopping when throwing a dice $H = 5$ times. This is a non-trivial example for which exact policy gradients can be computed. The simulations show that the theoretical results (in the exact gradient setup) are sharp up to constants.

The MDP corresponding to this example is defined as follows:

- a constant state space over the epochs $\mathcal{S} = \{1, \ldots, 6, \Delta\}$ containing all sides of the dice $1, \ldots, 6$ and a terminal state $\Delta$,
- a constant action space $\mathcal{A} = \{0, 1\}$, where $1$ indicates stopping and jumping into the terminal state and $0$ indicates continuing to the next epoch,
- a transition function $p$

$$
\begin{aligned}
p(s' \mid s, a) &= \mathbb{P}(S_{h+1} = s' \mid S_h = a, A_h = a) \\
&= \begin{cases} \frac{1}{6}, & \text{if } s', s \in \{0, 1 \ldots, 6\}, a = 0, \\ 1, & \text{if } s' = \Delta, s \in \mathcal{S}, a = 1 \text{ or } s' = s = \Delta, a = 0, \\ 0, & \text{otherwise.} \end{cases}
\end{aligned}
$$

Thus, we throw the dice iid until stopping for the first time, then we jump into the terminal state and stay there for the rest of the game.

- a reward function $r$

$$
r(s, a) = \begin{cases} s, & \text{if } s \in \{0, 1 \ldots, 6\}, a = 1, \\ 0, & \text{otherwise.} \end{cases}
$$

We only observe a reward when we choose action $1$ to top the game and the reward equals the number on the dice.

Having this model with known transition probabilities allows us to implement the simultaneous and dynamic PG under the exact gradient assumption. In the simulation we always initialised the parameters uniformly and chose $\theta \equiv 0$. Furthermore we chose the suggested learning rates from Theorem 3.2 in the simultaneous approach and from Theorem 3.5 in the dynamic approach.

First, note that Figure 2 (a) is the same as Figure 1 from the introduction. The dotted red line in this plot shows the target: $V_0^*$. On the $x$-axis we count the number of gradient computations in the algorithms, a way of measuring the computational complexity. The dashed magenta curve shows the

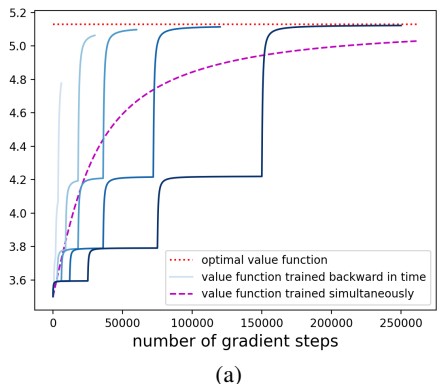 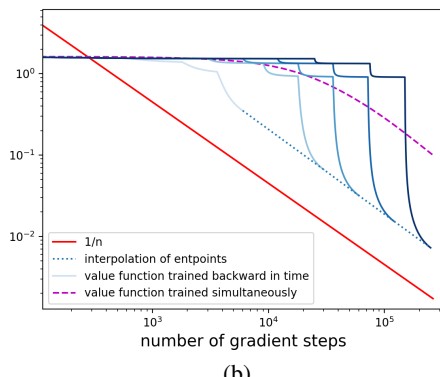

(a)                    (b)

Figure 2: (a) shows the behavior of $V_0^{\pi^{\theta^{(n)}}}$ during the training steps over all epochs. (b) shows the log-log plot of the same simulation visualizing the convergence rate towards $V_0^*$.

evolution of the estimated value function trained with the simultaneous training of all parameters. As $c$ is unknown for the approach, we trained the parameters until an error of $0.1$ was achieved. The blue curves show the evolution of the estimated value function trained with our algorithm backwards. Note that the number of gradient steps varies for different epochs, as suggested by Theorem 3.5, less training for later epochs. This can be seen in the plot by the different lengths of the plateaus of the blue lines. One plateau shows the training of one parameter. Just when the last parameter $\theta_0$ is trained, the value function $V_0^{\pi^{\theta}}$ finally converges towards the target. In this simulation we chose $\epsilon = 5, 1, 0.5, 0.25, 0.12$ to define the length of the training steps according to Theorem 3.5. Note that the uniform initialisation lead to $c_h = 0.5$ such that $N_h$ could be explicitly calculated. From light to dark blue $\epsilon$ decreases. It can be seen that the final error is better than the chosen epsilon, indicating that the rate of convergence from the dynamic approach is tight up to constants.

In Figure 2 (b) for comparison the red line is a constant times $\frac{1}{n}$. The dashed magenta line is the optimal value minus the dashed magenta curve from (a) of the simultaneous approach. Also, the blue curves are the optimal value minus the blue curves from (a). The dotted blue line is the linear interpolation of the end points of the blue lines. As the dotted blue line, the magenta line and the red line have the same slop, this shows the $\frac{1}{n}$-convergence rate in the accuracy level $\epsilon$. The larger difference from the dashed magenta line to the red line in comparison the dotted blue line to the red line indicates the larger constant in the rate of convergence.

Both plots show that the dynamic PG algorithm converges faster than the simultaneous one. As suggested by the upper bounds the effect gets much stronger for larger $H$.

