# OpenReview forum: "Beyond Stationarity: Convergence Analysis of Stochastic Softmax Policy Gradient Methods"
_ICLR.cc/2024/Conference — ICLR 2024 poster_

### Official Review · Reviewer_Y9J6 · 2023-10-27

**Soundness:** 3 good
**Presentation:** 2 fair
**Contribution:** 3 good
**Rating:** 6
**Confidence:** 3

**Summary:**

This paper presents a thorough investigation of policy gradient (PG) methods applied to finite-horizon Markov Decision Processes (MDPs). The authors introduce two PG methods: the conventional simultaneous PG and a novel dynamic PG. The convergence of these methods is mathematically proven by leveraging two key factors: (1) the smoothness property inherent in finite-horizon MDPs, and (2) the gradient dominance condition. Furthermore, the authors extend their analysis to the model-free setting, proposing two stochastic PG methods with proven convergence analysis.

**Strengths:**

This paper is easy to follow. The dynamic gradient method proposed is new to finite-horizon MDP, and it is complimented by strong theoretical guarantees, while the smoothness property and the gradient dominance condition existing in infinite-horizon MDPs' literature are adopted into the finite-horizon case. Meanwhile, the paper also considers the character of the coefficient $\inf_{n}\min_{s\in\mathcal{S}}\pi^{\theta}(a^{\star}(s)|s)$ and further shows how to determine its lower bound, which is a new insight.

To address the model-free setting, the authors introduce two related stochastic PG methods with proven convergence results. In their comparison, both dynamic PG and stochastic dynamic PG take less steps than their counterparts.

**Weaknesses:**

One reason for not giving a higher score at this point is that the global convergence results and their analysis are actually not that surprising. As mentioned in the paper, all proofs of convergence behaviors follow the idea of the smoothness property and the gradient dominance condition, which are well used especially in [1]. While they are adopted in the finite-horizon MDP, it would surely help if the authors could discuss more about the technical challenges of the analysis.

Furthermore, the discussion on motivation in the paper appears to be somewhat weak. It would be beneficial to include a more comprehensive review of relevant literature to emphasize the significance and importance of studying finite-horizon MDPs.

[i] On the Global Convergence Rates of Softmax Policy Gradient Methods, ICML, 2020.

**Questions:**

The proofs make sense to me. I have the following minor questions:

1. As mentioned above, it would be nice if the authors could further describe the technical challenges of establishing the convergence results.

2. Why does the term '$\exists s \in\mathcal{S}$' exist in the equation of Theorem 4.2? It seems the result of Theorem 4.2 is weaker.

---

> ### Author Response · Authors · 2023-11-16
>
> Thank you very much for taking the time to review our article!
>
> >Technical challenges of the analysis
>
> Thank you for bringing to our attention that we should delve more precisely into technical difficulties. In response, we have included additional comments in the new version (will be uploaded at the end of the discussion period) which are summarized as follows.
>
> There are two different kinds of challenges within this paper:
>
> 1. Adaptation for the deterministic setting: Under the assumption of exact gradients it is correct that the proof techniques are adapted from the infinite-time setting and from this point of view the global convergence result of the simultaneous PG may be seen as "not surprising". Nevertheless, we have adapted and combined them with dynamic programming in such a way that the benefits of dynamic PG became visible.
> To the best of our knowledge, so far it was not known that one could improve on the dependence of the horizon in the finite MDP setting by exploiting dynamic programming. Moreover, it is novel that for finite-time MDPs the unpleasant model dependent constant $c$ can be made explicit under uniform initialisation of the softmax parameter. Obtaining this result also leads to an interesting future research direction: Can we adapt dynamic PG to the infinite time setting to benefit from this result also in discounted MDPs? Tackling this question is ongoing work.
>
> 2. New techniques in the stochastic setting: The main technical challenges arise in the stochastic setting where we do not assume access to exact gradients. Even in the well studied discounted scenario stochastic softmax PG has not been analysed so far without regularisation. Ding et. al. 2022 (cited) analysed the regularised discounted setting and their analysis heavily depends on the existence of an optimal parameter $\theta^\ast$ which is due to regularisation. In the unregularised problem this is not the case since the softmax parameters usually diverge to $+/- \infty$ in order to approximate a deterministic optimal solution. Consequently, their analysis does not carry over to the unregularised setting. One of the main challenges in our proof was to construct a different stopping time, independent of optimal parameters, such that the stopping time still occurs with small probability given a large enough batch size. In addition, finite time MDPs require a non stationary solution making it a harder problem compared to infinite time horizons. Thus, we need to combine the stochastic errors over all epochs (in the dynamic approach) such that an overall error at most epsilon can still be guaranteed with high probability.
>
>
> >Importance of studying finite-horizon MDP
>
> We agree that this came a bit short in the introduction and we added references motivating  finite-time MDPs in the new version.
>
> >"Why does the term $\exists s\in\mathcal{S}$ exist in the equation of Theorem 4.2?"
>
> Thank you for carefully reading! We first thought we have a typo in the result but instead the result is formulated as a negation. The theorem says that the probability is small for the event that there exists an s with large error. Taking the complement we showed that the probability is large for the event that the error is small $\forall s \in \mathcal{S}$. We see that this causes confusion and we will reformulate Theorem 4.1 and 4.2 accordingly.

---

### Official Review · Reviewer_iprP · 2023-10-31

**Soundness:** 4 excellent
**Presentation:** 4 excellent
**Contribution:** 3 good
**Rating:** 8
**Confidence:** 3

**Summary:**

This manuscript delves into the convergence characteristics of policy gradient (PG) methodologies utilizing softmax policies within the context of MDPs that have a finite horizon, denoted as $H$. A prevalent approach to implementing PG methods in finite horizon scenarios involves augmenting the state space with a temporal coordinate, subsequently training across all epochs in a simultaneous fashion. The authors have extended the convergence analysis techniques from [Agarwal et al., 2021; Mei et al., 2020], originally applied to gradient ascent algorithms with exact gradients in infinite MDPs, to the domain of simultaneous PG with exact gradients for finite horizon MDPs. This adaptation has yielded a global convergence rate of $O(H^5)$. Building upon this, the paper introduces an innovative algorithm that synergizes PG with dynamic programming, termed as dynamic PG, which notably enhances the global convergence rate of simultaneous PG to $O(H^3)$. Finally, the authors extended their findings to encompass scenarios where a stochastic gradient is utilized instead of an exact gradient.

**Strengths:**

I believe this paper is comprehensively developed and presents a balanced view on the topic at hand. Despite the intricate nature of the subject, the paper is articulately written, ensuring clarity and ease of understanding for the readers. While I have not had the opportunity to delve into Appendix C, I feel reasonably assured about the soundness of the proofs presented in the rest of the document. To me, the concept of backward updating stands out as the most intriguing aspect of this paper, marking a novel application of this technique in proving global convergence, something I have not encountered in previous papers proving global convergence result for MDPs.

**Weaknesses:**

The only limitation I can think of is the limited experimental results; however, this is understandable given its theoretical nature.

**Questions:**

Correct me if I'm wrong, my intuition is that since dynamic PG is fixing policy from $h+1$ onward, it can be viewed as a specific coordinate ascent algorithm. Do you see any way to use the backward update idea for infinite horizon non-stationary MDPs?

---

> ### Author Response · Authors · 2023-11-16
>
> Thank you very much for taking the time to review our article!
>
> >Specific coordinate ascent algorithm
>
> Great observation! Indeed, you can view the dynamic approach as specific form of coordinate ascent algorithm applied to the parameters of the collection of policies over all decision epochs. Coordinate ascent methods heavily depend on the rule of choosing the coordinates and can therefore be very slow if this rule is not chosen cleverly. In our particular setting, we suggest to choose the directions such that each epoch is trained individually backwards in time such that the scheme benefits from the spirit of dynamic programming.
>
> >Use the backward update idea for infinite horizon non-stationary MDPs
>
> Good point, using the dynamic approach for infinite horizon MDPs is an interesting direction that we have already started to think about. Bertsekas and Tsitsiklis already followed in their book "Neuro-Dynamic Programming" the idea of transferring finite-time MDP algorithms to discounted MDPs as those are also finite-time MDPs with an independent geometrically distributed horizon (which makes them in a way simpler). Combining these ideas with our approach would use the dynamic programming structure more explicitly.
> For linear dynamical systems a related approach was recently carried out in
>  "Revisiting LQR Control from the Perspective of Receding-Horizon Policy Gradient", X. Zhang, T. Basar (2023). The considered receding horizon PG algorithm (RHPG) can be seen as a relative of dynamic policy gradient proposed in our work.

---

> > ### Comment · Reviewer_iprP · 2023-11-16
> > **respond.**
> >
> > Thanks for your detailed response. Sorry I wasn't able to give any substantial feedback, but I enjoyed reading your article a lot. A lot of literature on global convergence of PG methods, are basically replicating [Agarwal et al., 2021] method in different settings. I value any fresh perspectives on the dynamics of PG method convergence in RL across different scenarios.

---

### Official Review · Reviewer_EGiX · 2023-11-01

**Soundness:** 3 good
**Presentation:** 3 good
**Contribution:** 3 good
**Rating:** 8
**Confidence:** 3

**Summary:**

This paper proposed a policy gradient algorithm for finite-horizon MDPs that are trained backward, for the benefit of explicit constant dependence in convergence guarantees and faster convergence behavior comparing to policy gradient methods that are trained simultaneously.

**Strengths:**

The idea of using dynamic programming related idea to capture the structure of finite-horizon MDPs to improve efficiency of policy gradient algorithms is interesting, and the convergence guarantees with explicit constants are established with accompanying experiments.

**Weaknesses:**

More comparisons with state-of-the-art policy gradient methods for tabular finite-horizon MDPs could have been presented to increase the convincingness of the example. It should be clearly stated why unregularized softmax is interesting to analyze by itself given the volume of past works on softmax PG and mirror descent methods.

**Questions:**

- For simultaneous PG analysis of unregularized softmax, you claimed that you are the first to analyze the global convergence rate. However, there are a number of works that study convergence rate and sample complexity for stochastic policy mirror descent, which is a more general algorithmic framework that encompass softmax as a special case [1].
- Given the plethora of works on softmax policy gradient in the tabular case and discussions on its limitations [2], what would be practical benefits of using softmax aside from the fact that it gives rise an unconstrained optimization problem?
- In the unregularized MDP problem the optimal policy can be deterministic, in which case softmax parametrization does not cover all the possible policies and therefore is incomplete. Can you elaborate on the benefit of using an incomplete policy class in the tabular case here?
- Is it possible to compare with Guo et al. 2022 (appeared in original reference list of the paper) performance numerically?


[1] Alfano, Carlo, Rui Yuan, and Patrick Rebeschini. "A novel framework for policy mirror descent with general parametrization and linear convergence." arXiv preprint arXiv:2301.13139.
[2] Li, Gen, Yuting Wei, Yuejie Chi, Yuantao Gu, and Yuxin Chen. "Softmax policy gradient methods can take exponential time to converge." In Conference on Learning Theory, pp. 3107-3110. PMLR, 2021.

-----------

Post-rebuttal: I thank the authors for their clarification and agree that unregularized softmax would be of interest for theoretical analysis. I personally disagreed with the statement in the paper "...very little is known about convergence to global optima even in the discounted case" per the discussion before. I have increased my score.

---

> ### Author Response · Authors · 2023-11-16
>
> Thank you very much for taking the time to review our article!
>
> > Why unregularised softmax is interesting
>
> We agree that there are good reasons to consider regularised problems; for example in Mei 2020 (cited) regularisation results in a stronger PL inequality and one can guarantee faster convergence. Nevertheless, analysing a regularised problem always results in convergence to the regularised optimum which introduces a bias in the overall error compared to unregularised problems (see e.g. Agarwal 2020 (cited), Mei 2020 or also reference [1, Thm 4.3] from your review). There is another way to achive convergence to the unregularised optimum by reducing the regularisation over time.
> This is non-trivial, we (and certainly others) are currently working on it. In order to do so, one needs to balance the speed of reducing the regularization parameter such that convergence to the unregularised optimum can still be achieved. However, as already mentioned in Mei 2020 this is a challenging task and to the best of our knowledge still an open question.
> Since our main motivation is to introduce the dynamic PG variant, in order to achieve a satisfying convergence towards the true solution of the original unregularised problem we consider unregularised softmax in this paper.
>
> >Question 1
>
> Thank you very much to pointing us to the reference on stochastic policy mirror descent. Indeed, we agree that there is plenty of work regarding (softmax) PG for stationary policies. However, to the best of our knowledge most of these results are with respect to the discounted infinite horizon setting, but only a few results as in Guo 2022 in the finite horizon. When claiming that we are the first to analyse the simultaneous PG setting, we mean especially for *finite-time horizon* MDPs. Next to your reference there are for example Agarwal 2020 and Mei 2020 who analysed the global convergence rate of softmax in the discounted setting. Results for the discounted setting are not applicable in the finite horizon case without discounting as the rates depend on the factor $(1-\gamma)^{-1}$. To the best of our knowledge convergence rates for policy mirror descent are also with respect to the discounted infinite time scenario (please correct us if we missed something here). It would be a very interesting future research direction to consider this approach in the finite-time setting.
>
> >Question 2
>
> In practice, the softmax parametrisation is certainly not the optimal choice. However, it allows for rigorous mathematical analysis to highlight the advances of dynamic policy gradient in comparison to the simultaneous approach. This paper should not be seen as limited to the softmax case, but more like a kick-off to analyse a new approach which is beneficial in many scenarios. Here are two examples:
> 1. After the submission deadline we came across the paper "Revisiting LQR Control from the Perspective of Receding-Horizon Policy Gradient", by X. Zhang, T. Basar (2023). Their receding horizon PG algorithm can be seen as a special case of dynamic PG for linear quadratic regulator problems. This provides a different scenario where dynamic PG is beneficial in the specific linear setting.  Moreover, they used the dynamic approach also in the infinite horizon problem and it is ongoing work to exploit this also in the MDP setting.
> 2. In future research we plan to analyse finite time MDPs with continuous action space under Fisher-non-degenerate policies. In this setting one can also guarantee gradient domination, and for discounted MDPs convergence has been analysed by Fatkhullin et. al (2023) (cited). We expect that using dynamic PG within this policy class for finite horizon problems will again result in better convergence.
>
> >Question 3
>
> Good observation! It is correct that the softmax class does not contain deterministic policies. Still, softmax has the ability to approximate any policy (also deterministic ones) arbitrarily close. Therefore, the closure of the softmax family is a complete policy class.
> As an MDP problem is continuous in the policy, this is sufficient for finding an optimal solution.
> For example in the optimal stopping problem considered in the numerical experiment (appendix E) the solution is a unique deterministic policy. The softmax policy is able to approximate the deterministic policy and guarantees an error at most epsilon.
>
> >Question 4
>
> Interesting thought. We think this is not possible due to the following two reasons:
> 1. Guo et. al. 2022 analyse policy gradient in the finite time setting, but their algorithms learn a stationary policy to solve the finite-time problem which is in general not sufficient. In the results (Thm 15 and Thm 19) you can see that due to the stationarity their "convergence rate" includes a bias term; implying that they are not able to show convergence towards the global optimum.
> 2. Their results only hold over the minimum of gradient steps, while our analysis provides a last iterate convergence result.

---

### Meta-Review · Area_Chair_GHFe · 2023-12-04

**Metareview:**

This paper studies the softmax polity gradient method for finite horizon MDP problems. An innovative algorithm that enhances PG with dynamic programming is proposed and a convergence to global optimal solution is proved, for both deterministic and stochastic settings. Though there have been a rich number of literatures studying the convergence of softmax PG for infinite horizon MDP, the related works for finite horizon cases are rare. In addition, it is also an interesting finding that incorporating dynamic programming tricks to the PG method can improve the convergence and complexity result. Therefore, the AC believes that the paper has the merit to be published in ICLR. However, as policy search in a backward manner like dynamic programming is a classical method in finite-horizon MDP, see e.g., https://proceedings.neurips.cc/paper/2003/file/3837a451cd0abc5ce4069304c5442c87-Paper.pdf and http://proceedings.mlr.press/v32/scherrer14.pdf . Though there has not been related works that implement the one-step policy search in PSDP, it is still worth adding the discussion of how the proposed method relates to PSDP. The authors should add these connections in later revision of the paper.

Another problem with the paper is that the claimed advantage over PG comes from an upper-bound-to-upper-bound comparison, instead of a rigorous separation between lower and upper bounds. The authors should discuss this point in the revision.

**Justification For Why Not Higher Score:**

Though the paper fills in the blank of softmax policy gradient method for finite-horizon MDP, the results are not surprising and can be expected.

**Justification For Why Not Lower Score:**

The paper fills in the blank of softmax policy gradient method for finite-horizon MDP, which has not been discussed before.

---

### Decision · Program_Chairs · 2024-01-16

Accept (poster)